# dsRNAi-mediated silencing of *PIAS2beta* specifically kills anaplastic carcinomas by mitotic catastrophe

Joana S. Rodrigues [1,14,15], Miguel Chenlo [1,15], Susana B. Bravo [2], Sihara Perez-Romero[1], Maria Suarez-Fariña[1], Tomas Sobrino [3,4], Rebeca Sanz-Pamplona[5,6], Román González-Prieto [7,8], Manuel Narciso Blanco Freire[9], Ruben Nogueiras [10], Miguel López [11], Laura Fugazzola[12], José Manuel Cameselle-Teijeiro [13,16] ✉ & Clara V. Alvarez [1,16] ✉

The E3 SUMO ligase *PIAS2* is expressed at high levels in differentiated papillary thyroid carcinomas but at low levels in anaplastic thyroid carcinomas (ATC), an undifferentiated cancer with high mortality. We show here that depletion of the *PIAS2* beta isoform with a transcribed double-stranded RNA–directed RNA interference (*PIAS2b*-dsRNAi) specifically inhibits growth of ATC cell lines and patient primary cultures in vitro and of orthotopic patient-derived xenografts (oPDX) in vivo. Critically, *PIAS2b*-dsRNAi does not affect growth of normal or non-anaplastic thyroid tumor cultures (differentiated carcinoma, benign lesions) or cell lines. *PIAS2b*-dsRNAi also has an anti-cancer effect on other anaplastic human cancers (pancreas, lung, and gastric). Mechanistically, PIAS2b is required for proper mitotic spindle and centrosome assembly, and it is a dosage-sensitive protein in ATC. PIAS2b depletion promotes mitotic catastrophe at prophase. High-throughput proteomics reveals the proteasome (PSMC5) and spindle cytoskeleton (TUBB3) to be direct targets of PIAS2b SUMOylation at mitotic initiation. These results identify *PIAS2b*-dsRNAi as a promising therapy for ATC and other aggressive anaplastic carcinomas.

Thyroid carcinomas (TC) derived from follicular epithelium are the fifth most common cancer in women, and the second in women aged 20–59. The majority of cases are differentiated TCs, either papillary TC (PTC, ~85%) or follicular TC (FTC, ~10%)[1]. Some of these TCs evolve to high-grade follicular cell-derived non-anaplastic thyroid carcinoma (differentiated high-grade thyroid carcinoma [DHGTC] and poorly differentiated thyroid carcinoma [PDTC]), which are more aggressive but still maintain some differentiation[1,2]. Anaplastic (undifferentiated) thyroid carcinoma (ATC) accounts for 1% of TC in the USA and is the most aggressive cancer in humans[1,3]. The 8th AJCC staging classifies all ATCs as stage IV[3–5]. Patients are more likely to be female (F:M 2:1), aged 60 or older, and have a fast-growing neck mass. Over half have metastatic disease at diagnosis. In general, ATC patients die by 6 months after diagnosis, with a 100% disease-specific mortality. Only extensive surgery with multimodal therapy offers any survival benefit, but this comes at the cost of quality of life[6]. Adjuvant chemotherapy or radiotherapy is ineffective for long-term survival[3].

Recently, an effort to apply precision therapy in ATC led to the approval of dabrafenib plus trametinib as a first-line therapy, in a neoadjuvant setting combined with surgery for patients with ATC who have a *BRAF* V600E (BRAF+) mutation; this therapy extends lifespan by more than 1 year[3,5]. Unfortunately, only 25–40% of patients with ATC are BRAF+, and post-treatment can induce resistance through

---

subsequent additional BRAF mutations[7,8]. Ongoing clinical trials test other combinations with surgery[9–12].

ATCs usually conserve expression of cytokeratins and patchy E-cadherin. Regarding thyroid-specific markers, ATC lose expression of NKX2-1 (TTF1), Na+/I− symporter (NIS), thyroglobulin (TG), and thyroperoxidase (TPO), but half of cases maintain PAX8 expression[1,13,14]. They present > 50% Ki67 index, are aneuploid, and have an increased tumor mutational burden as compared with differentiated thyroid carcinomas[11]. ATC cases are characterized by a high mutation burden in multiple genes[15–18]. These mutations are heterogeneous across ATC cases, with few being shared among all tumors. Despite this diversity, certain genes exhibit a high prevalence of mutations like both early driver events (e.g., BRAF V600E, 10–40%; RAS mutations, 10–50%) and late molecular events (e.g., TP53 mutations, 40–80%; TERT promoter mutations, 30–75%)[19,20]. However, despite the great variability in the mutational profile between patients, the tumors share common activated signaling pathways[19,20], and the percentage of cells sharing the common mutational profile within each ATC tissue is high[21].

Our group has developed the specific medium h7H, which contains all components (ions, hormones, metabolites) adjusted to values found in human serum[22,23]. We established h7H thyroid cultures derived of surplus tissue from patients submitted to thyroid surgery. The cultures are >95% epithelial and maintain expression of thyroid phenotype (TTF1, PAX8, TG, TPO, and cytokeratins). Since 2012, we have generated the TIROCHUS collection, which includes tissue and culture samples obtained in parallel from consecutive patients. These samples are classified according to their pathological diagnostic as normal thyroid (NT), hyperplasia due to thyroid follicular nodular disease/multinodular goiter (MNG), follicular adenoma (FA), PTC, FTC, DHGTC/PDTC, or ATC carcinomas. Paired samples with different diagnoses (e.g. PTC/MNG or PTC/NT) are obtained from the same patient. Our long-term goal when establishing this collection was to compare cancer cells to normal/benign cells, with both growing in vitro in the same conditions, to identify specific cancer targets and distinguish them from proliferation-related targets (which commonly occur in cancer cultures).

The PIAS family (PIAS1-4) are nuclear, zinc-binding proteins that contain a Siz/PIAS (SP)-RING domain, which functions as E3 SUMO ligase. Pro- and anti-cancer functions have been proposed for this family (including immune system interactions, DNA repair, and signal transduction) via their modulation of STAT and other transcription factors, whereby each family member has specific substrates[24–26]. In recent years, the Xenopus and Caenorhabditis homologs of PIAS1 and PIAS4 have been demonstrated to participate in mitosis at chromatid centromeres at the kinetochores. These proteins are essential for sister chromatid segregation at the metaphase-anaphase transition[27–31]. The role of SUMOylation in mitosis has also been established[32,33].

Here, we analyze data derived from an initial 2D-proteomic study comparing benign to cancer thyroid cultures. We find that the PIAS2 protein significantly increased in differentiated PTC cultures and thus have studied PIAS2 and its isoforms, across the entire range of follicular-lineage thyroid neoplasms, to see if it plays a role in thyroid cancer. We observe that PIAS2 is a dosage-sensitive protein in ATC with an essential role at the mitotic spindle SUMOylating proteasome and the microtubule proteins. Strikingly, RNA interference using in vitro transcribed, double-strand RNA against the mRNA of the PIAS2 isoform beta (herein, *PIAS2b*-dsRNAi) kills ATC cells growing in full medium in vitro through mitotic catastrophe, and reduces orthotopic patient-derived xenografts (oPDX) in vivo. In contrast, survival or growth of non-anaplastic thyroid primary cultures from other origins (benign, differentiated carcinomas, metastasis) was not affected by *PIAS2b*-dsRNAi. On the other hand, non-thyroid carcinoma cells with the same three characteristics− anaplastic, aneuploid and undifferentiated−are also killed by *PIAS2b*-dsRNAi. A molecular

mechanism for *PIAS2b*-dsRNAi implicates untimely proteasome activation at prophase, and centrosome/spindle alterations, in cell death. Thus, PIAS2b is an essential mitotic protein in anaplastic cancers that could be targeted by RNAi therapies in the future.

## Results

### Identification of PIAS2b as a target protein in thyroid cancer

Using h7H culture conditions[22,23], we searched through 2D-gel electrophoresis for differentially expressed proteins that were enhanced in PTC cultures as compared to normal or benign thyroid cultures (Supplementary Fig. 1a–d, and proteomic data in Supplementary Data 1). One of the proteins was Protein Inhibitor of STAT2 (PIAS2; previously called ARIP3 or PIASX), which has not been previously studied in human thyroid. We therefore retrieved RNA-seq data from the TCGA consortium and HPRD for *PIAS2*[34–36]. Strikingly, *PIAS2* mRNA overexpression correlated with a poorer prognosis in PTC, with around 20% lower chance of 5-year survival (Supplementary Fig. 1e). Thus, we explored if PIAS2 has a specific role in thyroid cancer.

*PIAS2* is a complex gene, with 14 exons (Fig. 1a), and numerous transcription isoforms (16 in Ensembl but 29 in NCBI, with only partial overlap) (Supplementary Data 1). Few of the RNA isoforms are protein-coding, and only two express full-length protein: *PIAS2 beta* (*PIAS2b*) and *PIAS2 alpha* (*PIAS2a*) (Fig. 1a). Both proteins containing an N-terminal SAP domain that includes an LxxLL motif (as a potential DNA binding motif); ii) the PINIT domain that serves as scaffold for the E2-SUMO ligase; iii) a SP-RING domain, which is a zinc-finger domain that functions as an enzymatic E3 SUMO ligase; iv) a SIM (SUMO-interacting motif); and v) NLS consensus sequences (Fig. 1b). The C-terminal domain differs in both isoforms through alternative splicing of exons 13–14 (from a.a. 551), such that PIAS2b contains an additional serine-rich domain that includes a poly-serine stretch (SSSSSRS) (Fig. 1a, b).

Custom-designed, isoform-specific RT-qPCR (Supplementary Data 1) showed that in patient tissues, both *PIAS2b* and *PIAS2a* were expressed at significantly higher levels in PTCs as compared to NT or MNG hyperplastic lesions (Fig. 1c). In contrast, *PIAS2b* was significantly downregulated in the two most aggressive groups of thyroid cancer, PDTC and ATC (Fig. 1c). When PTC/NT or PTC/benign paired same-patient samples were compared, *PIAS2b* expression was consistently doubled in all PTC samples (Fig. 1d). We thus focused our study on PIAS2b.

Only 2 of 7 different antibodies for PIAS2 could be validated for both Western blot and immunostaining in low-expression anaplastic cells (Supplementary Fig. 1f; mPIAS2 and rPIAS2b epitopes shown in Fig. 1b). With a classic extraction buffer (e.g. including phosphatases inhibitors), mPIAS2, able to detect an intense band of exogenous Flag-PIAS2a (Uniprot expected MW 63,396 Da), did not detect endogenous PIAS2a in any ATC cell line (Supplementary Fig. 1f). Also, mPIAS2 detected well exogenous fusion PIAS2b proteins, and detected two endogenous PIAS2b isoforms (Uniprot expected MW 68,240 Da), intensely p75- and less intense p95-PIAS2b. rPIAS2b was better at detecting p95-PIAS2b, and less well p75-PIAS2b (Supplementary Fig. 1f). Thus, we assume that the molecular weight of the two PIAS2b bands detected is due to different levels of PTM on PIAS2b.

To investigate the possibility that rPIAS2b mostly detected phosphorylated PIAS2b, we transfected a small amount of EGFP-PIAS2b and lysed the cells with a buffer containing cysteine protease inhibitors (SUMO-lysis buffer). Input lysates were incubated with GFP-binding beads (GFP-Trap) and divided into two halves, one of which was incubated with lambda phosphatase. The western blot showed that mPIAS2 was the most sensitive antibody, capable of detecting at the Input exogenous EGFP-PIAS2b, as well as the two endogenous bands p95-PIAS2b and p75-PIAS2b. rPIAS2b again preferentially detected endogenous p95-PIAS2b, and weakly the other two bands.

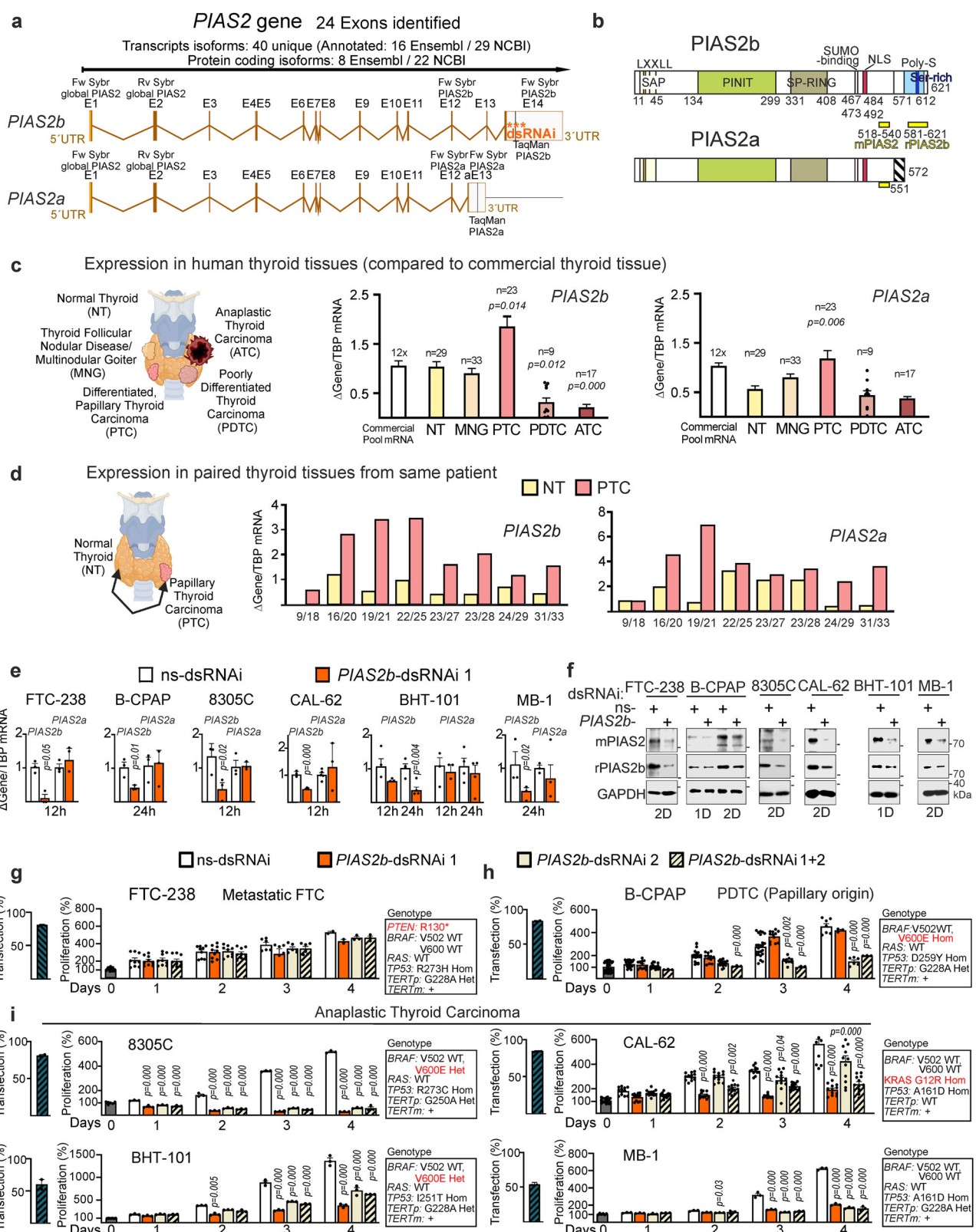

GFP only detected EGFP-PIAS2b after the GFP-Trap, where all three antibodies were the same. Dephosphorylation reduced the weight of EGFP-PIAS2b, but this did not affect the detection of the three antibodies (Supplementary Fig. 1g). These results suggest that p95-PIAS2b has other post-translational modifications (PTMs) in addition to phosphorylation, and that these PTMs affect detection by rPIAS2b. Specificity of both antibodies was confirmed through

immunofluorescence of endogenous or transfected proteins in cultured cells (Supplementary Figs. 1h, i). Both antibodies passed validation of transfected EGFP-C1-hPIAS2 cells (Supplementary Fig. 1i). Additionally, only rPIAS2b worked in FFPE immunohistochemistry, showing nuclear staining in all thyroid pathological samples (Supplementary Fig. 1j, k), while mPIAS2 stained some cells in ATC cryosections (Supplementary Fig. 1l).

**Fig. 1 | *PIAS2b*-dsRNAi blocks growth of thyroid cancer cell lines from ana-plastic origin but not those from poorly differentiated/ metastatic differ-entiated origin. a** Schematic view of human *PIAS2* exon/intron gene distribution to give the two full-length isoforms. Location of each primer set for RT-qPCR, and of the dsRNAi at the 3′ UTR of the beta isoform, is indicated. **b** The two PIAS2 protein isoforms contain similar domains (SAP and LXXLL; PINIT, SP-RING, SUMO-binding domain, nuclear-localization signal NLS) except at the C-terminal domain, where the PIAS2b isoform contains the serine (S)-rich domain, including a poly-S stretch. Yellow bars indicate the epitopes of the two antibodies; while mPIAS2 detects both isoforms, rPIAS2b detects only PIAS2b. **c, d** mRNA expression of *PIAS2b* and *PIAS2a* in thyroid tissues (NT, normal thyroid; MNG, multinodular goiter; PTC, papillary thyroid carcinoma; PDTC, poorly differentiated thyroid carcinoma; ATC, anaplastic thyroid carcinoma). Benign/PTC-paired tissues from a single patient are shown in **d**. **e, f** RT-qPCR of *PIAS2* isoforms and protein detection in thyroid cancer cell lines

transfected with non-target sequence (ns-dsRNAi) or *PIAS2b* sequence 1 (*PIAS2b*-dsRNAi-1). Original cancers for the cell lines were: FTC-238, metastatic FTC; B-CPAP, PDTC of papillary origin; 8305C, CAL-62, BHT-101, and MB-1 cell lines, ATC. **g–i** Time-course of cell growth in similarly transfected cell lines comparing *PIAS2b*-dsRNAi-1, −2, or both. Transfection efficiency is indicated by the hatched-bar at the left. Genotype of each cell line is indicated with the driver mutation highlighted (note that for TERT, TERTp is the promoter, and TERTm, the mRNA expression). **c, d** n indicated in the figure; **e** $n = 3$ independent experiments, **g** $n = 3$ (day 4)- 9, **h** $n = 3$ (combination)-12 (sequence 1)- 16 (sequence 2) independent experiments; **i** $n = 3$ or $n = 12$ (CAL-62) independent experiments. **c** Two-sided Kruskal–Wallis; **e** one-sided Mann–Whitney; **g–i** two-sided two-way ANOVA with repeated mea-sures. Bar indicates means ± SEM; when significant the exact *p* value is indicated in the figure. **c, d** Own design from Biorenders templates Source data are provided as a Source Data file.

## *PIAS2b*-dsRNAi has anti-cancer effects in anaplastic thyroid cancer cell lines and patients' cultures, but not in normal/benign or differentiated thyroid cancers

We next explored the function of PIAS2b by blocking its expression through RNA interference. No commercially available siRNAs specific against PIAS2b isoform were available (note that specificity was espe-cially important in light of the many non-translated isoforms ascribed to this gene). Thus, we turned to T7-transcribed double-stranded RNA interference (dsRNAi), which we have effectively used previously to reduce target proteins in primary cultures[37]. Five sequences were found in the *PIAS2b* 3′ UTR (exon 14) with BLAST scores for isoform specificity (asterisks in Fig. 1a and Supplementary Data 1). We synthe-sized dsRNAs corresponding to sequences 1 or 2, as well as a control, non-target sequence in the human genome (ns-dsRNAi) (Supplemen-tary Data 1).

We selected several cell lines from three types of human thyroid cancer: i) B-CPAP, which comes from a papillary-originated PDTC; ii) FTC-238, from a metastatic differentiated FTC; and iii) 8305C, CAL-62, BHT-101, and MB-1, from ATC patient samples. All cell lines expressed high levels of *TERT* mRNA independently of promoter mutation, can-cer driver gene, or pathology (Supplementary Fig. 1m). All cell lines expressed various levels of global *PIAS2*, *PIAS2b*, and *PIAS2a* mRNA (Supplementary Fig. 1n).

dsRNAis were transfected with comparative efficiency (>55% in all cases, with some >95%; Supplementary Data 2). For all cell lines, *PIAS2b*-dsRNAi specifically reduced *PIAS2b* mRNA and PIAS2b protein expression as compared to ns-dsRNAi, without affecting levels of *PIAS2a* (Fig. 1e, f; quantification of Western blots in Supplementary Fig. 1o). Cells were assessed for the following four days while growing in full growth medium (Fig. 1g–i). As expected, cells were not affected by not specific ns-dsRNAi (as compared to empty transfection). Strik-ingly, while *PIAS2b* RNAi using dsRNA-1 had no effect on the growth of follicular (FTC−238) or papillary (B-CPAP) thyroid carcinoma cell lines (Fig. 1g, h), it blocked growth of the four ATC lines (Fig. 1i). The anti-growth effect of silencing *PIAS2b* in ATC cell lines was dose-responsive, could be replicated using different transcribed batches of dsRNA-1, and was effective over weeks with repeated transfections (Supple-mentary Fig. 1p–r). Using a combination of dsRNA-1 and -2 did not increase the anti-growth effects of dsRNA-1 on ATC cells (Fig. 1i). *PIAS2b* RNAi using dsRNA-2 had partial anti-growth effect in B-CPAP but a lower effect than using dsRNA-1 in the anaplastic lines (Fig. 1h). Using nucleofection rather than lipids for transfection did not alter the result of the *PIAS2b*-dsRNAi: dsRNA-1 specifically blocked growth of ATC cell lines but not of FTC-238 or B-CPAP cell lines (Supplementary Fig. 1s–u). Therefore, we used dsRNA-1 for the remaining silencing experiments (called *PIAS2b*-dsRNAi).

We could not attribute the specific anti-growth effects of *PIAS2b*-dsRNAi in anaplastic lines to either: (i) a specific origin of the patient (e.g, France, Swiss, Germany, Hungary, or Japan); or (ii) to the muta-tional status (B-CPAP, FTC-238, and the four ATC lines have mutated

p53; the *BRAF* V600E mutation is homozygous in B-CPAP, and het-erozygous in BHT-101, but absent in the other lines; only CAL-62 has a RAS mutation; and all lines but one (CAL-62) have a *TERT* promoter mutation) (Fig. 1g–i and Supplementary Data 2; cancer driver genes highlighted in red; BRAF V502 SNP is wild type in all cell lines; infor-mation available at Cellosaurus and confirmed at our lab by Sanger). All cell lines expressed TERT and various levels of PIAS2 mRNAs (Sup-plementary Fig. 1m, n).

Overexpression of FLAG-hPIAS2a, FLAG-hPIAS2b, or EGFP-C1-hPIAS2b abolished growth of ATC cell lines but did not affect normal thyroid T-NT2 or PDTC-PTC B-CPAP cells (Supplementary Fig. 1v, w). Time-lapse experiments revealed condensed EGFP+ cells over-expressing hPIAS2b that were arrested and later dead (Supplementary Fig. 1v). These results revealed PIAS2 as a dosage-dependent protein in anaplastic thyroid cancer cells, since both its downregulation (*PIAS2b*-dsRNAi) and its overexpression (FLAG-hPIAS2a, FLAG-hPIAS2b, or EGFP-C1-hPIAS2b) were deleterious. We performed a dose-response titration and found that, a 20-times lower amount of FLAG-hPIAS2b allowed cell growth (Supplementary Fig. 1x).

If specific, the anti-growth effect of *PIAS2b*-dsRNAi, designed at the 3′ UTR, should be rescued by exogenous PIAS2b expression. We tested the effects of non-toxic amounts of PIAS2b, PIAS2a and point-mutant PIAS2b-C362A SUMO-dead, by transient transfection of a mix of pcDNA3 (empty vector): PIAS2 vector in a proportion of 19:1. This amount of FLAG-hPIAS2b or EGFP-hPIAS2b, but not of FLAG-hPIAS2a was still able to rescue the anti-growth effect of *PIAS2b*-dsRNAi (Fig. 2a, b), confirming its specificity. Moreover, SUMO-dead PIAS2b C362A mutants, FLAG-hPIAS2b-C362A or EGFP-hPIAS2b-C362A, did not rescue *PIAS2b*-dsRNAi effect, indicating that PIAS2b SUMOylase activity was required (Fig. 2a, b).

We next compared our in vitro transcribed *PIAS2b*-dsRNAi with other sources of RNA interference (Supplementary Fig. 2a). Chemically synthesized commercial siRNAs targeted for PIAS2 from different sources (On-Target from Dharmacon, Custom Select from Life-Tech-nologies), which used normal or structurally modified nucleotides for enhanced half-life, were transfected either as a set, or each single one at full doses; none of them were able to downregulate PIAS2b nor affected growth of any ATC cell lines (Supplementary Fig. 2b).

To compare the dsRNAi with short RNA (shRNA) hairpins (Sup-plementary Fig. 2a), we prepared lentivirus with the commercial vector TRIPZ Tet-on bearing scramble sequence (T-*Scr*-shRNA) or T-*PIAS2* shRNA, designed against various *PIAS2* RNA isoforms (Supplementary Data 1). Infected 8305C cells with both high-levels and sufficient-levels (see Methods) of T-*PIAS2* shRNA expression decreased their cell numbers after addition of doxycycline in correlation to decreased *PIAS2* mRNA and protein expression (Supplementary Fig. 2c–f). Time-lapse experiments after addition of doxycycline and transient trans-fection of cells with histone H2-GFP (to follow cell divisions) showed accumulation of arrested T-*PIAS2* shRNA GFP+ condensed/dead cells, while T-*Scr*-shRNA lost GFP over time as cells divided (Supplementary

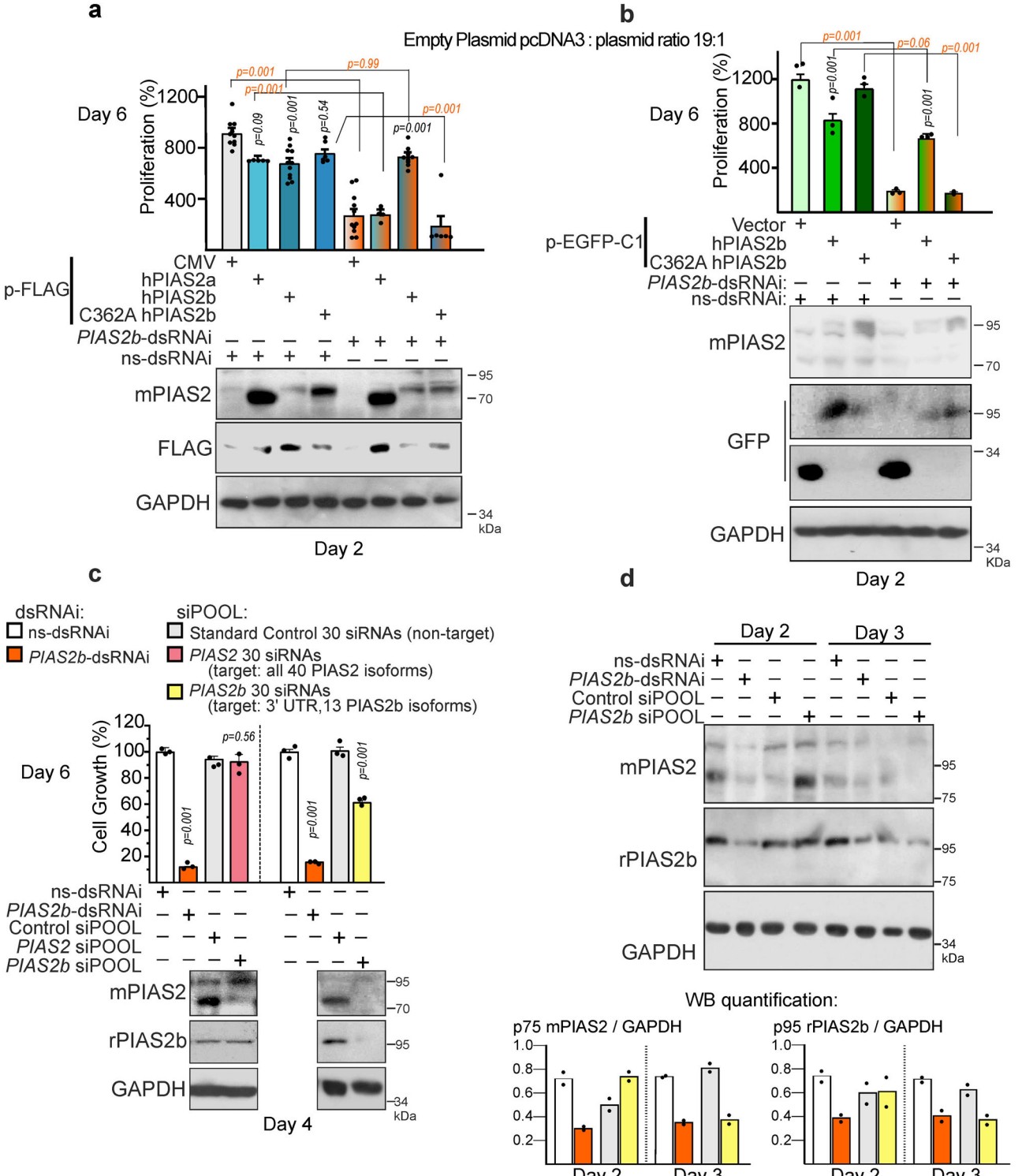

**Fig. 2 | *PIAS2b*-dsRNAi deleterious action in growth is rescued by PIAS2b SUMOylation activity. Commercial *PIAS2b* siPOOL reproduces the effect with less potency. a, b** Inhibitory effect of *PIAS2b*-dsRNAi 1 in 8305C cells is rescued by FLAG-PIAS2b or EGFP-PIAS2b, but not by FLAG-PIAS2a, FLAG-PIAS2b-C362A or EGFP-PIAS2b-C362A point mutant (PIAS2b SUMO-DEAD) cDNA transfection. As shown above, *PIAS2b*-dsRNAi is designed at the 3′ UTR not present in the expression plasmid. **c** Commercial siPOOL against all PIAS2 isoforms is not effective reducing protein or cell growth in 8305C cells. Custom-ordered siPOOL targeting *PIAS2b*

specifically (similar isoforms as *PIAS2b*-dsRNAi) reduces protein expression and cell growth, although less efficiently than *PIAS2b*-dsRNAi at Day 6. **d** Western blots comparing both treatments at day 2 and 3 post-transfection show delayed protein suppression by *PIAS2b* siPOOL. **a** $n = 8$ independent experiments; **b, c** $n = 3$ independent experiments; **d** $n = 2$ independent experiments. **a–c** Two-sided one-way ANOVA. Bar indicate means ± SEM; when significant the exact $p$ value is indicated in the figure. Source data are provided as a Source Data file.

Fig. 2e). However, after 10 passages, the *T-PIAS2* shRNA cells lost the ability to reduce PIAS2 expression and the anti-cancer effect (Supplementary Fig. 2f), and this prevented further use of these cell populations.

We also compared the dsRNA with antisense oligonucleotides (ASOs) for RNAi (Supplementary Fig. 2g–i)[38,39]. ASOs are short monocatenary locked nucleic acid (LNA) gapmer, modified oligonucleotides that target specific intron/exon sequences while the mRNA is being transcribed in the nucleus, inducing its degradation through nuclear RNAse H, independent of the cytoplasmic RISC complex[40]. Labeled off-target control as-LNA-GapmeR 56 FAM revealed high penetrance efficiency (Supplementary Fig. 2g). Nevertheless, neither PIAS2b-as-LNA-GapmeR 1 (targeted to the *PIAS2b* UTR) nor PIAS2-as-LNA-GapmeR 2 (the recommended commercial sequence that globally targets many *PIAS2* isoforms) had any effect on cell growth (Supplementary Fig. 2h). Critically, neither ASO markedly downregulated PIAS2b protein expression (Supplementary Fig. 2i).

siPOOLs are a set of 30 chemically synthesized siRNAs with different sequences, all directed against the same gene, used at less than 5 nM concentration for the POOL, so that at reduced concentration, each siRNA avoids the off-target effects of siRNAs at high concentrations (Supplementary Fig. 2a). However, when they all act on the same RNA, their efficiency soars[41]. We compared human PIAS2 siPOOL from the catalog against recommended siPOOL non-target control (Fig. 2c). *PIAS2* siPOOL failed to have any effect on cell growth, because it also failed to reduce protein expression. We thought that the existence of so many different RNA isoforms in *PIAS2* gene might be counterintuitive to the efficiency of a pool of siRNAs. We therefore commissioned a custom siPOOL against the 13 *PIAS2b* RNA isoforms which were targeted by our *PIAS2b*-dsRNAi. As shown by the yellow bar in Fig. 2c, Custom *PIAS2b* siPOOL at the same low concentrations reduced protein expression and ATC growth at day 6. The PIAS2b siPOOL was less efficient than *PIAS2b*-dsRNAi in concordance with its delayed ability to reduce PIAS2b protein, starting at Day 3 instead of at Day 2 like *PIAS2b*-dsRNAi (Fig. 2d).

Next, we performed similar experiments using patient-derived primary thyroid cell cultures. We refer to the culture using T- corresponding to its similar TH- (tissue original samples). These cultures are >95% epithelial (cytokeratin+) and maintain the characteristic presence or absence of markers as in their original cancers (Supplementary Fig. 3a). Whenever possible, we cultured in parallel cancer and benign cells from the same patient. Appearance and growth were visibly different, as shown for T-UC1, T-UC2, and T-MNG94, or T-UC3 with T-MNG178 and T-MNG179 (Supplementary Fig. 3b, c). Patient characteristics, pathology, and genetic mutations found in the original tissues are listed in Supplementary Data 2.

The primary thyroid cultures (Fig. 3a), and concordant with their tissues (Fig. 1c), expressed high levels of *PIAS2b* over *PIAS2a* mRNA in PTC cultures (T-PC), and reduced expression of both in ATC cultures (T-UC), compared to normal (T-NT) or nodular goiter (T-MNG) cultures. Transfection of *PIAS2b*-dsRNAi specifically reduced *PIAS2b* mRNA and PIAS2b protein expression (Fig. 3b, c–f) in all culture types. Cells were maintained in full growth medium for six days, and cell growth was compared to non-target ns-dRNAi (Fig. 3g, h). *PIAS2b* silencing had no remarkable effect on growth of T-NT, T-MNG, or T-PC from different mutational genotype, including one lymph node metastasis (T-M19). In one FTC culture (T-FC7), there was 40% growth inhibition (Fig. 3g). Consistently, *PIAS2b* silencing induced >80% growth inhibition in five T-UC cultures obtained from three different ATC patients (patient 1, T-UC1, T-UC2; patient 2, T-UC3; patient 3, T-UC7, T-UC8) (Fig. 3h).

T-NT2 is a normal thyroid culture from TIROCHUS that spontaneously immortalized[23], which expresses telomerase mRNA but is negative for the usual thyroid cancer gene mutations (Supplementary Fig. 1m and Supplementary Data 2). We performed a time-course

experiment in T-NT2, the T-MNG94 benign correlates of T-UC1 and T-UC2, and all five anaplastic cultures. Transfection efficiency was comparable (Fig. 3i–m). While normal T-NT2 or T-MNG94 grew progressively following *PIAS2b* silencing (Fig. 3i–j), the five T-UC cultures reduced growth progressively (Fig. 3k–m). This reduction suggested cell death induced by *PIAS2b* silencing.

In summary, based on these results, we concluded that PIAS2b is essential in ATC cells. We believe that the use of specific sequences designed and transcribed in vitro as *PIAS2b*-dsRNAi, as well as the TRIPZ lentiviral system and *PIAS2b*-specific siPOOL, provides strong evidence that PIAS2b is a valid target. In addition, it further validates our specific *PIAS2b*-dsRNAi, since in vitro transcription preparation somehow makes the RNA interference effect more powerful in ATC cells when comparing both strategies (Fig. 2b, c). The replacement experiments with the different PIAS2b constructs also provide strong evidence that is PIAS2b SUMOylation activity what is essential in ATC cells. Thus, we continued unveiling the mechanisms of cell death triggered by *PIAS2b*-dsRNAi.

### *PIAS2b* silencing induces mitotic catastrophe in ATC cells

*PIAS2b*-dsRNAi not only blocked cell growth but also reduced cell numbers with days. In flow cytometry, *PIAS2b* silencing increased the sub-G1 fraction, indicative of cell death (Supplementary Fig. 3d). To explore the mechanisms of cell death, we co-incubated 8305C dsRNAi transfected cells with an inhibitor for caspase 3 (Ac-DEVD-CHO), caspase 9 (Ac-LEHD-CHO), or RIPK1 (necrostatin, Nec-1S) and observed if *PIAS2b* silencing–induced cell death was prevented. Strikingly, no inhibitor changed the growth patterns of cells after *PIAS2b*-dsRNAi or ns-dsRNAi (Supplementary Fig. 3e–g). Thus, in ATC cells *PIAS2b*-dsRNAi did not induce apoptosis or necroptosis, which are two of the most frequent cell death types.

We next performed time-lapse tracing experiments in the presence of Sir-DNA, a far red wavelength fluorescent DNA-binding dye that allows long-term follow-up without oxidative toxicity due to laser excitation, to follow cell growth/death. Specifically, when cells reach mitosis, their mitotic (condensed) chromosomes at G2/M nuclei are intensely fluorescent; after cytokinesis, nuclei return to low intensity. Replicate wells in two ATC cell lines (8305C, CAL-62), two ATC primary cultures (T-UC1, T-UC2), are shown in Supplementary Fig. 4a, b. A representative field in ATC (8305C, T-UC1) compared to the PDTC (papillary origin) cell line (B-CPAP) in Fig. 4a (extended in Supplementary Movies 1–2). At two days after transfection of *PIAS2b*-dsRNAi, ATC cells were unable to complete mitosis and died of mitotic catastrophe (Quantification in Fig. 4b). In comparison, ns-dsRNAi control cells achieved mitotic completion in >80–95% of cells. B-CPAP, no anaplastic and no affected by *PIAS2b*-dsRNAi, was able to reach mitotic completion in 100% of cells (Fig. 4b). Individual tracing revealed that >75% anaplastic cells treated with *PIAS2b*-dsRNAi arrested at prophase and then died (Blue bars Fig. 4b).

### PIAS2b accumulates at the mitotic spindle and interacts with mitotic proteins

Analyzing mitotic 8305C cells by confocal double immunofluorescence for tubulin alpha (aTub) and mPIAS2 revealed that PIAS2b colocalized with aTub at the mitotic spindle in every mitotic phase, from prophase to telophase (Fig. 4c). PIAS2b was no longer associated to aTub in cytokinesis.

mPIAS2 staining at mitosis could indicate PIAS2a or PIAS2b. We transfected 8305C with a non-toxic amount of EGFP-PIAS2b (Empty pcDNA3 plasmid: pEGFP-C1-PIAS2b proportion 19:1) (Fig. 4d). Co-staining with phospho-Ser10 histone H3 (p-HH3) revealed mitotic cells. Direct detection of EGFP-PIAS2b by time-lapse or super-resolution confocal microscopy presented a weak signal (Fig. 4d and Supplementary Fig. 4c). We then tried using a GFP antibody after fixation, but although sensitivity increased, also resulted in a

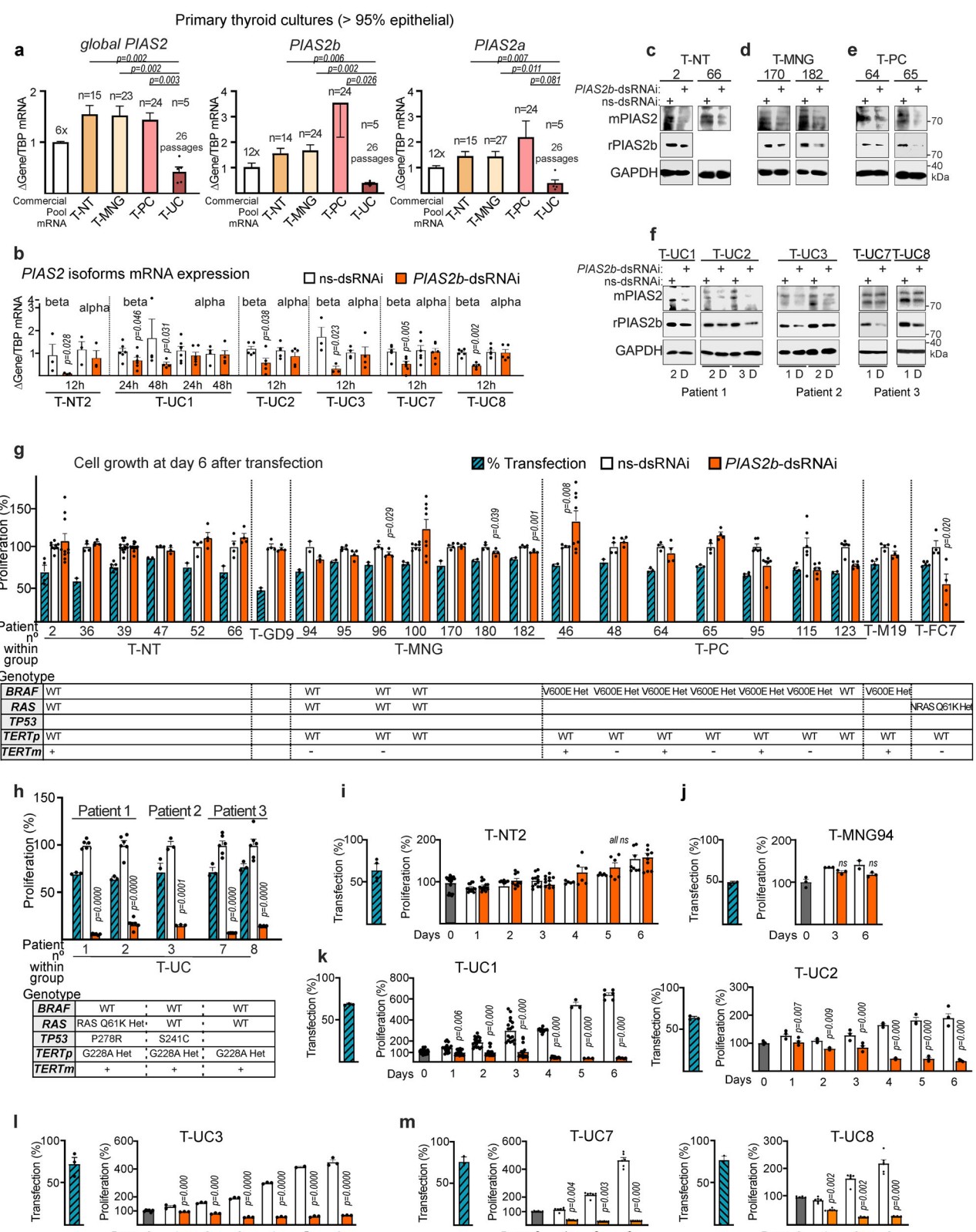

loss of specificity with border effects or diffuse signal. Regardless, EGFP-hPIAS2b was indeed located at the metaphase spindle, similar to mPIAS2 staining, confirming that it was indeed the PIAS2b isoform bound to the spindle. Finally, we purified EGFP-PIAS2b protein from transfected cells using the GFP-Trap, eluted it, and directly transfected the protein into cells with Xfect (Fig. 4d and Supplementary Fig. 4c), although we had not guarantee of full EGFP-PIAS2b renaturation.

Results improved signal at the spindle, but again reduced specificity with green spots floating at the cell periphery or at the plasma membrane (Fig. 4d and Supplementary Fig. 4c). Although we cannot use this approach for colocalization studies, we found that the GFP fluorescence in all mitotic cells transfected with EGFP-PIAS2b was located at the spindle as endogenous PIAS2b detected with mPIAS2 or rPIAS2b antibodies.

**Fig. 3 | *PIAS2b*-dsRNAi blocks growth of patient-derived primary cultures of ATC but not those derived from differentiated thyroid carcinomas, thyroid follicular nodular disease/multinodular goiter, or normal thyroid. a** Expression of *PIAS2* mRNA detected by RT-qPCR for detection of all isoforms (*global PIAS2*), *PIAS2b* or *PIAS2a* isoforms in primary cultures established from thyroid tissues. Our protocol h7H obtained cultures with >95% follicular epithelium. **b–f** Transfection of *PIAS2b*-dsRNAi (orange bars) significantly and specifically reduces *PIAS2b* expression. For **b**, mRNA expression of *beta* and *alpha PIAS2* isoforms in primary cultures from normal thyroid (T-NT2) or anaplastic thyroid carcinoma (T-UC) transfected with non-target sequence (*ns*-dsRNAi) or *PIAS2b*-dsRNAi. Western blot of treated extracts confirmed downregulation of PIAS2b in primary cultures from normal thyroid (T-NT) (**c**), multinodular goiter (T-MNG) (**d**), differentiated papillary thyroid carcinoma (T-PC) (**e**), and ATC (T-UC, five cultures from three patients) (**f**). **g, h** Cell growth of the primary cultures in the presence of *PIAS2b-dsRNAi* (orange bars) expressed as percentage of the ns-dsRNAi control (100%, white bar for each culture). Blue striped bars, indicate the percentage of transfection. Genotype and *TERT* expression for the most common events in thyroid cancer are shown below. Some of those cultures were established from single patient surgery paired

surpluses, obtained from different pathology as assessed by the clinical pathologist (Supplementary Data 2): T-NT39 and T-M19; T-NT52 and T-PC48; T-MNG100 and T-PC46; T-MNG197 and T-PC64, T-PC65; T-MNG94, and T-UC1, T-UC2; T-UC7 and T-UC8. Numbers refers to the code number for each within the pathological group. **i–m** Time-course experiments in some primary cultures showed no effect on cell growth of *PIAS2b*-dsRNAi in normal/ benign cultures (**i, j**) but confirmed progressive cell loss in ATC cultures from the three patients (**k–m**). **a** n included in the section, two-sided Kruskal–Wallis; (**b**) $n = 3–4$ (T-NT2), $n = 5–6$ (T-UC1, T-UC2, T-UC7, T-UC8), $n = 6–3$ (T-UC3) independent experiments, one-sided Mann–Whitney; **g, h** n of independent experiments: $n = 3$ (T-NT47, T-MNG94, T-UC3), $n = 4$ (all the other cultures), $n = 6$ (T-UC1, T-UC2, T-UC7, T-UC8), $n = 8$ (T-MNG100), $n = 10$ (T-NT2); **i** $n = 4$, **j–l** $n = 3$, **k, m** $n = 6$. **b, g, h** Two-sided Mann–Whitney; **i** two-sided one-way ANOVA with repeated measures and Dunnett's multiple correction; **j** two-sided one-way ANOVA with repeated measures and Bonferroni multiple correction; **k–m** two-sided one-way ANOVA with repeated measures, and Tukey's multiple comparison test. Bar indicate means ± SEM; when significant the exact *p* value is indicated in the figure. ns non-significant. Source data are provided as a Source Data file.

---

To explore the role of PIAS2b in mitosis, we performed standardization of mitotic synchronization in two ATC cell lines (8305C and CAL-62) using double-thymidine block and release (DT/R). Flow cytometry was performed in a time-course after release (DT/R–0 h) to follow cell cycle and mitosis. At 6 h after release (DT/R-6 h), >56% of cells were accumulated at G2/M (Supplementary Fig. 4d). Western blot of extracts at 0 h and at 6 h (mitosis) showed that PIAS2b accumulated at mitosis (Fig. 4e), similar to well-known mitotic proteins, such as aurora kinase A (AURKA) and gamma tubulin (gTub).

To identify PIAS2b-associated proteins at mitosis in 8305C cell extracts at DT/R-6 h, we performed immunoprecipitation using mPIAS2 and non-immune mouse IgG2a isotype (Proteomic Assay 1, Fig. 4f) and analyzed the results by LC-MS/MS coupled to 6600 Triple TOF proteomics. The experiment was also quantified by SWATH and the M1 normalized Spectrum Count (Scaffold) (Supplementary Data 3). Identified proteins that had significance in triplicate experimental repetition in both quantifications were selected and analyzed for functional pathways (Supplementary Data 3). Cell cycle, Mitotic (HSA-69278) was the most non-generic pathway identified at the REACTOME, with 122 identified proteins (Fig. 4g). The mitotic proteins associated to PIAS2 were dominated by those from two pathways: microtubule cytoskeleton and proteasome (Fig. 4g, green and red, respectively). Some of these proteins were specific for prophase (Fig. 4g, violet). Among them RANBP2, TUBGPC2, NEK7, NUMA1, AAAS, PPP2CA, CCNB1, CDK1, PLK1 and AURKA has been demonstrated to have an important role in spindle assembly at early Prophase[42–47].

Of note, PIAS2 is an E3 SUMO ligase, and our in silico study looking for SUMO consensus sites indicated that 110 of the 122 proteins presented a high-score consensus SUMOylation site (Supplementary Data 3). Similarly, crossing our list with the identified SUMOylated proteins in the high-throughput seminal study of Hendriks et al.[48] underscored many common SUMOylated proteins (Fig. 4g, underlined; Supplementary Data 3). To validate direct binding to PIAS2b, we performed proteomic analysis of pull-downs eluted from GFP-Trap, of extracts obtained after transient transfection of a non-toxic amount of EGFP-hPIAS2b or EGFP (pcDNA3: pEGFP-plasmid in a proportion of 19:1) (Proteomic Assay 2, Fig. 4h). This experiment was also quantified with Spectrum Count (Supplementary Data 3). Even with a low number of proteins identified, this experiment was useful to validate direct PIAS2b targets found in common to Proteomic Assay 1 (yellow line circles in Fig. 4g).

We validated some PIAS2b-associated proteins from both Proteomic Assay 1 and 2, confirming gTub, aTub, the catalytic subunit of the serine/threonine-protein phosphatase 2A alpha isoform (PPP2CA), and AURKA, as proteins associated to PIAS2b in mitosis (Fig. 5a–d, and Supplementary Fig. 5a).

To further validate association at mitosis, we performed confocal microscopy staining for PIAS2b (as mPIAS2) plus AURKA in 8305C cells synchronized at mitosis (DT/R-6 h). All mitotic patterns in which AURKA was positive (e.g., with two or more dots) were registered, and assigned to a mitotic phase according to their DAPI staining (Fig. 5e). As prophase initiated, there were multiple AURKA dots, which only partially colocalized with PIAS2b. AURKA dots progressively converged into two spots (spindle centrosomes); similarly, PIAS2b also progressively merged into these two spots. Thus, in early prometaphase, PIAS2b and AURKA coincided into the centrosomes with approximately equal intensities, at both sides of the chromosomes (Fig. 5e). As prometaphase progressed, PIAS2b accumulated and exceed AURKA spots, with PIAS2b filaments following the spindle between the two AURKA spots (Fig. 5e). At late prometaphase, a fully filamentous PIAS2b spindle surrounded AURKA, orientated between the two AURKA poles (Fig. 5e). As these are anaplastic (e.g., aneuploid) cells, a triple-pole organization was observed on occasion. The apparent relationship between PIAS2b and AURKA continued in metaphase, where PIAS2b filaments reached the organized condensed chromosomes at 90°. Upon reaching anaphase, PIAS2b filaments were arranged perpendicular to the chromatin between the two daughter chromosome alignments, with a weak amount of AURKA associated. The main AURKA intensity at anaphase remained associated with the centrosome poles of the mitotic spindle (which were also intense for PIAS2b). In telophase and initial cytokinesis, there was no AURKA staining, but PIAS2b bundles remained between the two daughter chromosomes until they were completely reorganized into independent daughter nuclei (Fig. 5e).

This mitotic spindle staining and co-localization for mPIAS2 and AURKA was observed not only in ATC cell lines (CAL-62, BHT-101, MB-1) and ATC primary cultures (including T-UC1, T-UC2, and T-UC3) but also in the normal thyroid cultures, such as T-NT2 (Supplementary Fig. 5b).

Quantitative colocalization of PIAS2b, SUMO1, and gTubulin in interphase and mitotic cells was carried out. We performed immunofluorescence for PIAS2b (mPIAS2 and rPIAS2b), SUMO1 (mSUMO1 and rSUMO1), mSUMO2, and gTubulin in interphase and mitotic cells after DT/R-6 h. The cells were then visualized using super-resolution confocal microscopy (Fig. 5f–i; negative controls are shown in Supplementary Fig. 5c, d). Pearson's Colocalization Coefficient (PCC) was quantified to assess the degree of colocalization between the different proteins and results plotted as violin plots (Fig. 5j).

SUMO2 was detected attached to chromatin (DAPI) and was not colocalized with any of the other proteins at mitosis (Supplementary Fig. 5e). In interphase, PIAS2b and SUMO1 colocalized at the nucleus with a PCC of ~0.9. This colocalization decreased at prophase, but then

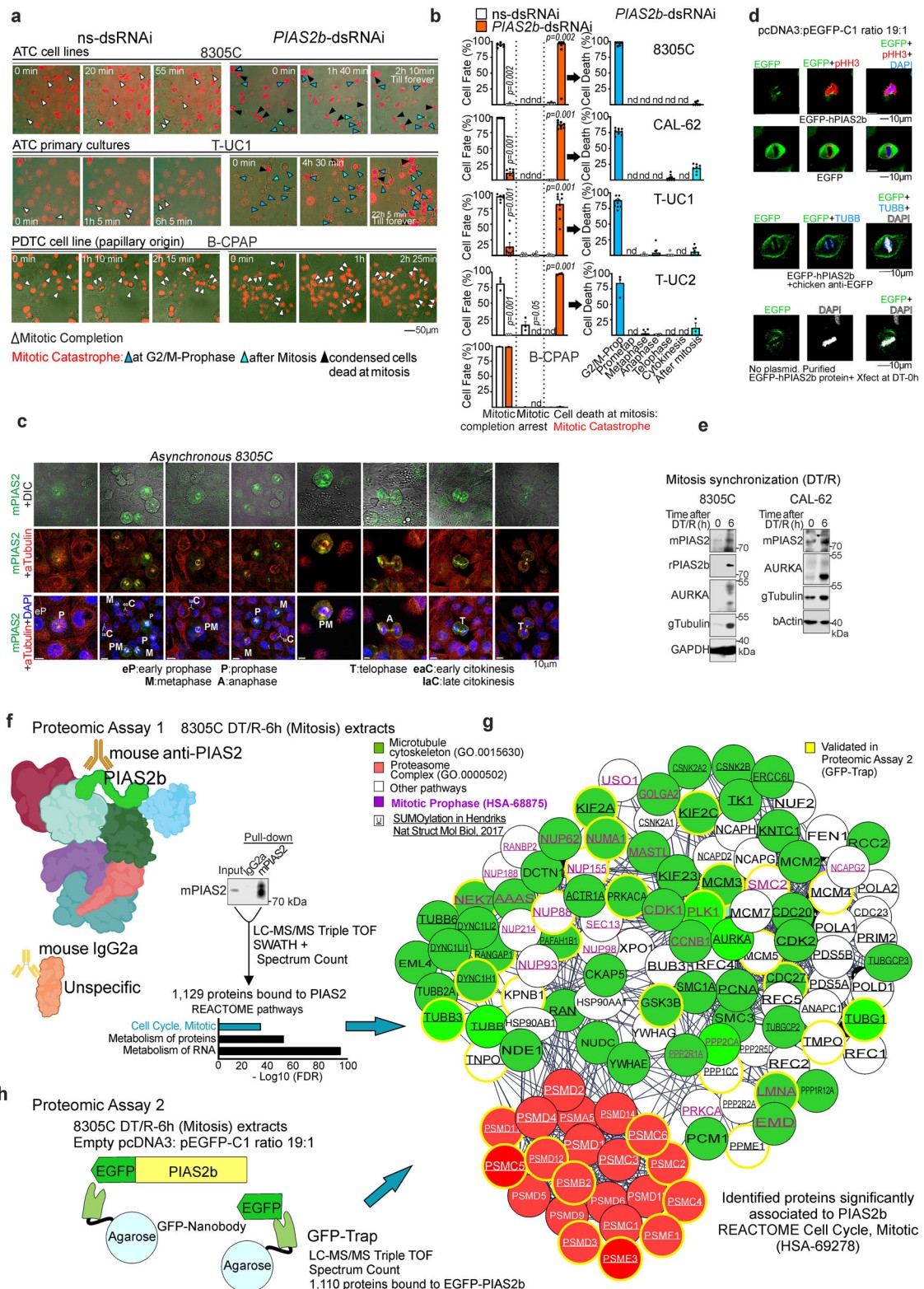

recovered at prometaphase and metaphase (Fig. 5f, j). At the end of mitosis, colocalization again decreased.

mPIAS2 and rPIAS2b followed an identical pattern and had a constant PCC of 0.8 at any phase (Fig. 5i, j, right). As mitosis began, PIAS2b colocalization with chromatin progressively decreased, and its colocalization with gTubulin or TUBB-647 progressively increased, revealing the spindle (Fig. 5f–h, j). The maximal PCC peaked

at metaphase, and then decreased again (Fig. 5j). Binding of PIAS2b to chromatin at the nucleus was recovered after telophase, precisely when colocalization to the spindle was lost. 3D surface renderings of both PIAS2 antibodies show PIAS2b progressively located at the centrosomal side of the spindle where maximal colocalization was achieved at Metaphase as shown by blended pseudocolors (Supplementary Fig. 5f).

**Fig. 4 | *PIAS2b*-dsRNAi induces mitotic catastrophe at the onset of mitosis in anaplastic thyroid cancer cells. PIAS2b localizes at the mitotic spindle and binds microtubule and proteasomal proteins. a** Representative photograms of 48-h Sir-DNA stained time-lapse with ns- or *PIAS2b*-dsRNAi in one ATC cell line (8305C), one ATC primary culture (T-UC1), and the PDTC (papillary origin) cell line B-CPAP; (fully in Supplementary Fig. 2). Individual Cells traced for mitotic completion (white arrowheads) or mitotic catastrophe (blue and black arrowheads). **b** The 4 anaplastic cultures show massive mitotic catastrophe at mitotic onset (G2/M-prophase) compared to mitotic completion in B-CPAP. **c** Mitotic colocalization of tubulin-alpha (aTub, red) and PIAS2b (mPIAS2, green), in asynchronous 8305C. DAPI in blue. DIC, differential interference contrast. **d** EGFP-hPIAS2b transfected at low amounts (Supplementary Figs. 1x, 4c) followed by enhancing signal strategies, shows localization at the spindle compared to broad cytoplasmic staining in EGFP-C1. Phospho-Histone H3 (pHH3) indicates mitotic cells. EGFP immunostaining, or direct Xfect of EGFP-PIAS2b protein also localize at the spindle, but with background plasma membrane signal. **e** 8305C and CAL-62 cells synchronized at mitosis with double-thymidine block followed by release (DT/R). Mitotic proteins AURKA

or tubulin gamma accumulate at 6 h after release. PIAS2b, detected with mPIAS2 or rPIAS2b, also accumulates at DT/R-6 h. **f** Immunoprecipitation of PIAS2b with mPIAS2 antibody or control isotype mouse IgG2a in DT/R-6 h synchronized 8305C extracts (called Proteomic Assay 1). Replicate pull downs were analyzed by LC-MS/MS and quantified (SWATH and Spectrum Count). Identified proteins specific for PIAS2 immuno-precipitation were enriched in the mitotic REACTOME. **g** The three GO pathways were microtubule/spindle cytoskeleton (green), proteasome complex (red) and prophase (violet). Underlined known SUMOylated proteins. Validated proteins in this and following figures in brilliant color. **h** GFP-Trap pull-downs of EGFP-PIAS2b or EGFP were analyzed by LC-MS/MS (Proteomic Assay 2). A cross-data analysis validates many mitotic proteins common to both Proteomic Assays 1 (**f**) +2 (**h**), yellow circles in **g**. For **b** $n = 6$ (8305C, T-UC1)-8 (CAL-62)-3 (T-UC3) and 15 (BCPAP) independent experiments; two-sided Mann–Whitney. Bar indicate means ± SEM; nd, not detected; when significant the exact $p$ value is indicated in the figure. For **c** and **d** $n = 5$ independent experiments. For **e** and **f** $n = 3$, and for **g** and **h** Proteomic Assay 1 $n = 5$ and Proteomic Assay 2, $n = 3$ independent experiments; quantifications in Supplementary Data 3. Source data provided.

## Direct targets of PIAS2b SUMOylation at mitosis: proteasome (PSMC5) and tubulins (TUBB3)

To understand which mitotic molecular events where affected by lack of PIAS2b SUMOylation, we silenced PIAS2b in 8305C ATC cells and then synchronized and progressively followed the cell phases by DT/R (Fig. 6a). Mitotic synchronization significantly potentiated the deleterious action on cell growth of *PIAS2b* dsRNAi at DT/R-day 6 (Fig. 6b and Supplementary Fig. 6a).

Protein extracts were compared by Western blot. PIAS2b silencing-induced accumulation of phosphorylated AURKA and progressive loss of gTub and PLK1; the phosphatase CDC25C was also accumulated at DT/R-6 h (Fig. 6c and Supplementary Fig. 6b). CDK1 was fully active in PIAS2b-silenced cells, as seen both by loss of inhibitory pTyr15 phosphorylation as well as by a single lower molecular weight band in total CDK1 detection, as compared with the doublet detected in control (*ns*-dsRNAi) cells (Fig. 6d). In PIAS2b-silenced cells, cyclin B progressively accumulated up to DT/R-4 h and began to decay at DT/R-6 h; however, and in contrast to CDK1 and AURKA activation, p-Ser10-Histone H3 (p-HH3) was strongly reduced already at DT/R-4 h (Fig. 6d). PPP2CA (a key phosphatase for the G2/M checkpoint) was detected with the antibody clone F8, which was recently demonstrated to not recognize the inhibitory p-Tyr307 but rather other post-translational modifications[49]; the Western blot showed a heavily modified ~95 kDa PPP2CA band accumulated in PIAS2b-silenced cells that was not recognized by a total PPP2CA antibody (clone 0T118) (Fig. 6d). No differences were observed between total or phosphorylated phosphatase 1A (pThr320-PP1A) in silenced or control cells (Supplementary Fig. 6b). Taken together this can only indicate that we are in mitotic arrest.

PIAS2b functions as an E3 SUMO-protein ligase. We choose the DT/R-5 h timepoint to perform LC-MS/MS coupled to triple TOF proteomic analysis, Proteomic Assay 3, comparing ns-dsRNAi (control) and *PIAS2b*-dsRNAi mitotic extracts. The extracts including SUMO-protease inhibitors were passed through a high-affinity commercial SUMO Qapture column that retained SUMOylated proteins, with not SUMOylated proteins washed out in the flow-through. Both fractions were analyzed (Fig. 6a, e–g and Supplementary Data 4). In total, 1146 unique proteins were quantitatively identified in control extracts, and 879, in PIAS2b-silenced cell extracts. Of these, 502 identified shared proteins were identified as SUMOylated (retained), not SUMOylated (flow-through), or partially SUMOylated (both fractions) (Fig. 6e, white circles).

A total of 462 proteins (SUMOylated or not) were found exclusively in control extracts (Fig. 6e, upper left). A total of 195 proteins (SUMOylated or not) were exclusively identified in PIAS2b-silenced cells (Fig. 6e, upper right).

Normalized spectrum count of the SUMO-retained proteins identified in both treatments revealed 132 proteins that lost

SUMOylation in PIAS2b-silenced cells (Fig. 6f, pink bar), while 52 proteins increased SUMOylation in PIAS2b-silenced cells (Fig. 6f, yellow bar). The 132 group was of interest for identifying putative direct PIAS2b substrates, which likewise were enriched in the GO pathways of proteasome and G2/M transition (Fig. 6f, right).

We crossed these 132 proteins with the list of PIAS2b-binding mitotic proteins resulting from Proteomic Assay 1 (IP-PIAS2) and Proteomic Assay 2 (binding to EGFP-PIAS2b) to find common proteins in the three studies (Fig. 6g and Supplementary Data 4). 20 proteins were common to all three studies, identified as bound to PIAS2b in immunoprecipitation and GFP-Trap, and SUMOylated in ns-dsRNAi cells whereas, although also identified, their SUMOylation was quantitatively lost in PIAS2b-silenced cells (Fig. 6g and Supplementary Data 4). At least 8 of them were registered in GO pathways with function related to the microtubules/spindle (ABCE1, DYNC1H1, MACF1, PLEC, PLS3, SPTAN1, TNPO1 and TUBB3), while 4 were related to the proteasome (PSMB2, PSMC5 also called PRS6A, PSMC6, PSMD12), (green and red labeled respectively in Fig. 6g). Other 30 proteins were only in common with Proteomic Assay 1 and 3, and should also be of notice since Proteomic Assay 2 was less sensitive; again many were related to microtubule/spindle (green) and proteasome (red) (Fig. 6g). One of them was PPP2CA, a protein with a striking high-molecular weight band found in PIAS2b-silenced cells (Fig. 6b).

Of note, TUBB3, PSMC3, and PSMC5 present consensus in silico SUMO binding sites and were previously found to be SUMOylated[48] (Supplementary Data 4). Five of all these putative PIAS2b SUMOylated proteins (PSMD1, DYNC1H1, TRIM28, EIF4G1, TNPO1) have recently being confirmed as direct PIAS2b targets in non-mitotic U2OS cells using advanced SUMO-activated target traps (SATTS)[50] (Supplementary Data 4).

We chose PSMC5 and TUBB3 for further validation. Western blot of the proteomic extracts showed that PSMC5 was exclusively detected in SUMO-captured extracts from ns-dsRNAi cells but not from *PIAS2b*-dsRNAi extracts (Fig. 6h). TUBB3 antibody did not work with these proteomic extracts.

We started by validating the PIAS2b-target association. FLAG-hPIAS2b or FLAG-PSMC5 were transfected followed by anti-FLAG pull down and Western blot. PSMC5 and TUBB3 were associated to transfected FLAG-hPIAS2 (Fig. 6i), while PIAS2b was associated to FLAG-PSMC5 (Fig. 6j). We next explored the association of these three proteins during mitosis in extracts from asynchronous, DT/R-0 h, and DT/R-6 h cells. PIAS2b, PSMC5, and TUBB3 were enriched at mitosis (Fig. 6k). Co-immunoprecipitation of endogenous PIAS2b and PSMC5 was only detected at DT/R-6 h (Fig. 6k), while that of PIAS2b and TUBB3, already detected at DT/R-0 h, increased at DT/R-6 h (Fig. 6k).

In *PIAS2b*-dsRNAi cells, both proteins were enhanced in asynchronous phases and at mitosis (DT/R-6 h), whereby specifically

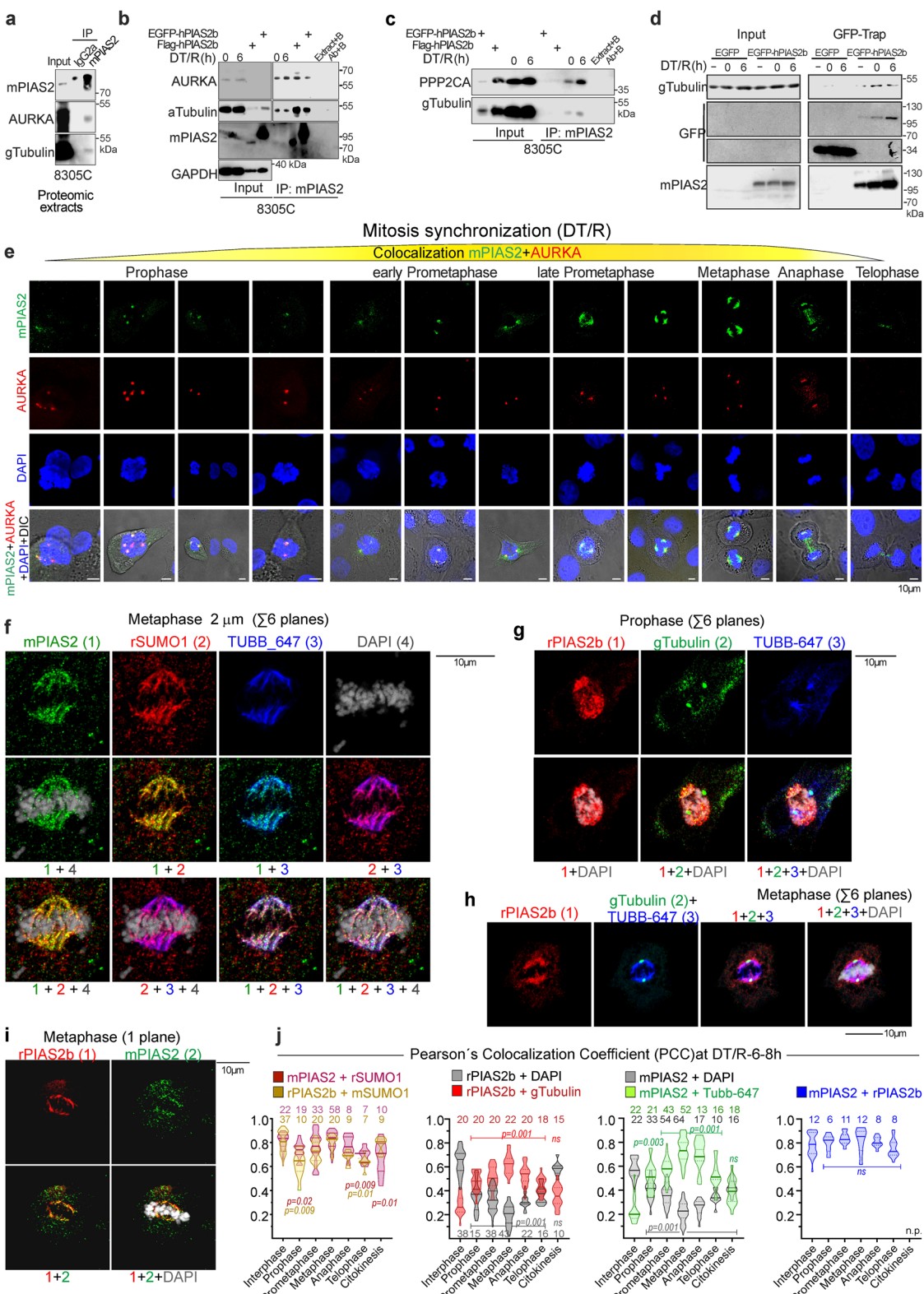

PMSC5 (expected weight ~50 kDa) presented stronger bands at higher molecular weights (70/95 kDa), indicative of post-translational modification (PTM) (Fig. 6l).

Next, we validated PIAS2b-dependent covalent SUMOylation of the two substrates using 8305C ATC cells in mitosis. Since PSMC5 has a molecular weight (~50 kDa) concordant with IgG heavy chain (IgG H), our first strategy was transfection of EGFP-PSMC5 or EGFP

and pull-downs after GFP-Trap, followed by SUMO western blot using non-denatured extracts with SUMO-protease inhibitors (Fig. 7a).

EGFP-PSMC5 was strongly SUMOylated with SUMO1 at mitosis (DT/R-6 h), with shifts in molecular weights compatible with SUMOylation (+17 kDa), and further PTM(s) with higher weights (Fig. 7a, left), and its SUMOylation was blocked when treated with *PIAS2b*-dsRNAi (DT/R-5 h) (Fig. 7a, right). SUMO2 did not give specific signal at the

**Fig. 5 | PIAS2b binds AURKA, gTubulin, SUMO1 and microtubule proteins like TUBB at the mitotic spindle. a–d** Validation of some of the PIAS2b-binding proteins obtained in proteomic assays 1 + 2 (AURKA, aTubulin, gTubulin, PPP2CA) in DT/R-6 h synchronized mitotic 8305C. AURKA and aTubulin was bound by transfected exogenous human PIAS2b. gTubulin was also validated by GFP-Trap pulldowns and western. **e** Confocal colocalization of AURKA (red) and PIAS2b (green). All AURKA+ cells were recorded and arranged in the successive mitosis phases according to DAPI staining. At the initial steps of prophase, multiple dots of AURKA and some PIAS2b did not colocalize. Progressively, they converge at the two characteristic AURKA dots (centrosomes). In early prometaphase, PIAS2b surpasses AURKA and stains the mitotic spindle. This continues throughout metaphase. In anaphase and telophase, PIAS2b remains between the daughter chromosomes, even in the absence of AURKA. **f–j** Confocal super-resolution colocalization using mPIAS2, rPIAS2, mSUMO1, rSUMO1, gTubulin, Alexa 647-labeled anti-beta Tubulin (TUBB-647) and DAPI (shown Metaphase and Prophase; controls and additional cells in Supplementary Figs. 5c, d, f). Quantification of Pearson's Colocalization Coefficient (PCC) for every two markers summarized as violin plots in **j**. Analyzed cells in each phase are indicated. Statistics are compared to interphase. For **a–d** n = 3 independent experiments. For **e** n = 5 independent experiments. For **j** violin plots with cells per condition indicated obtained from n = 5 independent experiments, two-sided one-way ANOVA with Dunnett multiple comparison test. Exact p value is indicated in the figure; ns non-significant; n.p. not performed. Source data are provided as a Source Data file.

westerns (Supplementary Fig. 7a). Of notice, exogenous EGFP-PSMC5 was bound to endogenous PSMC5, also specifically SUMOylated by PIAS2b in mitosis (Fig. 7a). PIAS2b was associated to EGFP-PSMC5, and also SUMOylated at mitosis or absent in *PIAS2b*-dsRNAi (Fig. 7a).

TUBB3 native molecular weight (Uniprot doublet 50,433 and 42,433 Da) is also concordant with IgG, and many PTMs for Tubulins like phosphorylation, acetylation, polyglutamylation, polyglycylation, and detyronisation are described[51,52]. Although PTMs add mass to the tubulin molecule, the type of PTM, charge and number of modifications will influence the extent of the shift in apparent molecular weight. Thus, our second strategy to confirm covalent SUMO1ylation was transfection with 10xHis-SUMO1, synchronization by DT/R, minus or with dsRNAi treatment, extraction with denaturing buffer, and pulldowns after Ni-NTA beads incubation and elution. When empty vector instead of 10xHis-SUMO1 was transfected, no band was detected in Ni-NTA-beads with any of the antibodies used, confirming pull-downs specificity (Supplementary Fig. 7b).

In denaturing extracts, inputs showed PSMC5 as a main band at 45 kDa, and less intense bands shifting upwards at mitosis (DT/R-6 h); TUBB3 showed a new higher band of ~70 kDa, together with the 50 and 42 kDa bands, with this smaller band being less intense at mitosis (Fig. 7b). The Ni-NTA pull downs followed by western blot revealed specific SUMO1ylation of the upper bands of SUMO1-bound PSMC5 bands at DT/R-6 h, and its reduction by *PIAS2b*-dsRNAi (Fig. 7b). The double transfection (10xHis-SUMO1 and dsRNAi) with different lipids gave us one unspecific band (u), that required longer gel running for separation. High-molecular-weight poly-SUMO PSMC5 bands were also increased at DT/R-6 h and reduced with *PIAS2b*-dsRNAi (Fig. 7b). Critically, ~p70-TUBB3 band (55kDa-TUBB3 band + 1 SUMO1), the 55kDa-TUBB3 band (42kDa-TUBB3 + 1 SUMO1), and the higher molecular weight TUBB3 band (poly-SUMOylated TUBB3), appear in the Ni-NTA pull-downs at G2/M (DT/R-0h) and markedly increase after release, supporting the conclusion of SUMO1ylation and poly-SUMOylation during mitosis (DT/R-6h) (Fig. 7b, left); and the three bands are strongly reduced with PIAS2b-depletion at DT/R-5h (Fig. 7b, right). The p42kDa-TUBB3 does not appear in the Ni-NTA pulldowns (Fig. 7b), indicating that is non-sumoylated TUBB3.

PIAS2b was also specifically SUMO1ylated and shifted at DT/R-6 h, and this was lost in *PIAS2b*-dsRNAi (Fig. 7b). The pattern of SUMO1ylation, both with total mSUMO1 and His-Tag antibodies, increased at mitosis, and decreased with *PIAS2b*-dsRNAi. Intriguingly, mono-10xHis-SUMO1 accumulated in these extracts, as in not bound to proteins (Fig. 7b).

We analyzed the association of PIAS2b (with mPIAS2 and rPIAS2b), PSMC5 (with EGFP-PSMC5 or rPSMC5 antibody), SUMO1 (with mSUMO1 or labeled SUMO1-647), and TUBB-647 in mitosis by triple colocalization using super-resolution microscopy followed by modeling through 3D surface rendering with Imaris (Supplementary Supplementary Fig. 7c, Supplementary Movie 3). The number of colocalized voxels was quantified as violin plots (Fig. 7c).

At interphase, PSMC5 was mainly cytoplasmic, while PIAS2b and SUMO1 were nuclear. TUBB was cytoplasmic with low intensity. Progressively from prophase to metaphase, while the spindle was polymerizing, PSMC5, SUMO1, and PIAS2b became significantly associated at the disintegrating nucleus, and with TUBB which was increasingly polymerizing (Fig. 7c, d). Maximal colocalization was observed at metaphase. In anaphase and telophase, PSMC5 binding to PIAS2b was progressively reduced.

## Untimely proteasome activation in *PIAS2b*-dsRNAi cells induces centrosome alterations and mitotic arrest at kinetochores, leading to mitotic catastrophe

The PIAS2b–TUBB and PIAS2b–PSMC5 associations, as well as PSMC5 modifications in the absence of PIAS2b and the number of proteins lost following *PIAS2b*-dsRNAi in Proteomic Assay 3, suggested that the PIAS2b role in mitosis was associated to the microtubule spindle retaining proteasome components to prevent untimely proteasome activation at early mitosis. Nuclear PSMC5 staining increased in *PIAS2b*-dsRNAi treated 8305C cells at DT/R-5 h as compared to ns-dsRNAi (Fig. 7e). We therefore quantified the proteasome activity in the 8305C cells progressively after DT release (Fig. 7f). At DT/R-0 h, proteasome (chymotrypsin-like) activity from *ns*-dsRNAi or *PIAS2b*-dsRNAi did not differ. However, it increased progressively in *PIAS2b*-dsRNAi treated cells at DT/R-2 h and DT/R-4 h (Fig. 7f). In mitosis (DT/R-6 h), proteasome activity fell sharply in *PIAS2b*-dsRNAi cells, coinciding with mitotic catastrophe. Specificity of proteasome assay activity was demonstrated by blocking with MG132 (Fig. 7f).

To observe how these molecular events were translated into the spindle and centrosome where PIAS2b is localized during mitosis, and looking for markers of mitotic catastrophe, we colocalized spindle and centrosome proteins through immunofluorescence at DT/R-5 h and performed conventional confocal microscopy. p-HH3–positive nuclei were significantly reduced in *PIAS2b*-dsRNAi–treated cells (Fig. 7g), in agreement with the previous western blot results (Fig. 6d).

CREST (the kinetochore marker) staining revealed colocalization of gTubulin and CREST in *PIAS2b*-dsRNAi but not in control ns-dsRNAi cells, where gTubulin spots were side-to-side with CREST (Fig. 7h). gTubulin staining in *PIAS2b*-dsRNAi cells presented a peculiar nuclear accumulation of 23 spots that coincided with the number of possible kinetochores, while in *ns*-dsRNAi an average of 5 spots were detected (Fig. 7h and Supplementary Fig. 7c). The protein in charge of controlling correct microtubule attachment to kinetochores is BUB3[53–55]. Co-staining for BUB3 and gTubulin showed co-localization at the spots in *PIAS2b*-dsRNAi treated cells (Fig. 7h and Supplementary Fig. 7d), but also revealed BUB3-negative metaphases, or improper BUB3-gTububulin colocalization in only one spindle pole. Some cells presented two nuclei (Supplementary Fig. 7d). Similar anomalies were found co-staining for BUB3 and CREST, whereby nuclei with colocalizing spots, together with nuclei BUB3+ devoid of CREST, and cells with double nuclei were observed (Supplementary Fig. 7d). This was

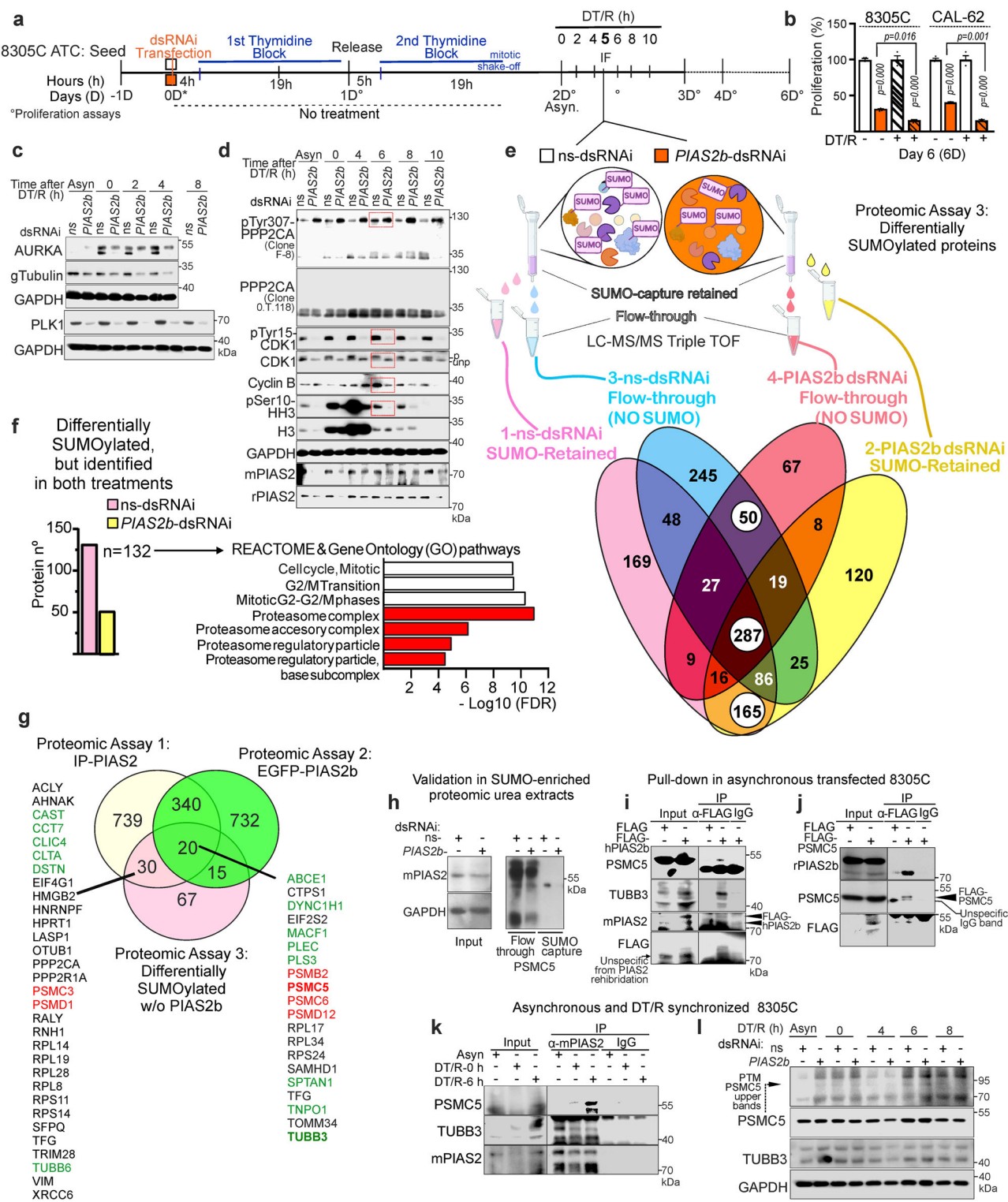

coincident with video time-lapse tracing experiments in cell lines and primary cultures (Supplementary Movies 1 and 2), which showed that cells treated with *PIAS2b*-dsRNAi tried to complete mitosis but returned brusquely to an interphase state or died.

The centrosome marker pericentrin (PCNT) is duplicated just before a normal prophase, showing two well-defined spots[56,57]. It is known that alteration of specific proteins implicated in the assembly of the mitotic spindle induces qualitative and quantitative alterations in centrosomes[58]. PLK1 (a marker of mitotic centrosomes) colocalized with PCNT in duplicated centrosomes of *ns*-dsRNAi treated cells, as expected.

However, in *PIAS2b*-dsRNAi centrosomes did not present colocalization of PLK1 and PCNT (Fig. 7h and Supplementary Fig. 7d). *PIAS2b*-dsRNAi treated cells displayed significantly more unequal and fuzzy centrosomes, and >3 centrosomes/cell (Fig. 7h and Supplementary Fig. 7e).

aTub and gTub co-staining revealed both centrosomal and microtubule cytoskeleton perturbations with *PIAS2b*-dsRNAi, such as duplicated centrosomes wandering away from the nucleus, multiple or unequal centrosomes, reorganization of aTub at the cell cortex, altered spindle poles with misaligned microtubules, or two nuclei (Supplementary Fig. 7f).

**Fig. 6 | Mitotic proteins specifically SUMOylated by PIAS2b include proteasomal and spindle proteins. a** SUMO-capture strategy to find PIAS2b SUMOylation targets at mitosis (DT/R-6 h) in ns- or *PIAS2b*-dsRNAi treated 8305C cells. **b** *PIAS2b*-dsRNAi has a greater anti-cancer effect in mitotic synchronized than in asynchronous anaplastic cells. **c, d** Time-course of mitotic proteins reveal key alterations induced by *PIAS2b*-dsRNAi: fully phosphorylated AURKA (**c**); tubulin gamma (**c**) and PLK1 (**c**) markedly decrease; fully active CDK1, negative for inhibitory pTyr15-CDK1 while a single band in whole protein detection (**d**); **d** cyclin B and pSer10-HH3 reduction at mitosis; and PPP2CA shifts to 130 kDa when detected with F-8 antibody, respect to normal protein (detected with 0.T.118). **e** Proteins identified by LC-MS/MS and quantified by Spectrum count: from ns-dsRNAi cells, retained SUMOylated (pink), and flow-through not SUMOylated (blue); from *PIAS2b*-dsRNAi cells, retained SUMOylated (darker pink), and flow-through not SUMOylated (yellow). White circles: proteins unaltered in both treatments. **f** 132 proteins (pink bar) identified in both treatments, but with significant SUMO-retention in ns-dsRNAi compared to *PIAS2b*-dsRNAi; 52 proteins (yellow bar) presented less SUMO-retention in ns-dsRNAi. In proteins that lost SUMOylation in absence of PIAS2b, GO

pathways highlights G2/M transition, mitosis and proteasome (red). **g** Venn diagram cross-analysis of Proteomic Assays 1 + 2 + 3. Shown the 20 proteins that are significantly bound to PIAS2b (Proteomic Assays 1 + 2) and loss SUMOylation in the absence of PIAS2b; 30 other PIAS2b-bound proteins losing SUMOylation are also remarkable although were not detected in the less sensitive Proteomic Assay 2. Highlighted in red, proteasome and green, spindle proteins. **h** Validation of PSMC5 as a PIAS2b-dependent SUMOylated protein in the SUMO-affinity column urea extracts as compared to flow-through. Direct association of FLAG-hPIAS2b to PSMC5 and TUBB3 (**i**), and of FLAG-PSMC5 to PIAS2b (**j**). **k** Specific association of PIAS2b to PSMC5 at mitosis (DT/R-6 h), compared to asynchronous (Asyn) or DT/R-0 h. A TUBB3–PIAS2b association increased at mitosis. **l** In *PIAS2b*-dsRNAi, TUBB3 and PSMC5 accumulated at mitosis, especially post-translationally modified (PTM)-PSMC5 (p70-p95). **b** $n = 3$ independent experiments, two-sided two-way ANOVA with repeated measures. Bar indicate means ± SEM; exact $p$ value is indicated in the Fig. **c, d** $n = 3$ independent experiments. For **e** and **h** Proteomic Assay 3, $n = 3$ independent experiments, also extracts used in **h**; quantifications in Supplementary Data 4. **i–l** $n = 3$ independent experiments. Source Data provided.

## *PIAS2b*-dsRNAi has anticancer effect in vivo against human ATC in preclinical trials with an orthotopic model

mRNA and RNAi therapies are currently being investigated as a method for treating human disease[59–61]. We performed a proof of concept experiment to show the effectivity of *PIAS2b*-dsRNAi in vivo. Populations of one anaplastic primary culture from each patient (Patient 1: T-UC1, Patient 2: T-UC3, and Patient 3: T-UC7) were transduced with our virus expressing luciferase together with mCherry for easier in vitro evaluation (Supplementary Fig. 8a–c). An orthotopic patient-derived xenograft model (ATC-oPDX) was developed by injection of those cells on the right thyroid lobe of immunodeficient mice (NOD-SCID or NSG). Follow-up was performed with the IVIS imaging system (Supplementary Fig. 8d). T-UC3 oPDX showed a weak signal in the first week after injection either in NOD-SCID, or in NSG, which did not increase with time either implanted as orthotopic at the neck or in the flank; those mice were not used for the experiment (Supplementary Fig. 8e). However, at five weeks after injection, T-UC1 and T-UC7 oPDX were growing in an exponential fashion and were used for further experiments.

A protocol for three pre-clinical trial (PCT) was designed, using IVIS imaging once a week for follow-up (Fig. 8a–c). The endpoint for all ATC-oPDX PCTs was 9–10 weeks from the orthotopic implantation of the tumor. From the beginning of week 9, daily monitoring is required until completion or reaching week 10. We could not perform longer experiments due to ethical restrictions, as the untreated oPDX ATC cancers grew as aggressively as real anaplastic cancers in patients, with similar complications of invading essential neck structures.

Each trial had a Primary Purpose of treatment effectiveness, as well as a secondary objective. Primary purpose was Treatment in all three PTC, and mice in PCT1 (T-UC1 cells) and PCT2 (T-UC7 cells) were randomized into two arms for treatment with ns-dsRNAi or *PIAS2b*-dsRNAi (Fig. 8a, b). The secondary objective of PCT1 was to determine whether twice weekly intravenous treatment through tail vein was possible, and treatment started after week 5, twice per week for 3 weeks. The secondary objective of PCT2 was to determine whether repeated intravenous treatment for many weeks was toxic, or whether the carcinomas became resistant after several weeks of treatment, and treatment started after week 3, twice per week for 6.5 weeks.

PCT3 (T-UC7 cells) was a more realistic trial (Fig. 8c). ATC patients typically present to the clinic with advanced tumors that have grown rapidly in recent weeks. PCT3 aimed to reproduce this situation and see if the treatment had an effect in these conditions. We requested an extension authorization until week 11. This was to allow us to start treating later, with larger tumors (Primary purpose), and treatment started after week 7, twice per week for 3 weeks. This more closely resembles many patients with ATC, which is why we call it

"Compassionate Use." At the same time, since the tumors already had strong signal in the IVIS, we were able to add an off-treatment period to observe tumor regrowth. The secondary objective of PCT3 was to determine what happened if the therapy was stopped (On-Off therapy) from week 10 to the extended endpoint at week 11.

In vivo systemic RNAi administration is usually coupled to lipid nanoparticles, similarly to transfecting cell cultures. However, this conduces to a major sequestering of the RNAi-lipid cargo by the liver, reducing efficiency in non-liver target organs. After reviewing the literature, we found a convincing siRNA targeting of luciferase in prostate cancer and bone metastasis xenografts by i.v. injection of luciferase siRNA coupled to AteloGene[62–64]. Thus, we injected dsRNAi-Atelogene in the tail-vein (Fig. 8).

Luminometry data in each mouse were normalized to the day of start of the treatment. Tendency Lowess curves were obtained that included all mice in both groups. In PCT1 and PCT2, there was a significant flattening of the Lowess curve of tumor growth in *PIAS2b*-dsRNAi–treated mice, while the oPDX continued growing exponentially in ns-RNAi–treated mice (Fig. 8d, e). The Lowess curves from PCT3 indicate that the treatment may even be effective in these advanced tumors reaching a significance of $p = 0.05$ after the three weeks of treatment; however, as soon as the treatment is discontinued, the tumor immediately regrows regaining $p = 0.146$ after one-week period off-treatment (Fig. 8f).

A representative mouse at IVIS of each treatment is shown in Fig. 8g. The beginning-to-end increment signal value was significantly reduced in the *PIAS2b*-dsRNAi group in PCT1 and PCT2, but reduced to $p = 0.06$ in PCT3 (Fig. 8h).

Dissection of oPDX cancers was complex due to massive extension at the neck, but in situ tumor volumes were visibly and significantly reduced with *PIAS2b*-dsRNAi in PCT1 and PCT2, but reduced to $p = 0.15$ in PCT3 (Fig. 8i and Supplementary Fig. 8f). Pathology analysis of FFPE oPDX was performed by an expert in thyroid pathology with clinically validated antibodies; note that p53 and Ki67 are specific for human proteins, while cytokeratin (CK AE1AE3), TTF1 (NKX2-1), PAX8, and TG recognize both mouse and human proteins. oPDX were highly infiltrative carcinomas of human cells (p53+, Ki67+, cytokeratin+) (Fig. 9a, b, and Supplementary Fig. 9). Tumors were undifferentiated carcinomas, negative for most of the thyroid-specific markers (TTF1, TG), which instead revealed mouse normal thyroid tissue (Fig. 9b and Supplementary Fig. 9). PAX8 expression was conserved like in the mouse thyroid, as it is in T-UC cultures and TH-UC tissues (not shown). The high Ki67 index did not vary with the treatment (Fig. 9c).

Detailed study with the microscope led to observation of what is called in clinical pathology as *apoptotic bodies*—e.g., dead/dying cells, with hematoxylin condensed chromatin and a strong pink cytoplasm

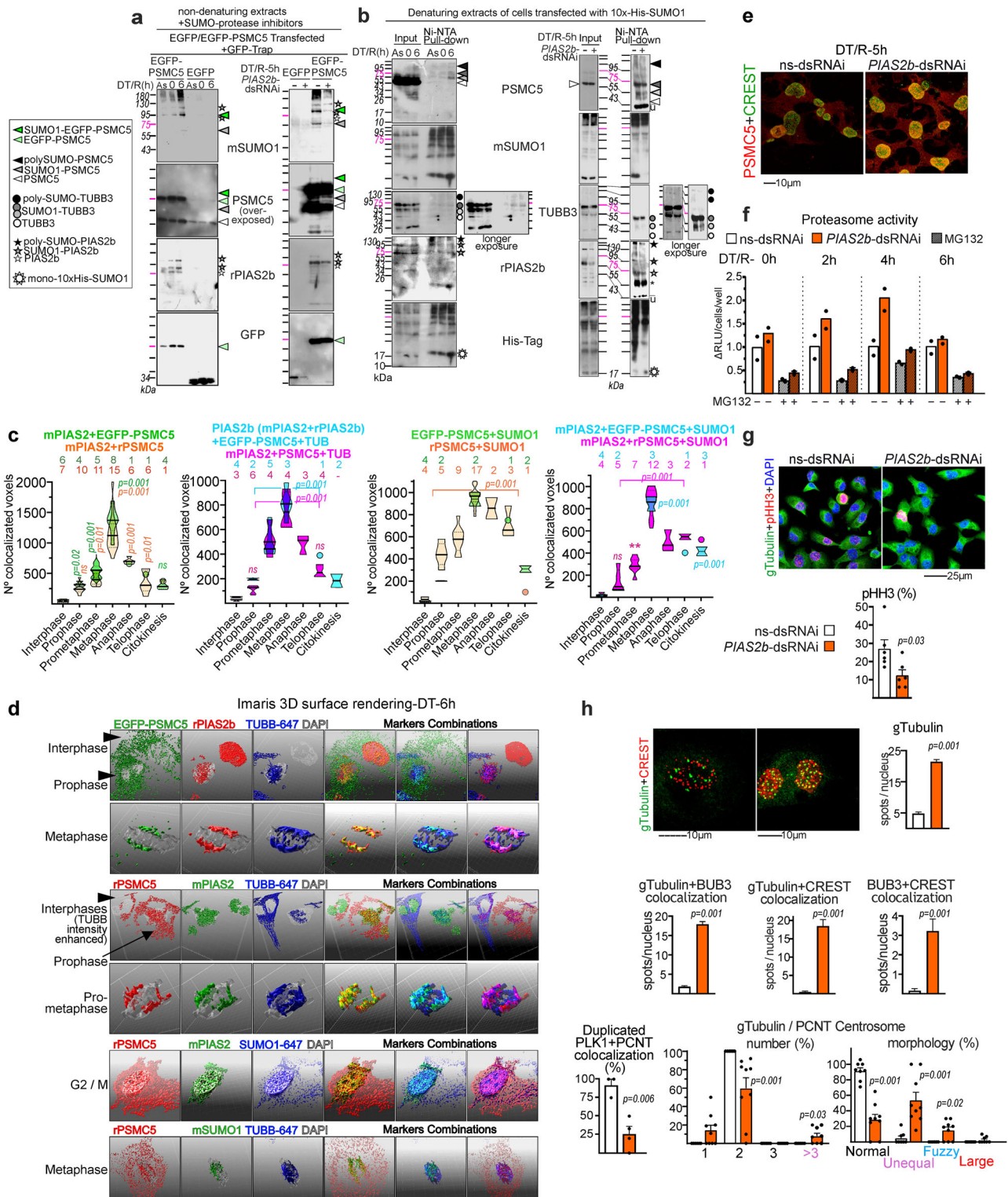

(Fig. 9d). The percentage of dead cells was significantly enhanced in the group treated with *PIAS2b*-dsRNAi in all three PCT.

Using primer sets specific for human genes, we measured mRNA gene expression in paraffin sections of both groups (Fig. 9e). *PIAS2b*, but no *PIAS2a*, was significantly reduced in *PIAS2b*-dsRNAi treated oPDX from PCT1 and PCT2, but tended to increase in PCT3, indicating a very immediate and sensitive marker of response to treatment. PLK1, one of the key genes reduced with *PIAS2b*-dsRNAi in the in vitro experiments, was also significantly reduced with *PIAS2b*-dsRNAi in

PCT1 and PCT2, but not significant in PCT3 (Fig. 9e). No significant change was found for PSMC5 gene expression.

Using immunohistochemistry, we studied two other key markers of *PIAS2b*-dsRNAi action in vitro, with the aid of AI scoring algorithms (Fig. 9f, g). Treated/Control oPDX pairs were mounted in the same slide to control for interassay staining variations. Phospho-Histone H3 immunohistochemistry presents two intensities in tumor sections, a spotty weak nuclear signal marks G2/M transition and early Prophase, while strong signal associated to hematoxylin marks Prometaphase to

**Fig. 7 | PSMC5 and TUBB3 are directly SUMO1ylated by PIAS2b, and retained at the spindle at mitotic onset.** *PIAS2b*-dsRNAi activates the proteasome too early in mitosis with catastrophic results. PSMC5 and TUBB3 PIAS2b-dependent SUMOylation at mitosis, using non-denaturing extracts with SUMO-protease inhibitors (**a**), or 10xHis-SUMO1 transfected denaturing extracts (**b**). From top-to-bottom re-stripped westerns presented. **a** GFP-Trap pull-downs of EGFP or EGFP-PSMC5 in asynchronous (As), DT/R-0 h, DT/R-6 h (Left); or treated with ns-dsRNAi or *PIAS2b*-dsRNAi at DT/R-5 h (Right). EGFP-PSMC5 and endogenous PSMC5 enhance SUMO1ylation at mitosis, lost with PIAS2b silencing. Endogenous PSMC5 associates to exogenous EGFP-PSMC5. PIAS2b is associated to- and SUMO1ylated in the GFP pull-downs. **b** 10xHis-SUMO1 Ni-NTA pull-downs in denatured extracts reveal covalent SUMO1ylated – or poly-SUMOylated, PSMC5, TUBB3 and PIAS2b at DT/R-6 h, that are lost with *PIAS2b* silencing. mSUMO1 and His-Tag patterns show enhanced SUMO1ylation at DT/R-6h, and reduced SUMO1ylation in PIAS2b absence. Unspecific bands (-u) pull-downs of double consecutively transfected, 10xHis-Tag-SUMO1+dsRNAi, required longer gel running (right). (Controls in Supplementary Fig. 7b). Western blots (anti-SUMO1 and anti-His-Tag) in Eluted lanes show similar patterns due to detection of identical SUMOylated proteins, while some variation in Input lanes reflects both SUMO1ylation and overexpressed 10xHis-tagged SUMO1ylation. However, PIAS2b depletion accumulates free mono-His-SUMO1. **c** Imaris quantification using 3D super-resolution microscopy of EGFP-PSMC5 or rPSMC5, mPIAS2 or rPIAS2b, TUBB-647 or SUMO1-647 combinations presented as violin plots of double (green, orange) and triple (blue, pink) colocalization. **d** Surface rendering of progressive mitotic phases with increasing association at the spindle (planes in Supplementary Fig. 7c). **e–g** Nuclear accumulation of PSMC5 is promoted by *PIAS2b*-dsRNAi silencing at DT/R-5 h, with earlier abnormal proteasome activity increase and reduced pSer10-HH3. **h** Mitotic perturbations at the kinetochore and centrosome at DT/R-5 h in *PIAS2b*-dsRNAi cells (pictures in Supplementary Fig. 7d–f): gTubulin + CREST colocalizing nuclear spots increase to 23, coincident with kinetochores, while the few in ns-dsRNAi are side-to-side with CREST; increased nuclear gTubulin colocalized with BUB3 + CREST (kinetochore); duplicated PLK1 (mitosis onset) and PCNT do not colocalize; gTubulin and pericentrin (PCNT) colocalization reveals qualitative and quantitative centrosome alterations. **a**, **b** $n = 3$ independent experiments. **c**, **d** Violin plots with cells per condition indicated, from $n = 5$ independent experiments; two-sided one-way ANOVA with Holm–Sidak's multiple comparison test. **f** $n = 2$ independent experiments. **g** $n = 6$ independent experiments, two-sided unpaired T-test. **h** (upper part) at least 100 cells analyzed per condition, and **h** (bottom) PLK1 + PCNT $n = 4$, Centrosomes analysis $n = 8$ independent experiments; two-sided Mann–Whitney. Bar indicate means ± SEM; exact $p$ value is indicated in the Figure. Source Data are provided.

Anaphase[65]. There was a non-significant reduction in the G2/M/early Prophase population in PCT1; however, there was a marked reduction in the Prophase-to-Anaphase mitotic population in the group treated with *PIAS2b*-dsRNAi in PCT1 and PCT2, and this was lost in PCT3 (Fig. 9e). PSMC5 presented a weak nuclear signal in control ns-RNAi–treated oPDX, mouse stromal cells were negative; with *PIAS2b*-dsRNAi nuclear staining was markedly and significantly increased in PCT1 and PCT2, but non-significant in PCT3 (Fig. 9g).

### *PIAS2b*-dsRNAi is effective in other non-thyroid cells derived from cancers similar to ATC

Finally, we addressed if *PIAS2b*-dsRNAi is effective against other non-thyroid cancers. In tissues, PIAS2b mRNA levels were very low in ATC, yet they were also low in PDTC. *PIAS2b*-dsRNAi did not have effect in cells from PDTC of papillary origin as B-CPAP, or metastatic FTC as FTC-238. Both cell lines grew as fast as the ATC cell lines and presented p53 mutations and chromosomic alterations (see Cellosaurus). Thus, any particular characteristic of ATC, such as lack of differentiation, anaplasia, or aneuploidy (or all of them combined) could be determinants for the effectiveness of *PIAS2b*-dsRNAi (Supplementary Fig. 10a).

Lack of differentiation in thyroid is defined as absence of some phenotypic characteristic thyroid markers –as TTF1 (NKX2-1) or thyroglobulin, in otherwise cytokeratin-positive (cytokeratin+) epithelial cells with some differentiation (such as maintenance of PAX8 expression). Anaplasia is defined as marked cellular atypia with pleomorphic features. Aneuploidy is defined as having a > or <2n genetic content together with polyploidy in coexisting cell populations.

In clinical pathology, there are other human non-neuroendocrine undifferentiated adenocarcinomas (expressing cytokeratins) with similar characteristics to ATC, including large cell anaplastic lung carcinoma and undifferentiated carcinomas of pancreas, stomach, liver, ovary, uterus, and prostate[66–72]. All of these have several characteristics in common, such as age at presentation (older than 50–60 y.o.), aggressive clinical course, frequent sarcomatoid or mesenchymal-like appearance in cytokeratin+ cells, coexistence of residual well-differentiated carcinoma areas, and *TP53* gene mutation (black or null immunohistochemistry patterns for p53), in a sea of other non-common genetic alterations. We selected three known cell lines derived from such cancers and transfected *PIAS2b*-dsRNAi in PANC-1 (pancreas, derived from anaplastic pancreatic ductal carcinoma) (Fig. 10a, b), COR-L23 (lung, derived from undifferentiated/anaplastic large cell lung carcinoma) (Fig. 10c, d), and HGC-27 (gastric, derived from undifferentiated gastric carcinoma) (Fig. 10e, f).

Strikingly, *PIAS2b*-dsRNAi had a potent anti-cancer effect on the three cell lines (Fig. 10a, c, e). PIAS2b protein levels were effectively down-regulated in all three lines (Fig. 10b, d, f).

Looking for markers of efficiency, we explored if the key molecular events detectable in the *PIAS2b*-dsRNAi-treated asynchronous 8305C cell line, were repeated in all ATC primary cultures (three patients, five cultures) responsive to *PIAS2b*-dsRNAi, but were not found in the spontaneously immortalized but normal thyroid origin, T-NT2 where treatment did not affect cell growth; we also compared similarly PANC-1, COR-L23, and HGC-27 (Fig. 10g). In all three ATC cultures, and non-thyroid anaplastic, undifferentiated, and aneuploid cell lines, we found consistently decreased levels of PLK1, CDK1 and gTubulin, together with increased levels of PTM-PMSC5 (p70) and >p95 PPP2CA. These proteins did not vary in T-NT2 (Fig. 10g).

Recently, aneuploidy landscape has been redefined through high-throughput sequencing and bioinformatics in human cancer cell lines[73]. We performed an in silico study with those of our cell lines that had been studied in that work. Some anaplastic cell lines (BHT-101, PANC-1) showed higher aneuploidy scores or ploidy compared to non-anaplastic lines (B-CPAP, FTC-238) but there were other anaplastic cell lines, equally responsive to *PIAS2b*-dsRNAi, presenting lower scores (CAL-62 or COR-L23) (Supplementary Fig 10b, c and Supplementary Data 2). No specific structural variants explaining *PIAS2b*-dsRNAi sensitivity were identified in anaplastic lines (Supplementary Fig. 10d, e). Moreover, flow cytometry revealed a range of aneuploidy in ATC primary cultures from similar to B-CPAP (not anaplastic) to extremely aneuploidy (T-UC3) (Supplementary Fig. 10f).

Together, these results suggest that aneuploidy per se is not the key characteristic that explains the response to *PIAS2b*-dsRNAi. This leads to the conclusion that pathology classification of Anaplasia is the key factor to respond to *PIAS2b*-dsRNAi inducing mitotic catastrophe.

## Discussion

The present study describes PIAS2b as essential for mitosis in anaplastic thyroid cancer cells. Depletion with *PIAS2b*-dsRNAi selectively kills these aggressive cancer cells by disrupting spindle assembly, chromosome-microtubule attachment, and promoting proteasome activity (Supplementary Fig. 11, Graphical Abstract). PIAS2b silencing leads to reduction (Tubulin gamma, PLK1, CDK1) and decreased SUMOylation (PSMC5, TUBB3, PPP2CA) of mitotic proteins, and mitotic catastrophe. *PIAS2b*-dsRNAi specifically kills human thyroid and non-thyroid anaplastic cancer cells, without deleterious effects on

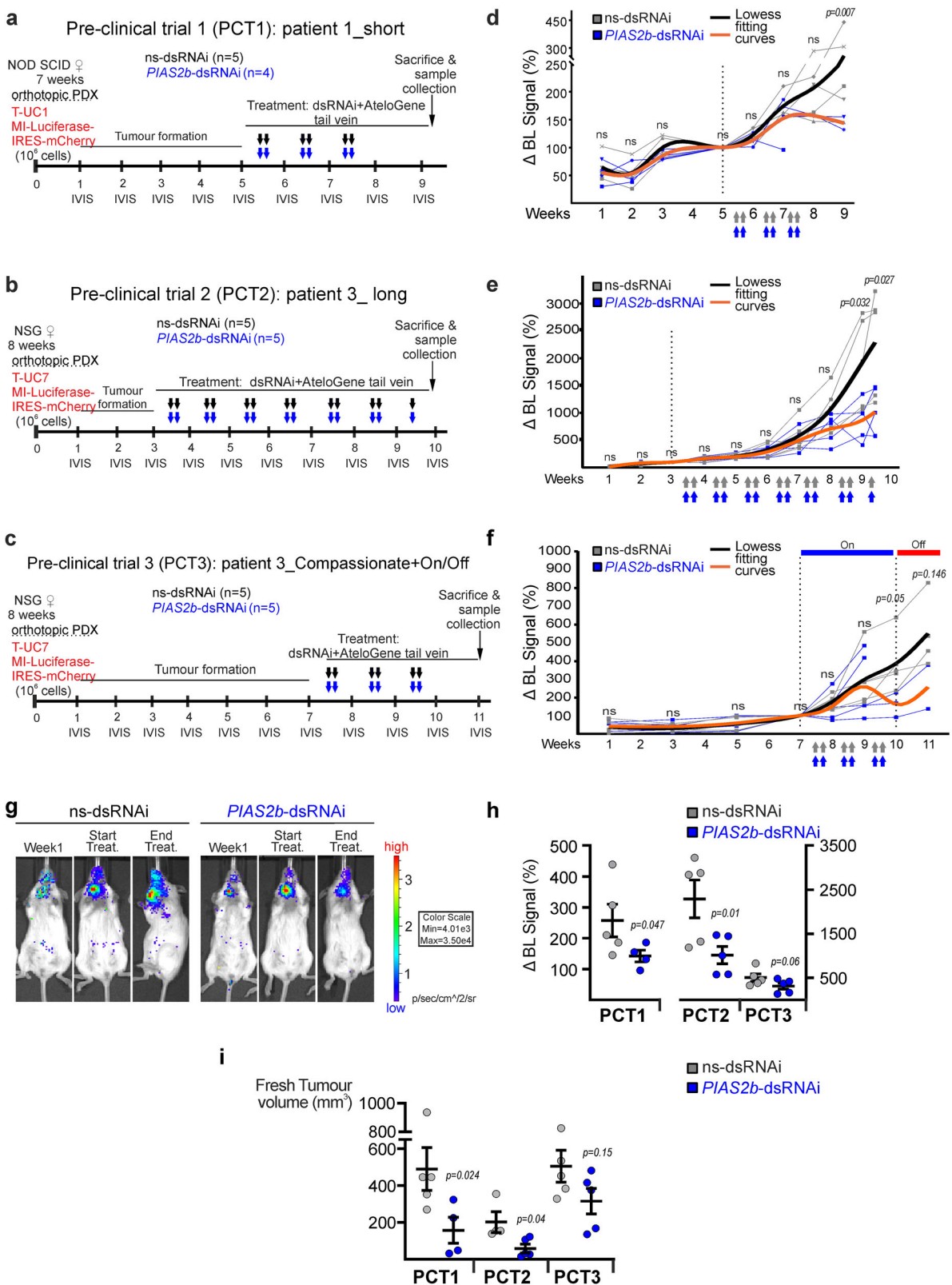

cells from normal or hyperplastic lesions, and warrants attention as a therapeutic tool for these aggressive human cancers.

Both centrosomal and kinetochore initiation events generate the molecular and mechanical events governing the mitotic spindle[74–76]. PIAS2b is an E3 SUMO-ligase that has been previously found to be dynamically SUMOylated, e.g. activated, at mitotic initiation[77,78]. SUMOylation has an essential role in mitosis that progressively is being

defined in yeast, Drosophila, Caenorhabditis, and some human cell models[32,33,79–81]. SUMO-regulated proteins have been implicated in chromosome condensation and kinetochore function at mitosis (reviewed in refs. 81–84).

SUMOylation regulates the ubiquitin-proteasome system in both meiosis and mitosis[85,86]. PIAS2b, uniquely SUMOylated, ubiquiti-nated, and CDK-phosphorylated in the cell-cycle[48], regulates

**Fig. 8 | *PIAS2b*-dsRNAi has anticancer effect in vivo against human ATC in an orthotopic model. a–c** Schematic cartoon of the design of the three preclinical trials (PCT1-2-3). NOD-SCID (PCT1) or NSG (PCT2 and PCT3) females were xeno-implanted orthotopically at the thyroid with patient-derived T-UC1 (PCT1) or T-UC7 (PCT2 and PCT3) cells (oPDX). Weeks later, intravenous treatment was started twice per week with ns-dsRNAi or *PIAS2b*-dsRNAi. Tumors were assessed weekly by luciferase intensity at IVIS device. At week 9, the endpoint was reached for PCT1 and PCT2. Primary purpose was Treatment. Secondary purpose for PCT2 was toxicity/resistance and treatment started after week 3. PCT3 was designed for Compassionate use and On-treatment started at week 7 with very advanced tumors, while Off-treatment was performed from week 10 to 11 (prolonged End-point). Arrows indicate treatment (color-coded similarly to animals): ns-dsRNAi (gray) or *PIAS2b*-dsRNAi (blue). **d, f** Lowess fitting curves performed for each treatment with all data included (black, ns-dsRNAi treatment; orange, *PIAS2b* dsRNAi treatment). Both datasets reach statistical significance at week 9 ($p = 0.01$) in PCT1 and PCT2, while no previous week presented significant differences. PCT3 datasets were $p = 0.05$ at week 10 (On-treatment) but lost significance at week 11 (Off-treatment). **g** Selected images from PCT1 of one participant in the ns-dsRNAi arm, and another from the *PIAS2b*-dsRNAi arm. **h** Increment signal at end-point compared to signal at start of treatment for each mouse in PCT1-2-3 were significantly reduced with treatment. Scale is bigger in NGS mice, due to enhanced signal. **i** Dissected oPDX fresh tumor volumes were significantly reduced in *PIAS2b*-dsRNAi treated mice. For **d–f, h**, and **i** ns-dsRNAi, $n = 5$ mice; *PIAS2b*-dsRNAi, $n = 4$–5 mice. **d–f** Two-sided two-way ANOVA with repeated measures, and fit spline/Lowess medium. For **h, i** (PCT1, PCT3) one-sided unpaired T-test, and (PCT2) one-sided Mann–Whitney. Indicated means ± SEM, and exact $p$ value. ns not significant. Source data are provided as a Source Data file.

proteasome activity at prophase in our data by interacting with PSMC5. PIAS2b depletion leads to high-molecular weight PSMC5 accumulation and untimely proteasome activation. This suggests PIAS2b restrains proteasome activity until proper timing, possibly to allow non-degradative ubiquitination or to prevent premature protein degradation. The observed anti-cancer effects of *PIAS2b*-dsRNAi in both thyroid and non-thyroid anaplastic cells, including downregulation of PLK1, gTub, CDK1, and the presence of high-molecular weight PP2CA and PSMC5 bands, might be linked to this dysregulated proteasome activity.

In addition to the role of PIAS4 E3 SUMO ligase at kinetochores, recent studies have also proposed a role for other E3 SUMO ligases in the assembly of the mitotic spindle[79,87]. Two alternative hypothesis have been proposed: i) specific targets for SUMOylation during the mitotic phases; ii) a role shepherding the multiprotein complexes as a whole, until the moment that must be disaggregated[87]. In our work, alpha, beta, and gamma tubulins, and many microtubule-binding proteins, were repeatedly found associated to PIAS2b at mitosis in thyroid anaplastic cells, and the SUMOylation of tubulins altered in *PIAS2b*-dsRNAi anaplastic cells. Tubulins beta (TUBB and TUBB4B) and alfa (TUBA1C) were also found specifically SUMO1ylated by PIAS2 in a recent high-throughput study using SAAT to delineate E3-SUMO ligases substrate specificity in asynchronous U2OS cells[50]. Tubulins are not considered as simple support filaments, but in the current concept there is a range of post-translational modifications that are becoming known and that control microtubule dynamics[51,52]. Specifically the TUBB3 is more dynamic than others[52]. Furthermore, it is known that although TUBB3 is specific for neurons in normal cells, many cancers overexpress TUBB3, which is a factor of poor prognosis and resistance to treatment[88,89]. SUMOylation of tubulins in cancer is still an open field to be studied[51,52].

In our data, PIAS2b associates with centrosomes throughout mitosis and interacts with AURKA, a key spindle assembly kinase that has also been found SUMOylated[48,90]. We found in PIAS2b-depleted cells, despite full AURKA phosphorylation, spindle abnormalities, suggesting a role for PIAS2b in regulating AURKA function. Furthermore, PIAS2b interacts with various prophase spindle proteins[42–47], and their SUMOylation or localization is altered in PIAS2b-depleted cells. We have found that at mitotic onset, PIAS2b is recruited at the duplicating centrosome. PLK1, a crucial centrosomal kinase[42,91,92] colocalizing with Pericentrin and Tubulin gamma, shows reduced levels and mislocalization in PIAS2b-depleted cells, potentially explaining the observed prophase centrosome defects. Our proteomic analyses revealed interaction of PIAS2b with 14 proteins linked to centrosome function[58], including RANGAP1 (strongly reduced) and PPP2CA (decreased SUMOylation and high-molecular weight band in PIAS2b-depleted cells). These findings suggest PIAS2b plays a critical role in regulating spindle assembly and function through interactions with centrosomal kinases and structural proteins.

PIAS2b depletion likely disrupts kinetochore-microtubule attachment through its interaction with RANGAP1, a SUMO E3 ligase associated with kinetochores at metaphase and anaphase[93,94]. RANGAP1 is associated to PIAS2b in our data, and RANGAP1 is strongly reduced in *PIAS2b*-dsRNAi–treated cells. This could explain the observed accumulation of BUB3 and unattached gTub at kinetochores in PIAS2b-depleted cells. Moreover, Siz1/Siz2 −PIAS2 homologs in yeast, SUMOylate shugosin/PP2A at kinetochores, promoting bi-orientation[95]. PIAS2b may similarly regulate kinetochore-microtubule attachment via PPP2CA SUMOylation in human cells.

Our preclinical trial in ATC-oPDX mice demonstrated significant efficacy of PIAS2b-dsRNAi in reducing tumor growth, with no observed toxicity or resistance. Tumors regrew after treatment cessation, highlighting the need for combination therapy including surgery. Pathological analysis revealed increased cell death, reduced PIAS2b expression, and markers of mitotic catastrophe and proteasome activation in treated tumors. This type of cell death is characterized by condensed and shrinking nuclei and non-swollen cytoplasms[96,97]. This therapy-induced cell death has also been observed in patients, for example in immunotherapy-treated cancers[97]. These findings suggest *PIAS2b*-dsRNAi as a promising therapeutic for ATCs. The use of in vitro transcribed dsRNAi offers a potential logistical advantage based on the rapid development of mRNA vaccines[98].

*PIAS2b*-dsRNAi specifically targets ATCs despite high PIAS2b expression in PTCs wherein is a marker of poor prognosis. In PTCs, PIAS2 may be involved in transcription regulation (e.g., p53 SUMOylation) more relevant to their slow proliferation[99]. Conversely, ATCs with high mitotic index and p53 mutations likely rely on PIAS2b for mitosis and cell cycle control. This will need to be clarified in future research.

The dosage sensitivity of PIAS2b in ATCs might be explained by its role in protein complexes or essential survival pathways unique to these aggressive cancers[100,101]. Topology[102] or the assembly mechanism[101] are factors for dosage-sensitivity. Aneuploidy has been associated to cellular stress, altered proteostasis, and both sensitivity and resistance to therapy[73,103–105]. While aneuploidy is a common feature of ATCs[73], we did not observe a direct link between aneuploidy and *PIAS2b*-dsRNAi response. We propose that additional molecular mechanisms linked to anaplasia and non-differentiation contribute to PIAS2b's dosage sensitivity in ATCs.

Limitations of our study include: i) we performed the preclinical cellular work in vitro. However, we have found the same anticancer results of *PIAS2b*-dsRNAi in commercial ATC and non-thyroid anaplastic cell lines, as well as in three patient ATC primary cultures, and in the oPDX in vivo; ii) SUMOylation has been studied selecting proteins and not directly in peptides since this required alternative cell-altering approaches[50,106], although we have validated two selected targets; iii) our ATC patients were all women; however, this is also the most affected sex in ATC; and iv) only oPDX from two out of three patients

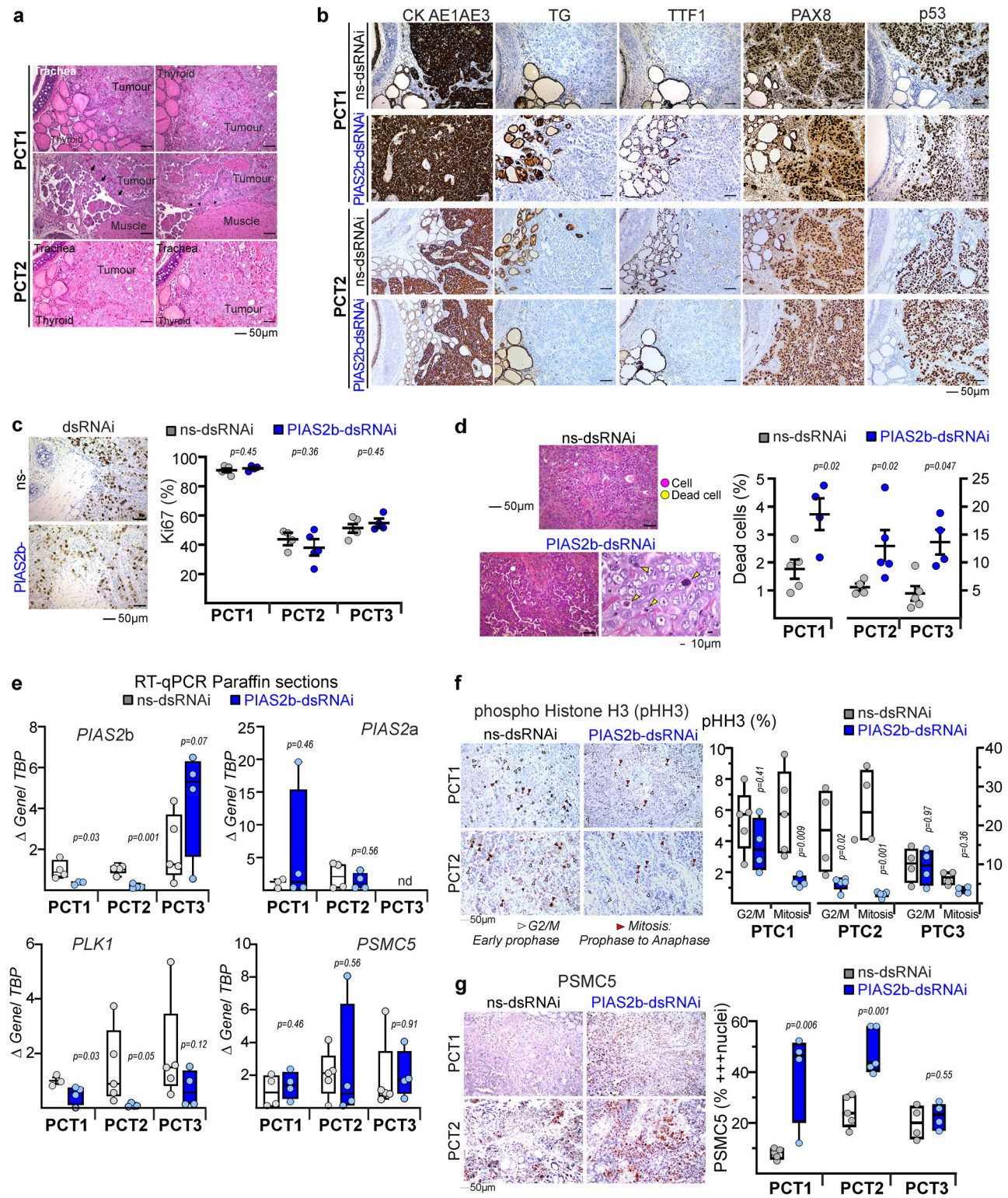

grew in the immune-deficient mouse models; we have tried to compensate this by increasing the number of mice per trial.

In summary, we have demonstrated that PIAS2b is a dose-dependent protein in ATC. The PIAS2 isoform b has an essential role in the early steps of mitosis bound to the mitotic spindle and controlling the proteasome activity in prophase. *PIAS2b*-dsRNAi kills human thyroid and non-thyroid anaplastic cancer cells through mitotic catastrophe but does not affect poorly differentiated or differentiated thyroid cancer cells, benign thyroid proliferation or normal thyroid

cells. *PIAS2b*-dsRNAi is also effective in an in vivo model of ATC, similar to the human cancer, and should be explored as a therapeutic possibility for these aggressive human cancers.

## Methods
### Ethical information
The study has the approval of the regional committee on ethics in clinical research (Comité Ético de Investigación Clínica de Galicia, CEIC, Servizo Galego de Saúde, SERGAS), and is registered with the

**Fig. 9 | Efficacy markers of *PIAS2b*-dsRNAi obtained in vitro are validated by the pathological analysis of tumors in vivo. a** H&E stained sections of different oPDX cancers from ns-dsRNAi mice of PCT1 and PCT2. Tumors were highly invasive for thyroid and local structures, such as muscles (arrowheads), nerves, trachea, vessels (arrows). **b** Immunohistochemistry of markers of clinical pathology classified all oPDX as ATCs, and did not reveal differences with treatment. Tumors were intensively positive for cytokeratins (CK, AE1-AE3), PAX8, and p53 (specific human detection), but were negative for normal thyroid markers, such as thyroglobulin (TG) or NKX2-1 (TTF1), positive in the mouse thyroid. **c** Ki67 index was very high (as expected) in ATC. No differences were found with treatment. **d** Assessment of dead tumor cells (so-called *apoptotic bodies* in clinical pathology). Magenta spots, normal H&E staining; yellow spots, condensed hematoxylin, with strong eosinophilic-stained cytoplasm. Lower right: magnification, showing significantly increment of dead cells (yellow arrowheads) in *PIAS2b*-dsRNAi treated oPDX of the three PCT (1, 2

and 3). **e** Human *PIAS2b* and *PLK1* gene expression was significantly reduced in *PIAS2b*-dsRNAi tumors of PCT1 and PCT2, but not PCT3. No change was found in PIAS2a or PSMC5. **f** phospho-Histone H3 (pHH3) immunohistochemistry revealed non-consistent reduction in early G2/M cells (spotty weak signal), but significant reduction in mitotic cells (Prophase to Anaphase) in *PIAS2b*-dsRNAi tumors of PCT1 and PCT2, but not PCT3. **g** PMSC5 immunohistochemistry showed increased intense nuclear staining in in *PIAS2b*-dsRNAi tumors of PCT1 and PCT2, but not PCT3. For **a**–**g** ns-dsRNAi, *n* = 5 mice; *PIAS2b*-dsRNAi, *n* = 4–5 mice per PCT; **c**, **d**, **g** two-sided unpaired T-test; **e** two-sided Mann–Whitney; **f** two-sided one-way ANOVA with Dunn's multiple comparison test. Represented: (**c**, **d**) means ± SEM, and (**e**–**g**) center median, bounds of box 25–75% percentile, whiskers minima and maxima. Indicated exact *p* value. nd not detected. Source data are provided as a Source Data file.

number 2016/239 and 2019/533 (to CVA). TIROCHUS is a registered collection at the ISCIII national bank with the number n° c.0003960 (to JMC-T). Informed consent was obtained from each patient with a specific signed document updated to the current legislation. The sample series from Italy were approved by the Ethical Committee of the IRCCS Istituto Auxologico Italiano (#2018_09_25_04 to LF). Animal procedures were carried out under the Procedures Act n° 15003/14/005 (to CVA), granted by Galicia Regional Government, following NC3Rs' ARRIVE guidelines. Patient data are given in Supplementary Data 2.

### Inclusion and diversity
Primary ATC patients could not be selected and entered in the study consecutively as diagnosed. Commercial ATC cell lines were selected to support different genetic backgrounds from Asian and European original patients.

The PI of the group supports gender and minorities access to translational research from its foundation, and this is reflected in group members. Authors are listed by their roles in the manuscript. CIMUS has a Committee in charge of gender equality. The USC has a specific policy to prevent any kind of discrimination.

### Cell culture: primary cell culture and cell lines
Cultures were carried out from surplus tissues of surgical specimens with the h7H system for human cells[22,23,107,108]. Cultures were maintained with passaging at 1:2 dilution. Cell lines were obtained from ECACC and DSMZ and authenticated every year through short-tandem repeats (STRs) following ICLAC guidelines as compared to profiles in Cellosaurus https://web.expasy.org/cellosaurus/.

### DNA and RNA extraction, STR profiles, and qRT-PCR from cultures and paraffin sections
Tissue extraction was performed with the Allprep DNA/RNA Mini kit 50 (80204, QIAGEN). Genetic profile (STRs) of the primary cultures and their tissues, the AmpFlSTR® Identifiler® Plus PCR Amplification Kit PCR kit (Applied Biosystems). For cells, DNA extraction was performed with the DNeasy blood & tissue kit 50 (69504, QIAGEN), and RNA was extracted using TRIzol™ (15596026, Invitrogen), treated with RNase-free DNaseI (EN0521, Thermo Fisher) and then purified using the GeneJET RNA Cleanup and Concentration Micro kit (K0842, Thermo Fisher). For mouse oPDX paraffin sections, RNA extraction was performed with the Allprep DNA/RNA FFPE kit 50 (80234, QIAGEN) using 16 FFPE tissue sections of each tumor, each with a thickness of 30 µm.

### dsRNAi design, enzymatic synthesis and transfection. Commercial siRNAs. Antisense locked nucleic acid (LNA) gapmers. siPOOLs
A double-strand interfering RNA (dsRNAi) was designed and synthesized for the PIAS2b isoform, and a negative control with no human target sequence, ns-dsRNAi, based on our previously used methods[37].

The bioinformatics program BlockiTTM siRNA Designer, Ambion, was used, and sequences were BLASTED with reviewed for max score (>31), e-values (<1). Ensembl and NCBI were also used in the search. dsRNAi were synthesized with Silencer® siRNA Construction Kit (AM1620M, Life Technologies Corporation; now Thermo Fisher Scientific).

Transfection conditions were standardized for each cell line or primary culture, and type of well. (see Supplementary Data 2 and Source Data Key Resources Table).

For transfection a suitable amount of dsRNAi, DNA or both is added to a volume of medium without serum and antibiotics, usually h7H medium without additives. The Turbofect/Viafect/Lipofectamine reagent is vortexed and the required amount to the diluted dsRNAi/DNA is added. The mixture is incubated at room temperature for 15–20 min. During this time, in the case of cultures plated 48 h before, the medium of the wells is changed to fresh one. The mixture of DNA/dsRNAi and Turbofect/Viafect/ Lipofectamine is droped by droped to each well. Cells are keeped at 37 °C and 5% CO2. After 8 h the rest of the medium is added to each well.

Commercial *PIAS2* siRNA sets from Dharmacon, and Custom siRNAs with our sequence were prepared by Life Technology, Thermo-Fisher either as usual siRNAs (Custom) or as Custom Select, an upgraded quality with longer stability and half-life. SiRNAs were transfected following manufacturer indications.

Antisense (as) LNA gapmers or control as-LNA gapmers labeled with 56 FAM oligonucleotides (11568846, Invitrogen) were resuspended in a TE-buffer solution.

Commercial siPOOLs have been designed for high efficiency with low off-target[41]. Two PIAS2 siPOOLs and the Negative Control siPOOL (siPOOLs; siTools Biotech GmbH, Planegg, Germany), were transfected in a range of concentrations using lipofectamine RNAiMAX (13778075, Thermo Fisher Scientific). Commercial PIAS2 siPOOL includes 30 designed siRNAs against all PIAS2 RNA isoforms. Custom-prepared PIAS2b siPOOL includes 30 designed siRNAs against the PIAS2 RNA isoforms codifying PIAS2b exclusively, the same isoforms that targets our PIAS2b-dsRNAi (13 RNA isoforms).

### Exogenous PIAS2 plasmid transfection. Exogenous EGFP-PIAS2b protein transfection
The N-terminal Flag-labeled Flag-hPIAS2a and Flag-hPIAS2b were purchased from Addgene. EGFP reporters pEGFP-C1 and pEGFP-C1-hPIAS2b were received from S. P Jackson[109]. Plasmids were re-sequenced at our lab. PIAS2b dead constructs (Flag-hPIAS2b-C362A, pEGFP-C1-hPIAS2b-C362A) were derived from the wild type plasmids, custom prepared and fully sequenced in GeneScript Europe. Cells were transfected in a 96-multiwell culture dish with a non-lethal amount of pEGFP-C1-hPIAS2b or vector pEGFP-C1 with Turbofect or Lipofectamine 3000 (L3000015, Thermo Fisher Scientific). To maintain transfection efficiency at 0.2 µg/well of DNA, 0.19 µg of empty pcDNA3 plasmid was added to each well, along with 0.01 µg of the PIAS2b plasmid.

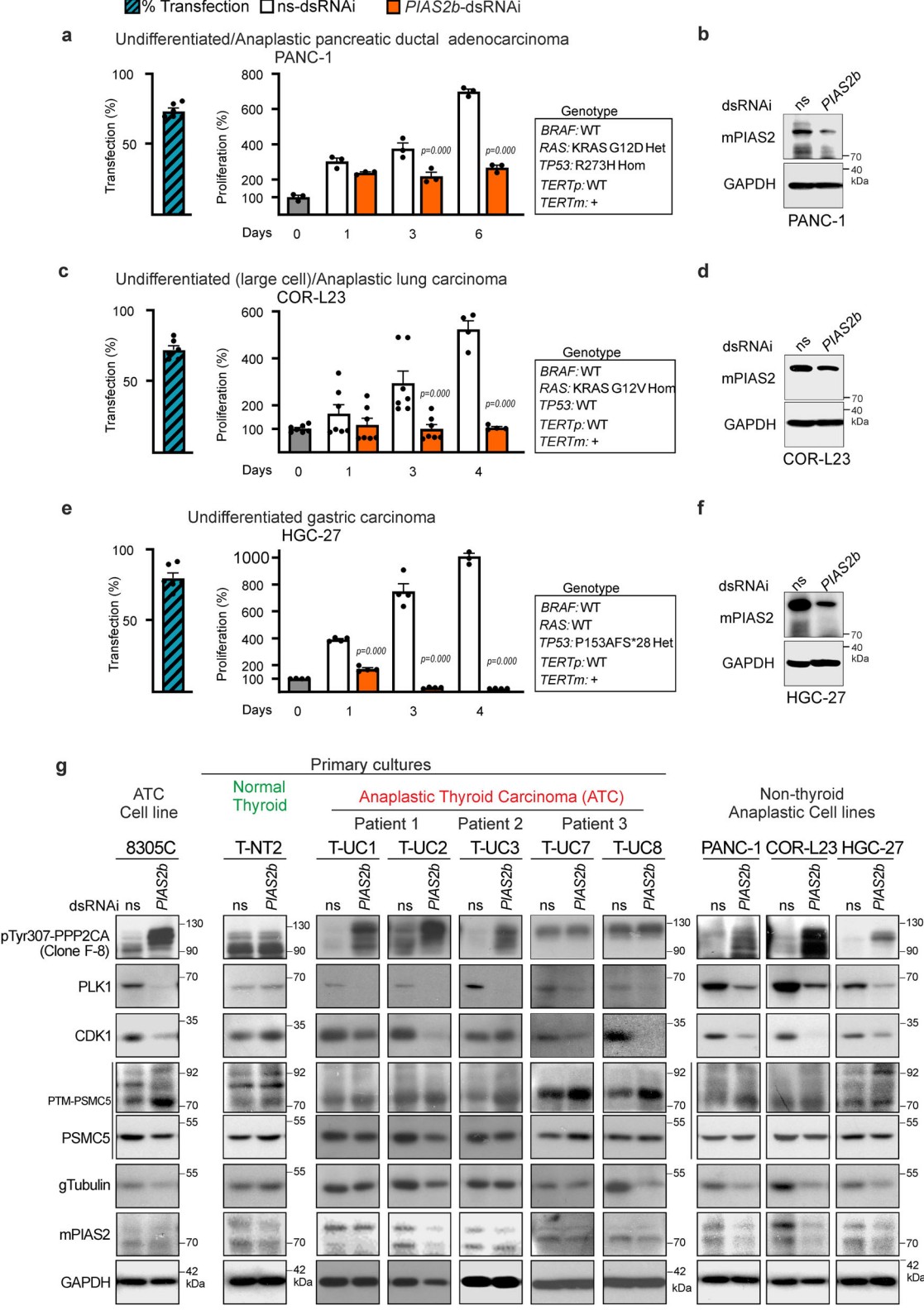

For direct protein transfection, first EGFP-PIAS2b was purified with the GFP-Trap (gtak20-20rxns, Chromotek) from pEGFP-C1-PIAS2b transiently transfected 8305 cells. After washing, protein was eluted following manufacturer instructions, and the concentration was measured. Cells were grown in mw96, and synchronizaed at mitosis. At DT-0h, 0.2 microg/well of purified EGFP-PIAS2b protein was mixed with 5 microL Xfect buffer and 0.75 microL Xfect Protein Transfection Reagent (Takada), and 40 microL serum-free medium, to be added to the wells until fixation (DT-6h).

**Viral infection**

For infection, PIAS2 or the non-target TRIPZ inducible lentiviral shRNA system (RHS4696-200680115, RHS4743 Thermo Fisher Scientific) was used. Viruses were prepared from HEK293FT cells, with the CaCl2

**Fig. 10 | *PIAS2b*-dsRNAi has a similar anti-cancer effect in thyroid and non-thyroid cancer cell lines originated from undifferentiated/ anaplastic cancers of different organs, sharing a common mechanism. a–f** Three cell lines originated in cancers with confirmed diagnostic (Cellosaurus database) of anaplastic cancers were selected: **a**, **b** PANC-1 (undifferentiated/anaplastic pancreatic ductal adenocarcinoma); **c**, **d** COR-L23 (undifferentiated (large cell)/anaplastic lung carcinoma); **e**, **f** HGC−27 (undifferentiated gastric carcinoma). Genotype is shown in the box at the right. Transfection efficiency is shown at the left (blue hatched bars). *PIAS2b*-dsRNAi silencing has a significant inhibitory effect in cell growth, decreasing the number of cells, as compared to ns-dsRNAi (**a**, **c**, **e**). In parallel, *PIAS2b*-dsRNAi downregulated PIAS2b protein (**b**, **d**, **f**). **g** The main mitotic protein alterations induced by *PIAS2b*-dsRNAi detected in asynchronous cultures are shared by all undifferentiated/ anaplastic cells (ATC cell line: 8305C; primary cultures from three patients, and non-thyroid anaplastic cell lines), but are not found in normal accumulation thyroid cells (T-NT2), in which *PIAS2b*-dsRNAi has no anti-cancer effect. *PIAS2b*-dsRNAi downregulated PIAS2b protein in all cell lines and cultures irrespective of their origin. *PIAS2b*-dsRNAi downregulated PLK1, CDK1, tubulin gamma (gTubulin) in the undifferentiated/anaplastic cell lines cell lines and cultures, but not in T-NT2. *PIAS2b*-dsRNAi induced the of the high–molecular weight PPP2CA band (detected with clone F-8 antibody, formerly known as pTyr307 PPP2CA antibody), and the accumulation of the higher p70/p95 PTM-PMSC5 band in the undifferentiated/anaplastic cell lines and primary cultures, but not in T-NT2. For **a** $n = 3$, **c** $n = 7$, and **e** $n = 4$ independent experiments; Two-sided two-way ANOVA with repeated measures, with Sidak's multiple comparison test. Bar indicate means ± SEM; exact $p$ value is indicated in the figure. **g** $n = 3$ independent experiments. Source data are provided as a Source Data file.

method. After infection, positive cells were selected with puromycin (10 µg/mL). Cells were grown in tetracycline-free FBS (631106, Clontech), and a doxycycline negative/positive selection strategy was designed using cell sorting by FACS, to obtain the regulated lines T-PIAS2_shRNA CAL-62 and T-Scr_shRNA with high (HI) or sufficient (SU) RFP expression.

## Cell lysis, immunoblot, immunoprecipitation, GFP pull-down (GFP-Trap) and 10xHis-SUMO1 pull-down

In non-synchronized cultures the appropriate volume of lysis buffer (50 mM Hepes pH 7.4 (H3375, Sigma), 1% Triton X-100 (T8787, Sigma), 10% Glycerol (G9012, Sigma), 150 mM NaCl (S9888,Sigma), 5 mM EGTA (E5134, Sigma), 1.5 mM MgCl2 (M8266, Sigma), 20 mM Na-pyrophosphate (S9515, Sigma), 2 ug/mL Aprotinin (A6279, Sigma), 4 mM PMSF (P7626, Sigma) and 92 µg/mL Na3VO4 (S6408, Sigma) is added to each sample and it is spread well with a plunger of a syringe. After 20 min incubation on ice, cells were scrapped and centrifuged at 12.000 rpm (centrifuge 5415 R, Eppendorf) for 10 min at 4 °C.

In mitotic synchronized cultures we used SUMO-lysis buffer [20 mM Tris-HCl pH 7.5, 150 mM NaCl, 0.5% Triton X-100, 1 mM MgCl2, 20 mM N-Ethylmaleimide, all from Sigma; and 1 tablet of protease inhibitors complete Mini EDTA-free (11836170001, Roche) and 500 Units/sample of Benzonase Nuclease (E1014, Sigma)].

For the quantification of the protein concentration, the commercial kit RC DC ™ Protein Assay Kit II (5000122, Bio-Rad) was used. BSA (bovine serum albumin) was used as a standard.

50 µg of protein was loaded per lane in a 9% SDS-PAGE to detect the PIAS2 protein, and 12% SDS-PAGE for the other proteins in GE Healthcare miniVE Vertical Electrophoresis System (80-6418-77, Fisher Scientific). Transfer to the membrane (Immun-Blot PVDF 0.2 µm, 162-0177, Biorad) was performed using semi-dry blotter system (Z34 050-2, Sigma).

For immunoprecipitation, 125−200 µg protein was incubated with gentle shaking overnight with 0.1 µg PIASx antibody (D-12, Santa Cruz Biotecnology) or non-immune isotype mouse IgG2a (M5409, Sigma-Aldrich); FLAG M2 antibody (F3165, Sigma), TUBB3 (TU-20, Santa Cruz Biotecnology) or non-immune isotype mouse IgG1 (M5284, Sigma). The total volume of each tube was 500 µL after equalizing with same buffer as lysates. The next day, 60 µL of protein G-Sepharose (17-0885-01, Gamma Bind, GE Healthcare) was diluted in 100 µL of lysis buffer. A common batch was prepared for all samples plus one. 100 µL/tube of this suspension was added and incubated for 1 hour at 4 °C. Tubes were centrifuged at 12,000 rpm (5415 R centrifuge, Eppendorf) for 1 min at 4 °C and the supernatant was discarded. 1 mL of HNTG buffer (20 mM Hepes pH 7.5, 15 mM NaCl, 10% glycerol, 1% Triton X-100) at 4 °C was mixed and centrifuged at 12,000 rpm, 2 min, 4 °C. In total, 5 washes were carried out with HNTG buffer.

Sample was boiled at 95 °C with 30 µL of 1.5X loading buffer to proceed with the standard western blot.

For GFP-Trap pull-down, cells were transfected with EGFP constructs using Turbofect (R0531, Thermo Fisher). Cells were lysed using SUMO-lysis buffer [20 mM Tris-HCl pH 7.5, 150 mM NaCl, 0.5% Triton X-100, 1 mM MgCl2, 20 mM N-Ethylmaleimide, all from Sigma; and 1 tablet of protease inhibitors complete Mini EDTA-free (11836170001, Roche) and 500 Units/sample of Benzonase Nuclease (E1014, Sigma)]. The lysates were then centrifuged for 1 hour at 20,000 × g and 4 °C. For pull-downs, 2 mg of lysates containing EGFP-PIAS2 or EGFP-PSMC5 were incubated with GFP-Trap beads (Chromotek) at 4 °C overnight with rotation following the manufacturer's instructions.

For 10xHis-SUMO1 pull-down, 8305C cells were transfected with 10 µg 10xHis-SUMO1-WT-IRES-GFP using Turbofect and synchronized with double-thymidine block. In another experiment, one day after the SUMO1 transfection, ns- or *PIAS2b*-dsRNAi was subsequently transfected and immediately after, the double thymidine block was started. At release, cells were scrapped with ice-cold PBS and centrifuged for 2 min at 500 × g. The pellet was lysed in 300 µL of lysis buffer (6 M Guanidine-HCl, 0.1 M Na₂HPO₄, 0.1 M NaH₂PO₄, 0.01 M Tris-HCl, pH 8.0) while vortexing vigorously. After thar, the lysate was snap freeze in liquid nitrogen, thawed and sonicated twice for 5 s at ~30 Watts using a microtip sonicator (Digital Sonifier S450D, BRANSON). The lysates were equalized before adding imidazole to 50 mM and β-mercaptoethanol to 5 mM and mix on a rotating mixer while preparing the Ni-NTA beads. 20 µL of agarose bead were used per sample, washing them 4 times with guanidine lysis buffer supplemented with 50 mM Imidazole pH 8.0 and 5 mM β-mercaptoethanol before incubation with the samples overnight at 4 °C in a rotating mixer. The beads were spinned down at 500 × g for 2 min, the supernatant removed and were wash with 1 mL of wash buffer 1 (6 M Guanidine-HCl, 0.1 M Na2HPO4 pH 8.0, 0.1 M NaH2PO4 pH 8.0, 0.01 M Tris-HCl, pH 8.0, 10 mM imidazole pH 8.0, 5 mM β-mercaptoethanol, 0.1% Triton X-100) and wash buffer 2 (8 M Urea, 0.1 M Na2HPO4, 0.1 M NaH2PO4, Tris-HCl, pH 8.0, 10 mM Imidazol pH 8.0, 5 mM β-mercaptoethanol, 0.1% Triton X-100). Beads were washed for 15 min in a rotating mixer with wash buffer 3 (8 M Urea, 0.1 M Na2HPO4 pH 6.3, 0.1 M NaH2PO4 pH 6.3, Tris-HCl, pH 6.3, 10 mM Imidazol pH 7.0, 5 mM β-mercaptoethanol) and twice with wash buffer 4 (8 M Urea, 0.1 M Na2HPO4 pH 6.3, 0.1 M NaH2PO4 pH 6.3, Tris-HCl, pH 6.3, 5 mM β-mercaptoethanol). Pellet beads at 500 G for 2 min and remove supernatant completely after every wash. Samples were eluted in the same beads volume of Elution buffer (1 M Tris pH 7.0, 1 M Na2HPO4, 1 M NaH2PO4, 9 M urea, 5 M imidazole) for 30 min in a rotating mixer and spin down at 10.000 RCF to separate the elution from beads. Elution was repeated twice to ensure full recovery of SUMOylated proteins.

## Cell viability assay and cell death assays

Proliferation and apoptosis assays in 96-multiwell plates were performed using the MTT and Hoechst method as described[37]. To distinguish the type of cell death, the following inhibitors were used: for apoptosis, caspase 3 (A0835, Sigma-Aldrich) and caspase 9 inhibitor (SCP0095, Sigma-Aldrich), and for necroptosis, the RIPK1 inhibitor II (504297, Calbiochem).

## Cell cycle analysis by flow cytometry and double-thymidine block

A minimum of 240,000 cells were seeded per condition or sample; the procedure was adapted from ref. 110. Cells were stained with 1 mL DAPI solution (1 μg/mL DAPI [D9542, Sigma-Aldrich] and 0.1% Triton (T8787, Sigma-Aldrich) in PBS) for every $1 \times 10^6$ cells. A total volume of 300 μL of each sample was distributed into a 96-well multiplate and analyzed on the BD Accuri™ C6 cytometer (BD Biosciences) using the violet laser (405 nm). Data were analyzed with FlowJo software v_10.1 (BD Biosciences).

To study mitosis, cells were synchronized with a double-thymidine block as previously described[111,112]. Cell synchronization was validated by flow cytometry.

## Time-lapse imaging

A 96-well multiplate was used in an automated stage Leica microscope (CTR7000 HS, Leica) equipped with a humidified cell incubation chamber at 37 °C and 5% CO₂, using 0.1 μM SiR-DNA (SC007, Spirochrome) as a marker. The H2B-GFP plasmid (11680, Addgene) was co-transfected to specifically label chromatin.

## LC-MS/MS proteomic assays

The project contains three proteomic assays performed in human 8305C ATC cell line synchronized at mitosis through double-thymidine block (19-h first thymidine block, mitotic shake-off, followed by 6 h release, and a second 19-h thymidine block, mitotic shake-off, and final release for 6 h, DT- 6 h).

All conditions were analyzed by replicates obtained from 3 to 5 independent experiments. In each experiment, PIAS2 and control conditions were seeded, transfected, extracted, or performed in parallel.

**Proteomic assay 1.** DT-6h cells were lysed as described in refs. 107,113,114 using Glycerol Lysis Buffer [50 mM HEPES pH 7.5, 150 mM NaCl, 10% glycerol, 1% Triton X-100, 5 mM EGTA, 1.5 mM MgCl2, 2 μg/mL aprotinin, 4 mM PMSF, 92 μg/mL Na orthovanadate, 20 mM Na pyrophosphate; all from Sigma]. Protein concentration was measured using RC-DC protein assay (BioRad).

1.5 mg protein per final proteomic sample was distributed in 6 parallel 2 mL tubes (0030108132, Protein LoBind Tubes, Eppendorf); in each, 250 μg of protein extracts was mixed with 0.2 μg of the antibody against PIAS2 (mPIAS2, clone D-12, Santa Cruz Biotec sc-166494) or 0.2 μg of mouse immunoglobulin of identical isotype IgG2a (M5409, Sigma-Aldrich). The total volume of each tube was 500 μL, after adding equalizing volume of lysis buffer. Incubation was made overnight at 4 °C in an end-to-end mixer.

The next day, 60 μL of protein G-Sepharose (17-0885-01, Gamma Bind, GE Healthcare) was diluted in 100 μL of glycerol lysis buffer. A common batch was prepared for all samples plus one. 100 μL/tube of this suspension was added and incubated for 1 h at 4 °C. Tubes were centrifuged at 12,000 rpm (5415 R centrifuge, Eppendorf) for 1 min at 4 °C and the supernatant was discarded. 1 mL of HNTG buffer (20 mM Hepes pH 7.5, 15 mM NaCl, 10% glycerol, 1% Triton X-100) at 4 °C was mixed and centrifuged at 12,000 rpm, 2 min, 4 °C. In total, 5 washes were carried out with HNTG buffer.

In the last wash, all 6 tubes of each sample were combined in a single tube, for a final proteomic sample. 1/10 of the sample was boiled at 95 °C with 30 μL of 1.5X loading buffer to proceed with the standard western blot. 9/10 were subsequently processed for LC-MS/MS.

This experiment was performed independently five times; all samples were analyzed in a single batch.

For LC-MS/MS identification, samples were processed as described in Shevchenko et al.[115]. 90% of the final immunoprecipitation was boiled in the presence of SDS at 95 °C for 5 min to break the antibody-protein binding and concentrated on a 10% SDS-PAGE gel. The gel was allowed to run until the front entered 3 mm into the resolving gel. The gel was fixed and stained with Coomassie (Lonza). Bands were excised with a scalpel and incubated with 10 mM DTT (10197777001, Sigma) for reduction. After washing 3× with 50% methanol in water, bands were dehydrated with acetonitrile, and alkylated with 55 mM iodoacetamide (I6125, Sigma). After washing and dehydrating again, tubes were dried in a SpeedVac.

The gel pieces were rehydrated at 37 °C for 1 min. Tryptic digestion was carried out by adding modified porcine trypsin (V5280, Promega) at a final concentration of 20 ng/μL in 20 mM ammonium bicarbonate for 16 h at 37 °C. The tryptic peptides generated were extracted by three 20-min incubations in 40 μL of 60% acetonitrile and 0.5% formic acid, pooled, and vacuum dried in a SpeedVac to be stored at −20 °C until use.

Analysis by LC-MS/MS using a triple TOF 6600 SCIEX was performed. The digested peptides were resuspended in 20 μL of mobile phase A (H2O with 0.1% formic acid LC-MS quality), by sonication for 10 min. 4 μL of each sample were injected in a loop (10 μL/min) where the peptide mixture was correctly resuspended in mobile phase A (H2O with 0.1% formic acid LC-MS quality). The peptides were transferred to a YMC TRIART C18 silica-based reversed phase guard column, 5 ×0.5 mm, 3 μm in particle size and 120 Å pore size (YMC Technologies) where possible contaminants that could damage the column were removed. This guard column is connected online to a YMC-TRIART C18 capture column, 150 ×0.30 mm, with a particle size of 3 μm and a pore size of 120 Å (YMC Technologies) where the peptides are separated by polarity at a speed of 5 μL/min. The elution gradient of the peptides ranges from 2% to 90% mobile phase B (acetonitrile (ACN) with 0.1% formic acid LC-MS quality). For DDA analysis a 90 min gradient was used, while for SWATH a shorter gradient of 40 min was used.

The following conditions were used for the sources and interfaces: ion spray floating voltage (ISVF) of 5500 V, collision energy (CE) of 10, and ion source gas 1 (GS1) of 25. The ions were selected based on their mass-to-charge ratio (m/z) being greater than 350 and less than 1400, with charge states of +2−5, a mass tolerance of 250 ppm, and an abundance threshold of more than 200 cps. The target ions from the above analysis were removed every 15 s. The equipment is automatically calibrated every 4 hours using a mixture of tryptic peptides known as pepCalMix (SCIEX).

Technical replicates were added to biological replicates for the final data processing.

Data-dependent acquisition (DDA) was performed using Analyst TF 1.7.1 (SCIEX) software. After the acquisition of the LC-MS/MS data, the files were processed with the ProteinPilot™ 5.0.2 (SCIEX) software using the Paragon™ (5.0.2) algorithm for the database search and Progroup™ for the grouping of the data. Searches were performed using a human-specific UniProt database (UniProt, PMID: 30395287; https://www.uniprot.org/uniprot) (UniProt release 2016-2, Swiss-Prot). For the initial main search, parent peptide masses were allowed mass deviation of 20 ppm. The mass ranges we used was our usual range 300–1400, raised to 1800 to include PTMs in the search, which increase the mass of the peptide. Using this software, it is not necessary to specify mass tolerances, individual modifications to search for, expected fragment ion types, or exceptions to cleavage rules like missed or semi-specific cleavages. All these decisions are made automatically based upon the sample treatment and experimental goals. Thus, in the search, we only specified trypsin digestion, Cys-carbamidomethylation, and Ubiquitinization modifications.

Two types of searches were made:

1. Of the MS/MS files (.wiff) obtained for each sample individually.
2. Of the MS/MS files of all the samples of the same group together.

The ProGroup™ algorithm reports two types of scores for each protein: Unused ProtScore and Total ProtScore. A critical false

discovery rate (FDR) of 1%, which corresponds to Unused Protscore from global FDR (fit), was used for filtering proteins[116].

Quantitation using the raw M1 intensity and spectrum count was performed using Scaffold (v5.3.0). Quantile normalizations were performed within the replicates of the same sample. A normalized spectrum count was obtained for each of the two conditions, with all replicates combined, and a Fisher's Exact Test. Only those spectra with >80% identification probability were counted. A QUANT Score ratio was calculated by dividing spectra in the mPIAS2-IP column by the control-IgG2a column. Proteins with a Score >1.5 were selected.

Data-independent acquisition (SWATH-MS) was also performed by analyzing first a library with equal proportion of every sample, followed by quantitative data comparing each sample with the library. This analysis was performed using Peak view 2.2 for retention time adjustment and areas extraction, and Marker view 1.3.1 for normalization and statistical analysis using a T-test. Significant proteins with an adjusted P-value < 0.1 were selected using the FunRich v 3.1.3 program.

Filtered proteins in SWATH or Spectrum Count analysis were curated for duplicate genes and analyzed for pathway enrichment.

**Data analysis.** Filtered proteins in SWATH or Spectrum Count analysis were curated for duplicate genes. Proteins above cut-off score (>1.5) or adjP value (<0.1) were analyzed in GPS-SUMO 2.0[117] to check for high-score SUMO consensus https://sumo.biocuckoo.cn/online_id.php. Proteins above scores were analyzed in Reactome v68[118,119] https://reactome.org/. For functional enrichment, we used String (v11.3)[120,121], https://string-db.org/. Selected proteins were analyzed for pathway enrichment using Venn diagrams with the tool InteractiVenn[122] https://www.interactivenn.net/ or FunRich v 3.1.3[123] http://funrich.org/index.html.

**Combined analysis.** 122 mitotic proteins common to both SWATH +Spectrum Count analyses in Proteomic Assay 1, were further combined with results from Proteomic assay 2.

**Proteomic assay 2.** Cells were transfected in a 6-well culture dish with a non-lethal amount of pEGFP-C1-hPIAS2b or vector pEGFP-C1 with Lipofectamine 3000 (Thermo). To maintain transfection efficiency at 2 µg/well of DNA, 1.6 µg of empty pcDNA3 plasmid was added to each well, along with 0.4 µg of the EGFP-C1 plasmid. Six hours later, thymidine was added to start the first block.

19 h later, the wells were washed and mitotic shake-off was performed. This was followed by first release, second thymidine block, and release up to completion of the DT-6h protocol (see above). Extracts were obtained at DT-6h using SUMO-lysis buffer [20 mM Tris-HCl pH 7.5, 150 mM NaCl, 0.5% Triton X-100, 1 mM MgCl2, 20 mM N-Ethylmaleimide, all from Sigma; and 1 tablet of protease inhibitors complete Mini EDTA-free (11836170001, Roche) and 500 Units/sample of Benzonase Nuclease (E1014, Sigma)]. Protein was measured with RC DC kit (5000122, BioRad).

A GFP-Trap was performed using extracts from each of the two conditions. Non-extract control beads were incubated in parallel with similar buffers and times. 25 µL of GFP-beads (GFP-Trap Chromotek kit) were washed with dilution buffer (kit included). After centrifugation, 2 mg of lysates was added in a final volume of 800 µL of lysis buffer and incubated overnight at 4 °C in an end-to-end mixer. The next day, samples were washed 3× (buffer included in kit). Bead-retaining proteins and Flow-through, together with non-extract control beads were subsequently processed for LC-MS/MS.

This experiment was performed three times; all samples were analyzed in a single batch.

LC-MS/MS was performed as in Proteomic assay 1 in each GFP-Trap (EGFP beads, or EGFP-PIAS2b beads) and control GFP-beads.

Technical replicates were added to biological replicates for the final data processing.

Data-dependent acquisition (DDA) was performed as in Proteomic Assay 1.

Quantitation using the raw M1 intensity and normalized spectrum count was also performed using Scaffold (v5.3.0), using a Fisher Exact test and biological replicates for each condition. Only spectra with >80% identification probability were counted. A QUANT Score was obtained by dividing the EGFP-PIAS2b column by the EGFP control column. Proteins with a Score >1.5 were selected.

Selected proteins from Protein Assay 2 were further combined with those from Protein Assay 1, to validate direct PIAS2b binding proteins.

**Proteomic assay 3.** Cells were transfected with PIAS2b-dsRNAi or scramble ns-dsRNAi, followed by DT-5h. Cells were lysed in lysis buffer with SUMO-protease inhibitors [50 mM Tris-HCl, pH 7.5, 150 mM NaCl, and 200 mM iodoacetamide, with 5 mM N-Ethylmaleimide (NEM) (04259) all from Sigma][124]. Protein was measured using the RC DC kit (BioRad).

Extracts from the two conditions were affinity purified using a SUMO-Trap. 100 µg of each lysate was subjected to SUMO-Qapture-T kit (BML-UW1000A, Enzo Life Sciences) following manufacturer's instructions. The kit contains a high-binding matrix, linked to a SIM peptide that has been shown to bind both SUMO1 and SUMO2/3 with high affinity. This kit includes in house validation, consisting in capture and detection of SUMOylated proteins from a HeLa lysate (S100 fraction). With high yield. It has been used previously in refs. 125,126. The SIM-affinity strategy has been reviewed in ref. 127.

From each sample, the eluted SUMO-protein conjugates (SUMOylated) and Unbound proteins (Flow-through or NON-SUMOylated) were subsequently processed for LC-MS/MS.

This experiment was performed three times; all samples were analyzed in a single batch.

LC-MS/MS was performed as in Proteomic assay 1 in the four fractions of each independent experiment (SUMO- or Flow-through/NON-SUMO). Technical replicates were added to biological replicates for the final data processing.

DDA was performed as in Assay 1. The identified proteins in each of the four fractions were compared using Venn diagrams.

Quantitation using the raw M1 intensity and spectrum count was also performed using Scaffold (v5.3.0) for the four conditions (1-SUMO-retained in ns-dsRNAi, 2-SUMO-retained in PIAS2B-dsRNAi, 3-Flow-through/NON-SUMOylated in ns-dsRNAi, and 4-Flow-through/NON-SUMOylated in PIAS2B-dsRNAi) with all biological replicates combined. Statistical analysis was performed with Fisher Exact Test. Only spectra with >80% identification probability were counted. A QUANT Score was obtained by dividing the 1-SUMO-retained-in-ns-dsRNAi column by the 2-SUMO-retained-in-PIAS2b-dsRNAi column. Proteins with SUMO Score >1.5 or <0.5 were selected.

After SUMO Scoring, proteins with SUMO Score >1.5 in Condition 1- that had no spectra in either Condition 2- or 4- were considered "missing", and thus, not possible to confirm about their SUMOylation status; they were eliminated from the final list of putative PIAS2b SUMOylation targets. Similarly, proteins with SUMO Score <0.5 in Condition 1 (e.g. >1.5 in Condition 2)- that were not identified in Condition 1- or 3- were considered "missing", and thus, not possible to confirm about their SUMOylation status; they were eliminated from the final list of proteins with increased SUMOylation in the absence of PIAS2b.

133 selected proteins with reduced SUMOylation in PIAS2b-dsRNAi were analyzed for enriched pathways using Reactome and String.

## Immunofluorescence confocal microscopy

In glass bottom, 96-well multiplates (655892, Greiner Sensoplate)[128] as recommended by The Human Protein Atlas consortium HPA, https://www.proteinatlas.org/ENSG00000078043-PIAS2/summary/antibody), 5000 or 10,000 cells were seeded for double-thymidine mitosis cell synchronization experiments alone or with dsRNAi transfection. At DT-6h or DT-5h, depending on whether the cells were transfected with dsRNAi, wells were carefully washed in serum-free medium, and fixed with commercial buffered formalin, followed by 0.1% Triton permeabilization at 4°. Wells were incubated with PBS or blocking solution, including mouse IgG2a when staining with anti-mouse PIAS2. The wells were then washed, followed by overnight incubation with primary antibody, washing, and secondary antibodies in PBS for one hour the next day. Mounting medium (17985-10, Fluoro-Gel, Electron Microscopy Sciences) with 2 mg/mL of DAPI (D9542, Sigma) was added to the wells and incubated for 10–20 min. The wells were then filled with 80% glycerol in sterile PBS.

Images were analyzed and captured using a scanning confocal microscope (Leica TC SP5-AOBS) with white laser (470–670 nm) and ultraviolet laser, through the HCX PL APOP 63× oil objective with numerical aperture (NA) 1.4 and LAS AF software (Leica Application Suite Advanced Fluorescence), 8 bits 1024×1024 format and 600 Hz speed, making serial planes every 0.20–0.35 microns in a sequential bidirectional way, with Line-average = 1, Frame-Average = 1, and Frame-accumulation = 2, obtaining a Z-stack around 2–3.5 µm. The UV and transmitted light channels were collected. Channel conditions were as follows: Channel 1 Ex Laser 405 nm UV, PMT1 414–479 nm Gain: 516, with active Transmission Gain at 222; Channel 2 White laser Ex 496 nm, HyD3 510–550 nm Gain: 22; Channel 3 White laser Ex 552 nm, HyD3 560–601 nm Gain: 31; Channel 4 White laser Ex 647 nm, HyD3 658-720 nm Gain: 23. For super-resolution microscopy, Airy was reduced to <1, with pinhole around 75 in the white laser channels, and 80 in the UV laser, resulting in a Voxel of 0.049 microns; optical zoom was kept at 5–6×. Laser power was reduced at less than 17% in both lasers.

LIF files were analyzed by a pipeline of deconvolution in Imaris v9.0.8. For two markers colocalization this was followed by a Fiji pipeline with the Coloc2 plugin to obtain the Pearson Colocalization Coefficient, (PCC) at the maximal intensity. For three markers, volumetric 3D colocalization was analyzed in Imaris for 3 markers using Coloc to measure thresholded Colocalized Voxels. 3D volume models were also obtained in Imaris.

## Proteasome activity assay

Proteasome activity was measured using Cell-based Proteasome-Glo™ Chymotrypsin-Like Assay (Promega, G8660) following manufacturer's instructions. Briefly, cells were seeded in a multiplate 96 well and after transfection for 24 h, cells were incubated in presence/absence of the proteasome inhibitor MG-132 (Sigma-Aldrich, 474790) for two hours at 37 °C and 5% CO2. After that, cells were incubated with the luminogenic substrate Suc-LLVY-Glo™ (Promega, G856C) for 30 min at room temperature and is measured in a white plate in a luminometer (Multimode Mithras LB 940, Berthold Technologies).

## Primary culture and cell lines aneuploidy score

In 997 cell lines, data about ploidy, aneuploidy score and aneuploidy group classification into "Highly", "Intermediate" and "Near-Euploid" categories, were extracted from Cohen-Sharir et al.[73]. Information about copy number gain and losses in each chromosome and in each cell line were extracted as well. Aneuploidy score and ploidy were used to plotted density graphs using data from all cell lines. Then, for eight cell lines of interest, aneuploidy score and ploidy values were marked in the plots to highlight their position in the global distribution. A hierarchical clustering using information about chromosome arms gain and losses in the eight cell lines was computed. A dendrogram was drawn to visualize the distance tree for cell lines and copy number events. The eight cell lines were also clustered using the ploidy value. Samples distances were computed via the R function dist, with Euclidean distance. Heatmaps were plotted for visualization purposes.

## Orthotopic patient-derived xenografts (oPDXs)

Nine 7-week-old NOD-SCID female mice (NOD.CB-17-Prkdc^SCID/Rj) were obtained from Janvier Labs. Twenty 6-week-old female NSG mice (NOD.Cg-Prkdc^SCID Il2rgt^m1Wjl/SzJ) were obtained from Charles River. Mice were housed with an artificial 12 h light/12 h dark cycle, under controlled temperature (22–24 °C) and humidity conditions (40%) and allowed to free access to standard laboratory chow and tap water. When 8-week-age, mice were orthotopically implanted into the right lobe of the thyroid gland by neck surgery using a 3-cm-long, blunt needle Hamilton syringe (100 µL/30 mm/PST3; 1710N, Hamilton). Kinetic analysis of the luciferase activity was performed once a week using an in vivo imaging system (IVIS Spectrum imaging system, Perkin Elmer). Thereafter, 300 mg/kg luciferin (122799, XenoLight D-Luciferin, Perkin Elmer) was injected intraperitoneally. Tumor growth was monitored weekly for 3–7 weeks before starting treatment.

Five preclinical trials (PCTs) were designed with a common primary purpose of intravenous treatment with either placebo (ns-dsRNAi) or PIAS2b-dsRNAi. T-UC3 cells from patient 2 did not grow in NOD-SCID or in NSG mice, and those two trials were therefore closed (Supplementary Fig. 8e).

Our orthotopic ATC Patient-derived Xenograft model has a time window defined by the approvals of the Ethical Committee. The Endpoint was 9–10 weeks from the orthotopic implantation of the tumor, with daily observation starting at week 9 checking for endpoints such as >20% weight loss or being moribund. The primary outcome was tumor growth/volume, which was monitored in the IVIS. A secondary purpose was established for each PCT in order to reduce the number of mice used, following the 3 R rule:

- PCT1: T-UC1 cells from Patient 1, in NOD-SCID mice, growth for 5 weeks. Primary Purpose: Treatment. Secondary Purpose: to establish toxicity for i.v. injection twice a week for 3 weeks.
- PCT2: T-UC7 from patient 3, in NSG mice, growth for 3 weeks. Primary purpose: Treatment. Secondary purpose: to establish toxicity for i.v. injection twice a week for 7 weeks.
- PCT3: approval of an extension to week 11 by the Ethical Committee. T-UC7 from patient 3, in NSG mice, growth for 7 weeks. Primary purpose: Treatment in advanced tumors –Compassionate use-. Secondary purpose: to observe relapse when Off treatment for 1 week.

Five mice per group were injected with ns-dsRNAi, or *PIAS2b* dsRNAi-1, in 200 uL of AteloGene (Koken, Cosmobio) mixed 1:1 with 40 µM dsRNAi per mouse[30,63]. Longitudinal curves were plotted by connecting the measurements from individual mice and the smooth (LOWESS) group. At the end-point, mice were deeply anesthetized, skin dissected and a direct IVIS image was taken. Tumors were dissected and measured, following by buffered formalin fixation. The pathologist cut the tissue blocks for FFPE. Immunohistochemistry of tumor markers and hematoxylin-eosin staining was according to routine clinical pathology procedures. Immunohistochemistry of PSMC5 and pHH3[65] was similarly performed. Staining quantification was performed using deep learning algorithms in CellSens standard software (version 4.1, Olympus) after making a training protocol in different mice sections (at least 10) until obtain a quality indicator by a similarity score >0.7.

## Statistics & reproducibility

Sample size calculation was obtained the calculator of Prof. Weyne W LaMorte (Boston University, https://sphweb.bumc.bu.edu/otlt/MPH-

Modules/Excel/Excel6.html) using previous known data. For in vitro experiments Sample Size was calculated as follows: The Standard Deviation (SD) of proliferation assays in cell lines and primary cultures was well established in our laboratory, and less than 10% with $n = 8$ technical replicates[22,23]; with this, expecting a difference in growth >35%, with a power of 90%, and a two-sided alpha level of 0.05, the sample size (minimum of independent experiments) was 3. For Primary Cultures, the $n = 3$ independent experiment was performed with only 3 technical replicates in each, due to the lower number of cells available.

For animal experiments, a Standard Deviation (SD) of the Luciferase signal at the IVIS was considered to be higher, at 25%. The difference in growth appreciated as different was considered >35%. Thus sample size was calculated for a power of 80% and alpha of two-sided 0.05, and it was $n = 5$ animals per group in each Pre-Clinical Trial (PCT).

Regarding randomization, or In vitro experiments, the distribution of the different sample groups is carried out by seeding the cells in columns inside the multiwells, while the different treatments are added vertically; or vice-versa. For the microscopy quantification, at least 3 fields were randomly photographed from each of the wells that form the groups of samples. Mice were randomized to treatment groups. Blinding was not possible for in vitro experiments but the whole manuscript was performed as contrary to the results presented, as follows. The initial data, based on 2D proteomics and in TCGA ATLAS on Differentiated Papillary Thyroid Carcinomas (PTC), suggested that the important effect would be in PTC. We were working and collecting data in all types of primary cultures of thyroid tumors and cell lines with this hypothesis. It was not until much later, that we were able to understand that the effect of PIAS2b downregulation was in Anaplastic Thyroid Cells.

Statistical analysis was performed using GraphPad Prism 8.0.1 (San Diego, USA), SPSS Statistics 20 (IBM, USA) and Corel Draw Graphic Suite 2017. The samples were included or excluded from the analysis by prior identification of the outliers using the GraphPad Prism 8.0.1 software (San Diego, USA). In animals no data were excluded from the analysis.

Results are represented by means ± SEM. Indicated in each section the number of independent experiments or samples. For each group of results, a normality test (Shapiro-Wilk) was first performed. All comparisons between groups of data that followed normal distribution were made with $t$-test between two groups, or with ANOVA between more than two groups. For non-normal distributions, the Mann–Whitney test was used between two groups, and the Kruskal–Wallis test was used between more than two groups. Exact $p$ values are shown in the Figures.

## Reporting summary

Further information on research design is available in the Nature Portfolio Reporting Summary linked to this article.

## Data availability

All data generated or analyzed during this study (and its supplementary information files) are provided as Source Data with this paper. The mass spectrometry proteomics raw data have been deposited to the ProteomeXchange Consortium via the PRIDE[129] partner repository with the dataset identifier PXD044110. https://www.ebi.ac.uk/pride/ LIF files with raw confocal conditions are provided upon request to C.V.A. The TIROCHUS collection has some restrictions regarding availability of tissue and cell cultures following a specific signed consent by the patients; any reasonable scientific collaboration can be applied to the Clinical Responsible (Dr. Jose M Cameselle-Teijeiro) and will be reviewed by the Hospital Clinical Committee. All remaining data is present in the Article, Supplementary and source data files. Source data are provided with this paper.

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

## Acknowledgements

We thank Andrew Fire (Stanford University School of Medicine, California) for helpful reading and scientific suggestions; Stephen P. Jackson (Wellcome Trust and Cancer Research, Cambridge, UK) and Alfred CO Vertegaal (Leiden University Medical Center, The Netherlands) for plasmids; Carmen Rivas (Group of Virus and Cancer, CIMUS) for the His.Tag antibody and debates, and Xose Bustelo (CIC, Salamanca), Roger Gomis (IRB, Barcelona), Marisol Soengas (CNIO, Madrid), Anna Santamaria and Joan Seoane (VHIO, Barcelona), Guadalupe Sabio (CNIC, Madrid) and Carlos Dieguez (CIMUS Head, Santiago de Compostela) for helpful discussions. We thank Ana Senra (CIMUS), Paula Vieiro and Magali Piso (IDIS) for technical help, and Veronica A Raker for scientific editing. Our deep appreciation for the patients' generosity.

## Author contributions

J.S.R., M.C., J.M.C-T., and C.V.A. developed the original hypothesis and designed experiments. J.S.R., M.C., R.S-P. and R.G-P. performed experiments and/or analyzed data. J.S.R., S.B. and R.G-P. performed proteomic analysis. M.S.F., S.P.R. and L.F. collected primary tissues, established the cultures, and/or analyzed DNA and RNA; J.S.R. and T. S. performed cell sorting; J.M.C-T., M.L. and R.N. provided critical reagents and/or samples; M.N.B.F., L.F. and J.M.C-T collected patients, samples and clinical data of thyroid cancer patients; L.F., J.C.M-T. and C.V.A. wrote the manuscript; J.S.R., M.C., M. N.B.F., L.F., R.S-P., R.G-P., S.P.R., M.S.F., T.S., M.L. and R.N. revised and edited the manuscript. L.F., J.C.M-T. and C.V.A. were responsible for ethical procedures and financial support.

## Funding

This work was supported by Agencia Estatal de Investigación (AEI, Spain), co-funded by the European Union, projects' codes BFU2013-46109-R, BFU2016-76973-R, PID2019-110437RB-I00, PDC2021-121621-I00 and PID2022-140149OB-I00 to C.V.A. This work has received financial support from the Xunta de Galicia (Centro singular de investigación de Galicia accreditation 2019-2022) and the European Union (European Regional Development Fund - ERDF). J.M.C.-T. is supported by grant no. ISCIII-PI23/00722, and PI19/01316 from Instituto de Salud Carlos III, State Research Agency (Spain), co-funded by the European Union. L.F. is funded by Italian Ministry of Health - Ricerca Corrente. T.S. is recipient of a research contract from the Miguel Servet Program from the Instituto de Salud Carlos III, Ref. CPII17/00027, co-funded by the European Union.

## Competing interests

The authors declare no competing interests regarding the science behind the data. A patent is being issued. Details: Patent number: Oficina Española de Patentes y Marcas (OEPM), N° Publicación WO2021028610 A1. Title: "COMPUESTOS Y MÉTODOS PARA EL TRATAMIENTO DEL CÁNCER", PCT/ES2020/070512 Investigators: J.S.R. (33%), J.M.C.T. (33%), C.V.A. (34%) Institutions: USC (67%) + SERGAS (33%).

## Additional information

[1]Neoplasia & Endocrine Differentiation, Centro de Investigación en Medicina Molecular y Enfermedades Crónicas (CIMUS), University of Santiago de Compostela (USC), Instituto de Investigación Sanitaria (IDIS), Santiago de Compostela, Spain. [2]Department of Proteomics, Complejo Hospitalario Universitario de Santiago de Compostela (CHUS), Servicio Galego de Saúde (SERGAS), Instituto de Investigación Sanitaria de Santiago (IDIS), University of Santiago de Compostela (USC), Santiago de Compostela, Spain. [3]Department of NeuroAging Group – Clinical Neurosciences Research Laboratory (LINC), Complejo Hospitalario Universitario de Santiago de Compostela (CHUS), Servicio Galego de Saúde (SERGAS), Instituto de Investigación Sanitaria de Santiago (IDIS), University of Santiago de Compostela (USC), Santiago de Compostela, Spain. [4]Centro de Investigación Biomédica en Red en Enfermedades Neurodegenerativas, Instituto de Salud Carlos III, 28029 Madrid, Spain. [5]University Hospital Lozano Blesa, Institute for Health Research Aragon (IISA), ARAID Foundation, Aragon Government and CIBERESP, Zaragoza, Spain. [6]Centro de Investigación Biomédica en Red en Epidemiología y Salud Pública, Instituto de Salud Carlos III, 28029 Madrid, Spain. [7]Cell Dynamics and Signaling Department, Andalusian Center for Molecular Biology and Regenerative Medicine, Universidad de Sevilla - CSIC - Universidad Pablo de Olavide-Junta de Andalucía, 41092 Sevilla, Spain. [8]Department of Cell Biology, Faculty of Biology, University of Sevilla, 41012 Sevilla, Spain. [9]Department of Surgery, Complejo Hospitalario Universitario de Santiago de Compostela (CHUS), Servicio Galego de Saúde (SERGAS), Instituto de Investigación Sanitaria de Santiago (IDIS), University of Santiago de Compostela (USC), Santiago de Compostela, Spain. [10]Molecular Metabolism, Centro de Investigación en Medicina Molecular y Enfermedades Crónicas (CIMUS), University of Santiago de Compostela (USC), Instituto de Investigación Sanitaria (IDIS), Santiago de Compostela, Spain. [11]NeurObesity, Centro de Investigación en Medicina Molecular y Enfermedades Crónicas (CIMUS), University of Santiago de Compostela (USC), Instituto de Investigación Sanitaria (IDIS), Santiago de Compostela, Spain. [12]Department of Endocrine and Metabolic Diseases and Laboratory of Endocrine and Metabolic Research, Istituto Auxologico Italiano, Istituto Di Ricovero e Cura a Carattere Scientifico (IRCCS); Department of Pathophysiology and Transplantation, University of Milan, Milan, Italy. [13]Department of Pathology, Complejo Hospitalario Universitario de Santiago de Compostela (CHUS), Servicio Galego de Saúde (SERGAS), Instituto de Investigación Sanitaria de Santiago (IDIS), University of Santiago de Compostela (USC), Santiago de Compostela, Spain. [14]Present address: Dana Farber Cancer Institute, Boston, MA, USA. [15]These authors contributed equally: Joana S. Rodrigues, Miguel Chenlo. [16]These authors jointly supervised this work: José Manuel Cameselle-Teijeiro, Clara V. Alvarez.
✉e-mail: josemanuel.cameselle@usc.es; clara.alvarez@usc.es

