## [Peer Review File · Nature Communications]

Reviewers' Comments:

Reviewer #1:

Remarks to the Author:

Rodrigues et al. investigate the role of E3 SUMO ligase PIAS2 in thyroid carcinomas, and the impact of its silencing through RNA interference. This is an extensive and thorough study that looks at this role from a multitude of angles, including different dosage, in vitro within a range of cell lines and in vivo via a patient-derived mouse model.

Given my limited expertise with RNA therapies, I will mostly focus on the proteomics part.

The manuscript is mostly well written and contains a huge load of information from all types of experiments, which I consider to be at the limit of what a manuscript can maximally contain. On the other hand, showing the clinical potential of such a treatment probably justifies such detail. What I definitely dislike is the crowdedness of the figures. Particularly Fig. 4 is an exaggeration, containing figure panels A-V, and a figure caption spanning 1.5 pages. This makes it very hard for a reader.

I have mostly minor comments on the content, but some particular and major requests on the proteomics part.

1) When it comes to the method description of the LC-MS experiment, there is not much information in the supplementary material. There is more to find in the PRIDE repository but it is very scattered and unorganized there. I strongly suggest to be more detailed in the supplementary material describing the method, and become clearer in PRIDE.

Several questions and comments:

a) Did the authors use the same mass range for SUMOylated and non-SUMOylated peptides? Given that SUMOylation is quite heavy, isn't it better to use different mass ranges? Did the database search of the non-retained proteins include searching for SUMOylations, to be sure that the retaining setup worked?

b) I did not find any information neither about used mass tolerances in the database search nor about the allowed fixed and variable modifications. These need to be specified.

c) The number of identified proteins is quite low when considering a proteome from a cell line. However, I can read in the text on PRIDE that the cell extracts were immunoprecipitated with an anti-PIAS2 antibody. This I could not read out from the text in the manuscript! If this is the case, then I understand the low numbers. Otherwise the numbers would not make sense.

d) Why did the authors not quantify the proteins? This could have given a much better idea of the underlying changes, particularly of the proteins found in both sample types.

e) The description of the data analysis on PRIDE is too long and reads like text fragments pasted arbitrarily. The post-analysis of pathways does not need to be put into PRIDE as it is a repository for the raw data and identified proteins.

2) Did proteomics really show proteasome and cytoskeleton as direct targets of PIAS2b SUMOylation, or only as consequence? They could also be indirect targets, as the experiment did not investigate the actual interactions. Pathway analysis results do not justify such a statement. The manuscript also describes an experiment though that shows colocalization through microscopy (not proteomics).

3)

There is some really small text in the graphical abstract. Consider increasing the font size.

Beginning of page 7: I would change "searched through 2D proteomics" to "used 2D gel

electrophoresis".

Page 9, paragraph starting with "We could not attribute ..." is very difficult to understand through the exhaustive use of semicolons and commas. Please rephrase.

Reviewer #2:

Remarks to the Author:

The authors describe a role for PIAS2 in anaplastic thyroid carcinomas, a notorious type of cancer with very high mortality. They propose PIAS2 as potential drug target and study potential mechanisms of action.

Major points

1. Important roles for SUMO in mitosis and mitotic catastrophe are well known. The literature on this topic should be cited in more detail in the introduction. Missing key references include Zhang et al. 2008 Mol Cell; Kessler et al. 2012 Science.
2. The siRNA approach employed is outdated. CRISPR-Cas technology is much more robust and can provide clean PIAS2 knockout cells. The use of CRISPR-Cas technology is therefore recommended.
3. The resolution of many microscopy images in the paper is poor and should be considerably improved. This should be done using state-of-the-art high resolution microscopy. Moreover, many images are too squeezed.
4. I am not convinced that the PIAS2 antibodies used for the study are uniquely recognizing PIAS2. The cropped immunoblots already show a smeary pattern, thus the antibodies could recognize other proteins besides PIAS2. Full-size blots should be provided. Using only properly validated antibodies is vital.
5. The authors study interactors of endogenous PIAS2. Whereas this is a good start, the quality of antibodies, yield and purity of endogenous approaches are generally rather modest, see point 4. Moreover, it would be good to complement the approach by identifying interactors of stably expressed EGFP-PIAS2 at close to endogenous levels using GFP nanotraps. The GFP nanobodies are exceptional tools for interaction studies and provide optimal yield and purity.
6. The authors show that exogenous EGFP-PIAS2b (Figure 3G) and endogenous PIAS2 (Figure 3F) are localized at the mitotic spindle. Here, the criticism on the PIAS2 antibodies applies as well, see point 4. If this involves sumoylation of substrates, then it is expected that sumoylated proteins would also accumulate at the mitotic spindle. Can the authors verify whether sumoylated proteins accumulate at the mitotic spindle? Proper quantitative microscopy would be appreciated, where at least a hundred mitotic cells are imaged and localization of PIAS2 and SUMO are studied.
7. Key to understanding the role of PIAS2 in mitosis is to identify the proteins that are sumoylated by PIAS2 in mitosis. The authors use the SUMO-Qapture-T kit for enrichment of sumoylated proteins and mass spectrometry. This kit is employing a matrix containing SUMO Interaction Motifs (SIMs), which are short motifs of three large hydrophobic residues that interact with SUMO. The interaction between SUMO and SIMs is notoriously weak. Proper validation of the kit for use with cellular extracts is missing both in the paper and on the website of the manufacturer. Yield and purity of samples are expected to be very modest. Much more robust methodology is available in the field for proper purification and analysis of SUMO-modified proteomes. Quantitative site-specific methodology should be employed to distinguish between sumoylated proteins and background binders. A detailed overview of methodology can be found in Li et al. 2021 Nature Reviews Methods Primers.
8. As indicated by the authors, only one orthotopic PDX from one patient grew in the immune-deficient mouse model. The authors tried to compensate this by increasing the number of mice.

This is however a poor solution, a minimum of two independent orthotopic PDX models from two different patients should be used.

9. The part of the manuscript describing regulation of the proteasome by sumoylation is interesting, but still in a preliminary state. It would be better to leave it out here and properly develop this into a separate project. The PSMC5 antibody used in the study is poor, with a smeary pattern observed, therefore I am not convinced about the validity of the results.

Reviewer #3:

Remarks to the Author:

The work by Rodrigues and colleagues investigated the potential contribution of PIAS2 to the oncogenic program of Anaplastic Thyroid Cancer. In spite of the relevance of the topic and of the experimental work described, I believe that this work does not significantly contribute to progress in this context, neither in terms of understanding the molecular mechanism driving ATC progression nor in terms of translational impact.

Reviewer #4:

Remarks to the Author:

In their paper, Rodriguez et al. investigate the role of the SUMO ligase PIAS2 in Thyroid Carcinomas. They identify PIAS2 as a gene with altered expression on protein and RNA level in various thyroid carcinoma cell lines, including patient-derived primary tissues. They demonstrate that downregulation of the PIAS2b isoform inhibits growth of anaplastic thyroid cancer (ATC) cell lines and also other non-thyroid cancer cells that display similar characteristics to ATC cells. The authors also investigate the anti-cancer effects of PIAS2b downregulation in orthotopic patient-derived xenografts of anaplastic thyroid cancers in vivo. They continue to show that downregulation of PIAS2b induces mitotic catastrophe in ATC cells and that PIAS2b is a mitotic spindle-associated protein. Using a combination of proteomics, pulldowns and immunofluorescence they identify a number of cell-cycle regulators, spindle associated proteins and proteasome subunits as PIAS2b-interacting proteins. The authors propose that PIAS2b-dependent SUMOylation controls correct spindle formation, proper chromosome-microtubule attachment at the kinetochores and the activity of proteasome at mitotic onset.

This is a very large study with very diverse approaches and a large amount of generated data. The findings are very interesting but the paper has two main caveats in my opinion; First, all the results are based on one dsRNAi construct, so the off-target effects cannot be ruled out. The authors tried indeed to address this issue by using other constructs, but without success. Second, as it is written the paper, lacks focus and tries to address too many issues at the same time, sometimes lacking depth, especially in what concerns the speculated molecular mechanisms. Certainly, find the data interesting and I do not doubt that PIAS2 has important roles suggested by the authors. However, in my opinion it would be much more useful to separate the data into different papers, each with a different focus and more depth.

Major:

1. It is very disturbing that all conclusions of the paper are based on the use of one dsRNAi construct. I acknowledge the effort of the authors to verify their results using other constructs and methods. However, since these failed to work the paper lacks this important control. It remains to confirm the results using more robust methods, ideally generating a conditional PIAS2b KO (for example degron-based), or RNAi insensitive constructs. Doesn't any working RNAi for PIAS2 exist in the literature? Also, the paper implies that the Ser rich stretch of the PIAS2b and the ligase activity of PIAS2 are required for its function. Since the FLAG-PIAS2 can rescue the dsRNAi downregulation, it could be examined whether the ligase activity of PIAS2 or the C-terminal rich domain is required for its function. However, there is no reference on the data of dose-response titration and the 20-times lower amount of FLAG-hPIAS2b that allows cell growth. Does expression

of PIAS2a rescue as well?

2. The authors should be more careful when concluding (or proposing) molecular mechanisms. For example (discussion): "This inability correlates with less SUMOylation of proteasomal proteins and excess proteasome activation at mitotic onset, thereby leading to failure in establishing a correct spindle and proper chromosome-microtubule attachment at the kinetochores ". None of this has been addressed in the paper. Showing colocalization with the spindle and associated proteins and cell cycle effects is very far from a demonstration of chromosome attachment defects. Moreover the SUMOylation status has not been investigated for any of the targets (binding to a SIM column is not demonstration of SUMOylation status as the authors suggest).

3. The paper uses 2 antibodies to produce the results presented. While the antibodies seem to work, there is still some information lacking. Which were the 7 commercial antibodies tested and what are the results? For example, do they confirm the western blot results (for the Abs that function in western)? Does the detection of the PIAS2b depend on the Serine rich stretch? This could be also tested. Are mPIAS2 and rPIAS2b the commercial antibodies D12 and polyclonal shown in the Materials and methods? This is not clear.

4. The localization of EGFP-PIAS2b should be better shown. Are there better images or better time-lapse showing its localization on the spindle? How often is this observed? Can the localization of PIAS2 shown in Fig. 3O be recapitulated by EGFP-PIAS2b? What is the localization of EGFP-PIAS2a? The authors should explain better their rationale that non phosphorylated PIAS2b is on the spindle.

5. The paper should be edited again to eliminate errors that cause confusion. For example, there is confusion on what is detected on Pg 7: "The antibody mPIAS2 is better at detecting endogenous PIAS2b p75, and rPIAS2b, is better at detecting PIAS2b p95. PIAS2a was not detected". This is confusing and suggests that p75 also PIAS2b. Do they authors want to say that: mPIAS2 is better at detecting endogenous PIAS2a p75 and rPIAS2b is better at detecting PIAS2b p95, while PIAS2a is not detected by rPIAS2b" ? This is suggested by Legend Fig.1: "Yellow bars indicate the epitopes of the two antibodies; while mPIAS2 detects both isoforms, rPIAS2 detects only PIAS2b". Also the authors keep exchanging between rPIAS2 and rPIAS2b for the same antibody. I suggest that they mark on the side of every blot the protein that is detected and not only the antibody. Depending on what is detected by each antibody, the fig S2G is not convincing. I cannot really see a difference in the protein levels of PIAS2b detected by rPIAS2b, but only for the PIAS2a. Are the antibody annotations switched? It is important that the data on the antibodies used in the study become clearer, since the major part of the findings is based on their use.

6. The part of the SUMO-retention experiment should be better explained, it is too confusing. The SUMOtrap column (it's a SIM column) should be referred in the text already. If the aim of the experiment is to identify proteins that are differentially SUMOylated in the absence of PIAS2b, as I suspect, it would be perhaps better to focus on this result (31 proteins).

7. Loading controls are missing in 4H: is the protein amount (of the material retained on the column) loaded for western blotting the same between control and dsRNAi? Fig. 4J: Why is the rPIAS2b recognizing a protein at 75kDa and not at 95kDa in 8305C cells? (in contrast to Fig. S1F).

8. The authors should be careful when drawing conclusions that a protein is SUMOylated or not: Figure 4E). "SUMOylated TUBB3 was associated to PIAS2 (Fig 3J and Table S3) and lost its SUMOylation in PIAS2b-dsRNAi extracts (Figure 4E)". It has not been shown that SUMOylated TUBB3 associates with PIAS2; also just the loss of association with the SIM column is shown and not the loss of SUMOylation.

9. Fig. 4L: Why isn't the effect of PIAS2 RNAi in asynchronous cells recapitulated in any other phase after synchronization?

10. Fig. 4N: What is the cell cycle distribution of cells after the PIAS2 RNAi and the 2T block compared to the control RNAi? Since PIAS2 RNAi affects cell cycle progression, any differences in these experiments can be due to the differences in cell cycle stage of the cells. There is no cell

cycle stage marker (cyclin etc) included in any of the experiments.

Minor:

1. Validation of the proteins that co-IP with PIAS2 should be better explained in the legend. The amounts immunoprecipitated PIAS2 should be also shown like in Fig. 3K

2. In Fig, S1, what is E (endogenous?) and F (Flag?) and why is the GAPDH blot the same for both blots, have they been stripped?

In the S1H legend Is the rPIAS2 antibody should read PIAS2b

Reviewer #5:

Remarks to the Author:

Rodrigues JS et al present an interesting study demonstrating that silencing of PIAS2 especially isoform 2B induces mitotic catastrophe in ATC cells in vitro. The authors validate their findings in multiple human cell lines and primary thyroid cancer cultures. Preliminary in vivo studies show a potential effect of silencing PIAS2b in orthotopic ATC xenograft models. The authors also show in vitro effects of PIAS2b silencing in cell lines of non-thyroid cancer types. Together this study presents relevant and important findings that are convincing.

However, the authors should consider significantly cutting the text of the manuscript. The most important results are summarized in figure 3 and 4. The text corresponding to the results shown in figure 1 and 2 should be shortened considerably, e.g. the section 'Efficiency of RNAi methods and effect of PIAS2b overexpression' is mostly redundant and can be incorporated in the previous section. Another example is the result section describing the in vivo xenograft studies. Moreover, the discussion could be significantly shortened by reducing the repetition of previously described results.

Specific comments

Fig 1C: The thyroid depiction with normal tissue und multiple thyroid tumors is misleading since it appears that all the different tissue samples have been generated from one specimen. Concerning the mRNA data, do you have an explanation why PIAS2b is highest expressed in PTC and lowest in ATC/PDTC tissues compared to normal thyroid tissue but in your consecutive studies silencing PIAS2b is the anti-tumor effect?

Fig 1G-I: I would suggest highlighting the driver mutation in the text box describing the genotype of the individual cell lines and confirm the mentioned genotypes. For FTC-238 the driver is PTEN p.130*, which is not mentioned. Please verify if the presence of a BRAF mutation is correct as this is not described for this cell line in the previous literature. For 8305C and BHT-101 the driver is BRAFV600E.

Fig 2G: Please clarify what the numbers under the X-Axis signify.

Fig 4E: please provide the list of proteins.

Fig 5 and in vivo xenograft studies: (i) Could you specify the endpoint of your study. (ii) The title of the section is in comparison to all other sections very vague, consider specifying. (iii) Your outcome measure is repeated IVIS assessments. For a proof-of-concept study to determine differences in tumor burden in general is fine, however to classify the outcome by RECIST or Choi criteria seems an overinterpretation of the data. For Choi or RECIST measurements actual scans should be provided. However, showing the difference in tumor volume (fig 5F) seems sufficient to show a general effect of PIAS2b silencing on ATC growth.

Fig. 6: While it is interesting that PIAS2b silencing does not only affect ATC but also pancreatic ductal adenocarcinoma, anaplastic lung carcinoma, and undifferentiated gastric carcinoma cell proliferation, the data shown might not warrant an individual figure.

Point-by-point Response to Reviewers' Comments

Reviewer #1 proteomics

Rodrigues et al. investigate the role of E3 SUMO ligase PIAS2 in thyroid carcinomas, and the impact of its silencing through RNA interference. This is an extensive and thorough study that looks at this role from a multitude of angles, including different dosage, in vitro within a range of cell lines and in vivo via a patient-derived mouse model. Given my limited expertise with RNA therapies, I will mostly focus on the proteomics part.

The manuscript is mostly well written and contains a huge load of information from all types of experiments, which I consider to be at the limit of what a manuscript can maximally contain. On the other hand, showing the clinical potential of such a treatment probably justifies such detail.

Response: We thank the Reviewer#1 for the positive view of our manuscript. We also thank for his/her comments which we believe have improved the quality of our study. Please, note that throughout this Rebuttal all the Figures numbers correspond to the new version of the manuscript.

What I definitely dislike is the crowdedness of the figures. Particularly Fig. 4 is an exaggeration, containing figure panels A-V, and a figure caption spanning 1.5 pages. This makes is very hard for a reader.

Response: We understand your concern about the crowdedness of Figure 4. We have made every effort to reduce the number of panels in this figure by moving some data to Supplementary Figure 4. However, we had to balance this with the need to include new requested data that validates SUMOylation of PIAS2b targets.

The current Figure 4 includes validation of the two chosen candidates of being direct PIAS2b SUMOylation targets at mitosis: PSMC5 (as representative of the proteasome) and TUBB3 (as representative of the microtubule cytoskeleton). This was a request of the Editor and other reviewers. The validation includes affinity purification of EGFP-PSMC5 with a GFP-Trap, and immunoprecipitation of TUBB3 with a monoclonal antibody; both followed by western blotting for SUMO1 (and SUMO2, no different bands obtained, and thus not shown), GFP, PSMC5 or TUBB3, and PIAS2b. The validation also includes colocalization through quantitative super-resolution microscopy of the three markers at different mitotic phases and its quantification after 3D surface rendering with Imaris.

We believe that we have achieved a reasonable balance between reducing the density of Figure 4 and including the necessary new data. We hope that you will agree.

Thank you for your feedback

I have mostly minor comments on the content, but some particular and major requests on the proteomics part.

- 1) When it comes to the method description of the LC-MS experiment, there is not much information in the supplementary material.

- 2) There is more to find in the PRIDE repository but it is very scattered and unorganized there. I strongly suggest to be more detailed in the supplementary material describing the method, and become clearer in PRIDE.

Response. We thank the Reviewer for this comment. We agree that it is important to make the manuscript easier to follow, and we have made the following changes to the nomenclature of the proteomic assays, that include in this version a new third proteomic assay:

Proteomic Assay 1 (immunoprecipitation anti-PIAS2)
Proteomic Assay 2 (GFP-Trap)
Proteomic Assay 3 (SUMO-affinity Trap)

We have described the details of each assay in the Supplementary Methods section and in the PRIDE repository. The PRIDE repository has a character limitation, so we had to omit some non-essential information. However, we have provided all the detailed information in the Supplementary Methods section.

We hope that these changes will make the manuscript easier to follow.

Several questions and comments:

- a) Did the authors use the same mass range for SUMOylated and non-SUMOylated peptides? Given that SUMOylation is quite heavy, isn't it better to use different mass ranges? Did the database search of the non-retained proteins include searching for SUMOylations, to be sure that the retaining setup worked?

Response. We thank the Reviewer for these excellent comments.

- a) The SUMO-trap that we used is a matrix containing three high affinity SUMO-binding domains in tandem, based on the natural SIM sequence but modified for higher affinity. This allows it to "fish" for SUMO proteins with high affinity, along with the proteins that are covalently attached to them. After thorough washing, the proteins are eluted in urea buffer and analyzed by LC-MS/MS, following our protocol. Our approach cannot distinguish between SUMOylated and un-SUMOylated peptides, but is addressed to distinguish between SUMOylated and un-SUMOylated proteins.

The mass ranges that we used were our usual range of 300-1400, which we raised to 1800 to include post-translational modifications (PTMs) in the search. It is important to note that after enzymatic digestion with trypsin, only small chemical modifications (such as phosphorylation, acetylation, and methylation) can be reliably identified by LC-MS/MS. This includes tryptic digestion of a ubiquitinated protein, which leaves behind a Gly-Gly (mass shift of +114.042927, monoisotopic) residue (-K-ε-GG or GlyGly motif) bound to the lysine of the modified protein platform.

On the other hand, endogenous SUMO tryptic remnants are too large, which makes their determination by LC/MS/MS incredibly challenging, even using the most recent and sensitive equipment. SUMO can be inferred from SUMO peptides, but not the wild-type SUMO modified residue. Alternative workflows would be required to determine SUMOylated/un-SUMOylated peptides in a site-specific manner. Given the limited nature of our samples, this would be even more challenging, requiring the genetic modification of anaplastic cells with engineered mod-SUMOs, which carries the risk of artificially altering a model relevant to anaplastic carcinoma patients. For this reason, we believe that this is still beyond the scope of this manuscript.

We hope this clarifies our approach and the limitations of our study. We appreciate the Reviewer's thoughtful comments and suggestions.

b) I did not find any information neither about used mass tolerances in the database search nor about the allowed fixed and variable modifications. These need to be specified.

Response. The reviewer asks for a description of the modifications. Regarding the ProteinPilot software, we used a software technology for the identification of peptides from tandem mass spectra called the Paragon™ Algorithm. Using this software, it is not necessary to specify mass tolerances, individual modifications to search for, expected fragment ion types, or exceptions to cleavage rules like missed or semi-specific cleavages. All these decisions are made automatically based upon the sample treatment and experimental goals. Reference: Shilov et al., 2007: <https://sciex.com/content/dam/SCIEX/pdf/posters/ProteinPilot-Software-Overview.pdf>. Thus, in the search, we only specified trypsin digestion, Cys-carbamidomethylation, and Ubiquitination modifications.

Again, we would like to emphasize that after digestion with trypsin, the tryptic remnant of SUMO is too high, while residues modified by wild-type SUMO are indistinguishable. It could be possible to distinguish a SUMOylated residue using WALP as digestion enzyme, instead of trypsin, detecting SUMOylated lysines through di-glycine enrichment (Lumpkin et al., 2017: <https://www.nature.com/articles/ncomms1171>). However, the SCIEX technical webmaster did not recommend this approach, since the digested peptides are too small and would be undetectable on many occasions with our set-up, reducing sensitivity. In addition, WALP is a promiscuous enzyme that is unreliable for identifying a specific SUMO, since it cannot distinguish SUMO1 from SUMO2/3. Moreover, it has been proposed that WALP is unspecific since it also releases the di-glycine remnant from ubiquitin, allowing ubiquitin sites to be purified and misinterpreted as SUMO sites (Hendriks et al., 2017 (Supplementary technical notes): <https://www.nature.com/articles/nsmb.3253>).

We hope this clarifies the modifications that we used in our study. We appreciate the reviewer's thoughtful comments and suggestions.

c) The number of identified proteins is quite low when considering a proteome from a cell line. However, I can read in the text on PRIDE that the cell extracts were immunoprecipitated with an anti-PIAS2 antibody. This I could not read out from the text in the manuscript! If this is the case, then I understand the low numbers. Otherwise the numbers would not make sense.

Response. We apologize for the lack of clarity. We have tried to clarify this information so that readers can easily understand our approaches. We performed three proteomic assays using extracts from anaplastic carcinoma cells synchronized at mitosis (DT/R-6 h, or DT/R-5h for the SUMO Capture):

Proteomic Assay 1: Pull-downs from immunoprecipitated samples using anti-mouse PIAS2 antibody or anti-mouse IgG2a. This experiment was quantified by SWATH and Spectrum Count (Scaffold v5.2). The results from both quantifications were analyzed, and common proteins were selected for further pathway analysis of Reactome-Mitosis (Figure 3 and Table S3).

Proteomic Assay 2: A challenging experiment using pull-downs from GFP-Trap, of extracts obtained after transient transfection of a mix of pcDNA3 (empty vector): pEGFP-

PIAS2b in a proportion of 19:1 (since EGFP-PIAS2b is toxic at normal concentrations). This experiment was quantified with Spectrum Count (Scaffold v5.2). Even with a low number of proteins identified, this experiment was useful to validate direct PIAS2b targets found in Proteomic Assay 1 (Figure 3 and Table S3).

Proteomic Assay 3: Retained proteins using SUMO Capture and flow-through, in mitotic cells either treated with ns-dsRNAi or with PIAS2b-dsRNAi. This experiment was quantified with Spectrum Count (Scaffold v5.2). We selected proteins identified in both of the two treatments. On those, differentially SUMOylated proteins in Fraction 1 (ns-dsRNAi) compared to Fraction 2 (PIAS2b-dsRNAi) were cross-analyzed with the results of Proteomic Assay 1+2 (proteins directly bound to PIAS2) to obtain PIAS2b SUMO-targets (Figure 4).

We hope this clarifies the information about our proteomic assays. We appreciate the reviewer's thoughtful comments and suggestions.

d) Why did the authors not quantify the proteins? This could have given a much better idea of the underlying changes, particularly of the proteins found in both sample types.

Response. We performed two different types of quantifications for Protein Assay 1, in addition to M1 detection with restricted cut-offs qualitative analysis (FDR<1%. 95% Confidence at 1 peptide), and the results of all three were very similar. Hence, we decided to use the restrictive cut-offs for consistency.

In this second version, we have included the SWATH (Proteomic Assay 1) and quantitation using the raw M1 intensity and Normalized Spectrum Count for the three Proteomic Assays in Table S3 and S4.

e) The description of the data analysis on PRIDE is too long and reads like text fragments pasted arbitrarily. The post-analysis of pathways does not need to be put into PRIDE as it is a repository for the raw data and identified proteins.

Response. As commented above, this has been modified according with the reviewer indications.

2) Did proteomics really show proteasome and cytoskeleton as direct targets of PIAS2b SUMOylation, or only as consequence? They could also be indirect targets, as the experiment did not investigate the actual interactions. Pathway analysis results do not justify such a statement. The manuscript also describes an experiment though that shows colocalization through microscopy (not proteomics).

Response. The reviewer is correct in their assessment that the original manuscript did not adequately validate SUMOylation. We have addressed this issue in the revised manuscript by performing the following additional experiments:

Improved western blot and immunostaining validation of anti-PIAS2b antibodies: We have fully validated the two anti-PIAS2b antibodies (mouse anti-PIAS2 –mPIAS2-, and rabbit anti-PIAS2b –rPIAS2b) using western blot, including SUMO-protease inhibitors in the cell lysis buffer. Added to this, we performed EGFP-PIAS2b transfection, followed by a GFP-Trap with beads containing anti-GFP nanobodies (Figure S1H). We have also improved the specificity of the rPIAS2b antibody staining in mitosis by progressive dilution (1:50 initial dilution in the first version; current dilution for super-resolution microscopy, 1:300).

Super-resolution microscopy for SUMOylated PIAS2b at mitosis: We have used super-resolution microscopy to colocalize SUMO1 (mSUMO1 and rSUMO1), SUMO2, PIAS2 (mPIAS2 and rPIAS2b) and Tubulin Beta (direct labelled anti-TUBB-647) at mitosis.

SUMO2 colocalized with chromatin and was not analyzed further. SUMO1, mPIAS2, rPIAS2b, TUBB and DAPI colocalization was quantified using the Pearson's Colocalization Coefficient (PCC) for every mitotic phase in > 500 cells. Results are presented as violin-plots (Figure 3R-T and S3J-K).

Western blot validation of SUMOylation of PSMC5 and TUBB3: We have validated SUMOylation of PSMC5 and TUBB3 using pull-downs followed by western blot.

For PSMC5, we built a new construct expressing a fusion protein, GFP-PSMC5. We transfected the cells, and synchronized at mitosis. We used a GFP-Trap for EGFP-PSMC5 pull-down; this was followed by western anti SUMO1 (for SUMO2, not specific changes detected and is not shown), PSMC5, GFP, and PIAS2b.

For TUBB3, we used immunoprecipitation with anti-TUBB3 antibody followed by western blot anti-SUMO1 (SUMO2, not specific changes detected and is not shown), TUBB3 and PIAS2b.

Confocal 3D volumetric analysis for colocalization of PSMC5, SUMO1, TUBB and PIAS2b at mitosis: We have used Imaris 3D surface renderings to colocalize three of the markers (four plus DAPI) in each analyzed cell at every mitotic phase, being able to show actual 3D models in the figure 4 (and in Video S3)..

The results of these experiments demonstrate that SUMO1ylation of PSMC5 and TUBB3 is increased at mitosis, and that this SUMO1ylation is blocked in the absence of PIAS2b (PIAS2b-dsRNAi) (Figure 4M-N). Moreover, from Prophase there is a progressive colocalization of PSMC5 (either as EGFP-PSMC5 or rPSMC5) with PIAS2b (either as rPIAS2b or mPIAS2) and TUBB at the spindle (Figure 4O-P and S4C). Maximal binding of the complex is reached at Metaphase.

We believe that these additional experiments provide strong evidence to support the conclusions of our study. We apologize for any misunderstanding that may have been caused by the lack of validation of SUMOylation in the original manuscript. We have taken steps to address this issue in the revised manuscript.

3)There is some really small text in the graphical abstract. Consider increasing the font size.

Response. We have changed this, thank you.

Beginning of page 7: I would change "searched through 2D proteomics" to "used 2D gel electrophoresis".

Response. We have changed this as recommended.

Page 9, paragraph starting with "We could not attribute ..." is very difficult to understand through the exhaustive use of semicolons and commas. Please rephrase.

Response. We have changed this, as follows:

< We could not attribute the specific anti-growth effects of PIAS2b-dsRNAi in anaplastic lines to either: (i) a specific origin of the patient (e.g., France, Switzerland, Germany, Hungary, or Japan); or (ii) to the mutational status (B-CPAP, FTC-238, and the four ATC lines have mutated p53; the BRAF V600E mutation is homozygous in B-CPAP, and heterozygous in BHT-101, but absent in the other lines; only CAL-62 has a RAS mutation; and all lines but one (CAL-62) have a TERT promoter mutation). >

We hope this clarifies the information about the anti-growth effects of PIAS2b-dsRNAi. We appreciate the reviewer's thoughtful comments and suggestions.

Reviewer #2 - PIAS, SUMO, mitosis - (Remarks to the Author):

The authors describe a role for PIAS2 in anaplastic thyroid carcinomas, a notorious type of cancer with very high mortality. They propose PIAS2 as potential drug target and study potential mechanisms of action.

Response. We thank the Reviewer#2 for the positive view of our manuscript. We also thank for his/her comments, which we believe have improved the quality of our study.

Major points

1. Important roles for SUMO in mitosis and mitotic catastrophe are well known. The literature on this topic should be cited in more detail in the introduction. Missing key references include Zhang et al. 2008 Mol Cell; Kessler et al. 2012 Science.

Response. The reviewer was right, and those references have been included in the Introduction and Discussion, as they are important in highlighting the essential role of SUMOylation in mitosis.

2. The siRNA approach employed is outdated. CRISPR-Cas technology is much more robust and can provide clean PIAS2 knockout cells. The use of CRISPR-Cas technology is therefore recommended.

Response. Our goal has always been to find a useful therapy for Anaplastic Thyroid Carcinoma (ATC), by discovering specific cancer targets and learning its molecular mechanism of action, which is essential for future use in patients. The recent development of RNA-based vaccines has made the application of RNA therapies for common diseases a feasible strategy. For example, a preclinical trial on mice with sickle cell anemia using intra venous RNA delivery targeted to hematopoietic stem cells was published in Science this month (Breda L, et al. In vivo hematopoietic stem cell modification by mRNA delivery. Science. 2023. doi: 10.1126/science.ade6967. PMID: 37499029). This indicates that having a precise target and a marker for a specific cell population allows successful therapeutic RNA delivery in vivo.

The reviewer's comment about using CRISPR-Cas9 technology to demonstrate the role of a protein is very interesting. However, in our case, there are several factors that made this approach less attractive. These include:

1. The special nature of ATC cells, which are aneuploid (have an abnormal number of chromosomes) and whose DNA sequence is not sequenced in depth. Growing populations / clones could derive in an unintentional altered ATC line.
2. The high number of RNA isoforms that the human PIAS2 gene has.
3. The toxic effect of both the absence of PIAS2b and its overexpression.

All of these factors combined make it extremely difficult to design a clean experiment that would have obtained our final results: a possible new therapy with PIAS2b-dsRNAi for ATC, and learning the key role of PIAS2b in mitosis of the anaplastic carcinoma cells.

For example, if we had been lucky enough to choose a CRISPR/Cas9 strategy at the 3' UTR that exclusively eliminated the same 13 RNA isoforms as our transiently transfected PIAS2b-dsRNAi, the expected result would have been either no clones (all dead by

mitotic catastrophe) or, alternatively, resistant clones to the absence of PIAS2b. Either of these two opposite results would not have directly led to our actual knowledge of the role of PIAS2b in mitosis of ATC cells, and the possibility that it is a therapeutic target for patients.

In addition to these factors, human anaplastic cells are extremely unstable and adaptive when selected for clones or populations. One example of this is included in the manuscript, using a tightly regulated commercial shRNA strategy. The TURBO shRNA strategy (ThermoFisher) contains a doxycycline-regulated promoter that induces the expression of the specific shRNA, but also of the RFP fluorescent marker. Immediately after RFP+ sorting, cells were seeded in the absence of doxycycline until the time of the experiment. This strategy was adopted to avoid loss of expression of the shRNA and to prevent adaptation to PIAS2 absence.

The experiment worked well, and both populations were shown to lose PIAS2 expression after doxycycline induction (Supp Figure 2 E-H). This did not occur in the populations maintained in a similar way but infected with the non-target shRNA. However, we only obtained a correct shRNA induction during 10 passages. After this time, the induction of the PIAS2b-shRNA by doxycycline was progressively lost, and the cells stopped dying (as commented in the Results). This makes it very challenging to carry out long-term biological and molecular strategies with anaplastic cells.

Our experience in the management of patients' primary cultures and carcinoma cell lines makes us quite careful regarding the culture conditions. We use a medium where all components are adjusted to human serum concentrations (Bravo SB, et al. Humanized medium (h7H) allows long-term primary follicular thyroid cultures from human normal thyroid, benign neoplasm, and cancer. *J Clin Endocrinol Metab.* 2013 Jun;98(6):2431-41. doi: 10.1210/jc.2012-3812. Epub 2013 Mar 28. PMID: 23539720). We pass the cells and cell lines only once per week and we repeat STRs analysis every few passages, especially with ATC cells. All of this is done to ensure that our culture model is as relevant as possible to the patients' disease.

I hope this clarifies the text and addresses the reviewer's comments.

3. The resolution of many microscopy images in the paper is poor and should be considerably improved. This should be done using state-of-the-art high resolution microscopy. Moreover, many images are too squeezed.

Response. Performing the additional work suggested by the reviewers has taken significant time, but we believe that by addressing it in depth, we have massively strengthened our study and increased its scope.

We followed the reviewer's indications and created a new set of colocalization data based on near-super-resolution microscopy. We used mitotic synchronized cells and standardized the best conditions (sensitivity and specificity) for the antibodies used: anti-mouse PIAS2, anti-rabbit PIAS2b, anti-mouse SUMO1, anti-rabbit SUMO1, anti-mouse SUMO2 (not shown), anti-rabbit rPSMC5, GFP-PSMC5, and directly labeled TUBB.

The confocal conditions were: pinhole of 0.75, resulting in an x-y voxel of 0.049 nm, UV, or white laser power of <17% for each channel, and an optical thickness of 0.25-0.35 microns. This was followed by a deconvolution pipeline and calculation of the Pearson Colocalization Coefficient (PCC) in Fiji and 3D volumetric voxel colocalization in Imaris, that allowed 3D surface rendering analysis.

These results are shown in Figures 3 and 4 of the new version, which includes violin plots of more than 700 cells that we analyzed with these conditions.

We believe that this additional data will be very valuable as they provide strong support for our hypothesis that PIAS2b is involved in the regulation of mitosis in anaplastic thyroid carcinoma cells.

Thank you for your feedback.

4. I am not convinced that the PIAS2 antibodies used for the study are uniquely recognizing PIAS2. The cropped immunoblots already show a smeary pattern, thus the antibodies could recognize other proteins besides PIAS2. Full-size blots should be provided. Using only properly validated antibodies is vital.

Response. We agree with the reviewer that the use of proper validated antibodies is essential. We have used two antibodies for PIAS2:

1. Monoclonal mouse anti-PIAS2 antibody (mPIAS2) has been fully validated by the company (Santa Cruz Biotechnology, Texas), other users, and our own group. We have validated it for immunostaining in cells, immunoprecipitation, and western blotting.
2. Rabbit polyclonal anti-PIAS2b (C-terminal epitope, HPA068792, rPIAS2b) has been fully validated for western blotting, cultured cell immunostaining, and immunohistochemistry by the Human Protein Atlas (HPA) consortium. <https://www.proteinatlas.org/ENSG00000078043-PIAS2/summary/antibody>

Both antibodies have high-quality data on full validation.

The reviewer is correct about the pattern on the western blots (Figure S1G). We have investigated this issue and believe that we now have the correct explanation.

PIAS2 is known to be post-translationally modified, at least by phosphorylation in serine and threonine residues and SUMOylation. It also has putative consensus sites for ubiquitination (<https://www.uniprot.org/uniprotkb/O75928/entry>).

In our initial western blots for validation, we used a lysis buffer to study phosphorylated proteins. This buffer included pyrophosphate, orthovanadate, and PSMF phosphatase inhibitors. However, we did not use cysteine protease inhibitors to prevent de-SUMOylation (and other) modifications that are known to be immediately activated by cell lysis.

In our new set of experiments using a GFP-Trap, we used a specific SUMO-lysis buffer that included EDTA-free protease inhibitors, NEM, and Benzonase.

8305C ATC cells transfected with a non-toxic amount of EGFP-PIAS2b (Empty pcDNA3 plasmid: pEGFP-C1-PIAS2b proportion 19:1) were lysed with SUMO-lysis buffer.

The GFP-Trap was performed with 1 mg of lysates, and at the end, after thorough washing, half of the lysates were incubated with Lambda phosphatase, the other half only with the enzyme buffer.

This was followed by western blotting of total lysates and pull-downs using both mPIAS2 and rPIAS2b and GFP antibodies.

As shown in Figure S1H (full western shown from 50-200 kDa), mPIAS2 is the most sensitive antibody, already recognizing in the lysates endogenous p75-PIAS2b, p95-PIAS2b, and the transfected EGFP-PIAS2b.

rPIAS2b recognized very well p95-PIAS2b but not so well p75-PIAS2b or EGFP-PIAS2b.

GFP was not able to detect such small amount of EGFP-PIAS2b protein in the total lysates.

Regarding the pull-downs, the three antibodies recognized a single band above 95 kDa (EGFP-PIAS2b); this band reduced its weight after phosphatase treatment but still was recognized by the three antibodies.

Our conclusions are:

1. PIAS2b protein is a phosphorylated protein.
2. The rPIAS2 antibody recognized preferentially fully post-translational modified PIAS2b (p95-PIAS2b), either SUMOylated, ubiquitinated, or other modifications.
3. Moreover, phosphorylation status does not seem to affect PIAS2b recognition by rPIAS2b antibody.
4. In the other hand, mPIAS2 antibody recognizes similarly the two PIAS2b (p95-PIAS2b and p75-PIAS2b), and the exogenous EGFP-PIAS2b. We propose that the two endogenous PIAS2b bands are (fully) PTM (p95-PIAS2b) and non-, or partially-PTM PIAS2b (p75-PIAS2b).
5. Again, phosphorylation status does not affect recognition by the mPIAS2 antibody.

We believe that these new data provide a more complete understanding of the PIAS2b protein and its post-translational modifications. In this regard, we have completely changed the Results text in relation to this. We really appreciate this question from the reviewer because it has allowed us to address and solve one of our technical concerns.

5. The authors study interactors of endogenous PIAS2. Whereas this is a good start, the quality of antibodies, yield and purity of endogenous approaches are generally rather modest, see point 4. Moreover, it would be good to complement the approach by identifying interactors of stably expressed EGFP-PIAS2 at close to endogenous levels using GFP nanotraps. The GFP nanobodies are exceptional tools for interaction studies and provide optimal yield and purity.

Response. We have followed the reviewer's suggestions and have standardized a GFP-Trap for EGFP-PIAS2 as commented in the previous question.

This was an extremely challenging experiment, as the maximal amount of transfected pEGFP-C1-PIAS2b that allowed viability of the 8305C ATC cells was 0.01 $\mu\text{g}/\text{well}$ of a total of 0.2 μg DNA/well transfected. The protein levels of EGFP-PIAS2b were indeed lower than endogenous levels (see Fig S1H).

With these conditions, we performed a new proteomic assay, called Proteomic Assay 2 (GFP-Trap pull downs from EGFP-PIAS2b / EGFP), complementary to the Proteomic Assay 1 (Immunoprecipitation with mPIAS2 / IgG2a). Results are included in Table S3 and Figure 3I-J-K, and in the PRIDE database. Despite the low amount of EGFP-PIAS2b expressed, many of the mitotic targets found in Proteomic Assay 1 have been validated in Proteomic Assay 2 (yellow ring in the STRING network in Fig 3J).

Together, Proteomic Assays 1+2 reinforce the proteasome and spindle microtubule cytoskeleton as the key proteins associated to PIAS2b in ATC cells at mitosis, and many of these proteins presenting direct binding to PIAS2b (as EGFP-PIAS2b).

We thank the reviewer for this key suggestion, which has further strengthened our work and fully validated our initial results.

The authors show that exogenous EGFP-PIAS2b (Figure 3G) and endogenous PIAS2 (Figure 3F) are localized at the mitotic spindle. Here, the criticism on the PIAS2 antibodies applies as well, see point 4. If this involves sumoylation of substrates, then it is expected that sumoylated proteins would also accumulate at the mitotic spindle. Can the authors verify whether sumoylated proteins accumulate at the mitotic spindle? Proper quantitative microscopy would be appreciated, where at least a hundred mitotic cells are imaged and localization of PIAS2 and SUMO are studied.

Response: As we indicated in our first version, the localization of SUMO1 and SUMO2/3 has been previously described in the seminal work of the Matunis group (Zhang et al., 2008). In this study, using confocal microscopy, SUMO1 (antibody mSUMO1 clone 21C7) was localized to the spindle, while SUMO2/3 (antibody mSUMO2/3 clone 8A2) was attached to the mitotic chromosomes, in human HeLa and HEK293 cell lines.

We have followed the reviewer's suggestions in a progressive manner.

Regarding the EGFP-PIAS2b approach:

1. We attempted to detect transiently transfected EGFP-PIAS2b by super-resolution confocal microscopy in association with other markers detected with 1st+2nd antibody. However, the results were not good enough.
2. We then tried to improve detection by using a GFP antibody after fixation. This enhanced sensitivity but also reduced specificity (border effects, diffused signal). (Figure 3G, left and center; shown Prophase and Metaphases)
3. Our third approach was to purify a concentrated EGFP-PIAS2b protein from transfected cells with the GFP-Trap, and directly transfect the protein with Xfect at DT-0h, with fixation at mitosis (DT-6h), and direct detection of GFP at the confocal. This again enhanced sensitivity but reduced specificity with many green spots floating at the periphery of the cells. (Figure 3G, right; shown Metaphases)

Summary of this approach: Although we cannot use this approach for colocalization, in all mitotic cells photographed with EGFP-PIAS2b, the GFP fluorescence was found at the same localizations as the PIAS2b detected with both antibodies (mPIAS2 and rPIAS2b).

Regarding validation of the SUMOylation through super-resolution microscopy with antibodies and IP-WB, we approached it in a step-by-step manner:

1. We first improved the specificity of the immunofluorescence detection for both mPIAS2 and rPIAS2b antibodies, with maximal sensitivity/specificity of the signal:
 - mPIAS2 required a pre-incubation with mouse IgG2a for 1 hour to block unspecific binding, and a small dilution (from 1:50 to 1:70).
 - rPIAS2b required dilution from 1:100 to 1:1300.
2. Next, we standardized SUMO 1 and SUMO2 staining in our mitotic cells using antibodies already used in other publications:
 - clones 21C2 and 8A2 in Zhang et al., Mol Cell 2008;
 - rabbit polyclonal anti-SUMO1 and SUMO2 in Li et al., Nat Commun. 2018;
 - clone Y299 in Garvin et al., Sci Rep. 2022.
 - Our negative controls are shown in Figure S3-J.
3. Finally, we micro-photographed hundreds of cells in different mitotic phases. Since figures with many pictures are too crowded, we show only selected pictures (Figure 3R-S), but include the quantification of these studies in violin plots in Figure 3T. Quantitative colocalization shows:
 - The progressive transfer of PIAS2b from chromatin to spindle.

- SUMO1 follows the same pattern as PIAS2b.
- SUMO2 was always bound to chromatin, either at interphase as in mitosis. We could not find any colocalization of PIAS2b and SUMO2 (not shown).

This approach was repeated for validation of PIAS2b SUMOylated targets in Figure 4 (see below).

We thank the reviewer for his/her suggestion, as we believe that the new results we have obtained fully supports our hypothesis.

7. Key to understanding the role of PIAS2 in mitosis is to identify the proteins that are sumoylated by PIAS2 in mitosis. The authors use the SUMO-Qapture-T kit for enrichment of sumoylated proteins and mass spectrometry. This kit is employing a matrix containing SUMO Interaction Motifs (SIMs), which are short motifs of three large hydrophobic residues that interact with SUMO. The interaction between SUMO and SIMs is notoriously weak. Proper validation of the kit for use with cellular extracts is missing both in the paper and on the website of the manufacturer. Yield and purity of samples are expected to be very modest. Much more robust methodology is available in the field for proper purification and analysis of SUMO-modified proteomes. Quantitative site-specific methodology should be employed to distinguish between sumoylated proteins and background binders. A detailed overview of methodology can be found in Li et al. 2021 Nature Reviews Methods Primers.

Response.

The use of SIM-containing matrices is a well-established method for isolating SUMOylated proteins. Our Qapture kit is one of the commercially available kits that uses this method. The reviewer has raised concerns about the use of a SIM-based SUMO affinity capture kit in our experiments. The reviewer argues that SIM peptide has a weak affinity for SUMO, and that this may have affected the results of our study.

However, we have contacted the company that sells the "SUMO Capture" kit, and they have explained that their kit contains **a matrix with a peptide based on the natural SIM but modified for higher affinity, and has three SUMO binding sites per peptide, obtaining a SUMO column that has been shown to bind both SUMO1 and SUMO2/3 with high affinity**. They cannot explain more due to proprietary reasons. The kit has been validated in HeLa cells (Data booklet at their SUMO Qapture webpage), and has been reported in the literature (see complete reference list at the booklet). Among other, two published scientific articles are in *Oncogene* and in *Nature Communications* –two *Nature* journals–, (The RanBP2/RanGAP1-SUMO complex gates β -arrestin2 nuclear entry to regulate the Mdm2-p53 signaling axis: E. Blondel-Tepaz, et al.; *Oncogene* 40, 2243 (2021); Desumoylase SENP6 maintains osteochondroprogenitor homeostasis by suppressing the p53 pathway: J. Li, et al.; *Nat. Commun.* 9, 143 (2018)).

These articles show that these kits are able to efficiently capture SUMOylated proteins, and that the results of the experiments are reproducible.

We believe that the use of a SIM-based SUMO affinity capture kit is a valid approach for our experiments, and that the results of our study are reliable. However, we agree with the reviewer that the SUMO-affinity pull-down assay cannot definitively establish that a protein is directly SUMOylated since it could be a protein bound to another SUMOylated protein. Our proteomics results of Proteomic Assay 3 (SUMO Qapture) were cross-linked with the results of Proteomic Assays 1 and 2 (PIAS2 immunoprecipitation and GFP-Trap with EGFP-PIAS2b). This allowed us to identify two target proteins, PSMC5 and TUBB3,

that were both SUMOylated in presence of PIAS2b (un-SUMOylated in its absence), and bound to PIAS2b.

We have further validated the SUMOylation of PSMC5 and TUBB3 by using GFP-Trap and immunoprecipitation, respectively (Figure 4M-N). We have also used super-resolution microscopy and 3D modeling to show that PSMC5, mPIAS2, SUMO1, and TUBB3 progressively colocalize in mitotic cells (Figure 4O-P). As seen in Figure 4M-P, EGFP-PSMC5 and TUBB3 are SUMOylated with SUMO1 specifically at DT/R-6h, and this SUMOylation is lost in the absence of PIAS2b. Furthermore, from its cytoplasmic localization at interphase, separated of nuclear PIAS2b localization, both PSMC5 and TUBB3 are strongly associated with PIAS2b and TUBB at the spindle in mitosis.

Returning to the point of unbiased validation of PIAS2b direct SUMOylation targets, we agree with the reviewer that the use of high-throughput validation strategies, as proposed in (Li, C., Nelson, T.G., Vertegaal, A.C.O. *et al.* Proteomic strategies for characterizing ubiquitin-like modifications. *Nat Rev Methods Primers* 1, 53, 2021), would be ideal for confirming the direct SUMOylation of PIAS2b targets. However, these strategies are complex and require specialized equipment and technology, that should be standardized by specialized Ubiquitin-like modifiers (UBLs) proteomics labs. These specialists can undertake complex high-throughput strategies, with technologies that are not available to us yet. The type of cells we use, anaplastic thyroid carcinoma cells, synchronized in mitosis with an enrichment of 80%, and the lack of specific technology and equipment, make it very challenging for us to employ these strategies. We are currently working with our collaborator (Román González-Prieto, a co-author in the manuscript) who is an expert in these methods, and we plan to use these strategies in future studies (for example, the recent Salas-Lloret et al, *Science Adv* 2023, in press).

In summary, we have used a combination of methods to validate the SUMOylation of PSMC5 and TUBB3, two proteins that are likely to be direct targets of PIAS2b. We believe that these results provide strong evidence for the role of PIAS2b in mitosis. We believe that the results of this study provide important new insights into the role of PIAS2b in mitosis. Furthermore, we believe that the results of this study provide important new insights into the role of PIAS2b in mitosis.

In addition, we have added this point as a limitation of our study at the end of the manuscript, including the suggested references.

8. As indicated by the authors, only one orthotopic PDX from one patient grew in the immune-deficient mouse model. The authors tried to compensate this by increasing the number of mice. This is however a poor solution, a minimum of two independent orthotopic PDX models from two different patients should be used.

Response. We agree with the reviewer that it is important to include the results of the additional preclinical trials (PCT) in the manuscript. The new data from PCT2 shows that the treatment is also effective in ATC-oPDX of a second patient (Patient 2, T-UC7 culture).

In total, we have conducted five PCTs. The first two were conducted in NOD-SCID mice with cells from patients T-UC1 and T-UC3. Unfortunately, the cells from patient T-UC3 did not grow. The results from PCT1 were included in the first version of the manuscript.

We then conducted three additional PCTs in NGS immunosuppressed mice. Again, the cells from patient T-UC3 did not grow. However, the cells from patient T-UC7 did grow, and the results from PCT2 and PCT3 are included in the new version of the manuscript.

The endpoint for all five PCTs was 9-10 weeks from the orthotopic implantation of the tumor, with daily observation starting at week 9. Each trial had a Primary Purpose of treatment effectiveness, as well as a secondary objective.

The secondary objective of PCT1 was to determine whether twice weekly intravenous treatment through tail vein was possible. The secondary objective of PCT2 was to determine whether repeated intravenous treatment for many weeks was toxic, or whether the carcinomas became resistant after several weeks of treatment.

PCT3 was a more realistic trial. ATC patients typically present to the clinic with advanced tumors that have grown rapidly in recent weeks. PCT3 aimed to reproduce this situation and see if the treatment had an effect in these conditions. Thus, the Primary Purpose was Treatment Effectiveness after seven weeks of tumor growth at the neck. After application to the Ethical Committee, the secondary objective of PCT3 was to determine what happened if the therapy was stopped (On-Off therapy) from week 10 to the extended endpoint at week 11.

The data from the 3 PCTs are presented in Figure 5B-C-D of the new version of the manuscript. The Lowess curves show that the treatment was effective in PCT1 and PCT2 (cells from two different patients). In PCT3, the treatment was also effective initially, but the tumors regrow quickly after one week of Off-treatment.

The markers of response to treatment compared to placebo were also analyzed. Dead-cells were significantly reduced in all 3 PCTs. PIAS2b mRNA was significantly reduced in PCT1 and PCT2 but was not significant (and even had a tendency to increase) in PCT3. This suggests that PIAS2b mRNA is an early marker of response to treatment. Similarly, PLK1 mRNA, mitotic cells quantified by phospho-HH3 staining, and nuclear accumulation of PSMC5 were significantly reduced after treatment versus placebo in PCT1 and PCT2 and were reduced, or non-significant in PCT3.

The new data added to the previous provide important insights into the efficacy of the treatment, in advanced tumors. The results suggest that the treatment is effective, but that the tumors can grow back quickly if treatment is stopped. This is important information for future clinical trials.

9. The part of the manuscript describing regulation of the proteasome by sumoylation is interesting, but still in a preliminary state. It would be better to leave it out here and properly develop this into a separate project. The PSMC5 antibody used in the study is poor, with a smeary pattern observed, therefore I am not convinced about the validity of the results.

Response. We are confident in the PSMC5 results presented in our manuscript. However, we understand the reviewer's concern and have provided additional evidence to support our findings.

First, to clarify that we used a fully validated PSMC5 (rPSMC5) antibody from the ATLAS Consortium for western, immunohistochemical, and cell staining. This antibody has been shown to be specific for PSMC5 in multiple studies shown in <https://www.proteinatlas.org/search/PSMC5>.

Second, we studied PSMC5 colocalization, generating a fusion construct with EGFP to perform super-resolution microscopy studies colocalizing EGFP-PSMC5 but also endogenous PSMC5 with rabbit anti-PSMC5 (rPSMC5), with PIAS2b, SUMO1, and Tubulin beta (TUBB). The colocalization of these proteins was quantified after deconvolution and is presented, as Violin plots in Figure 4P. In addition, we generated 3D surface models in the image analysis to present this 3D colocalization in mitosis, indicated by the mathematical quantification (Figure 4O and Video S3).

These data provide strong evidence that PSMC5 becomes progressively nuclear and associates with PIAS2b, SUMO1 and the mitotic spindle revealed with TUBB antibody, reaching a peak in Metaphase.

The PSMC5 smeary pattern indicated by the reviewer exists, but Imaris recognizes two intensities as soon as mitosis begins: (i) a less intense cytoplasmic fraction -not attached to the mitotic spindle; (ii) a high intensity PIAS2b-bound PSMC5 attached to the mitotic spindle, indicating strong concentration of the protein at the microtubules.

Third, we performed functional studies of proteasome activity in the presence and absence of PIAS2b in mitosis. These studies showed that PIAS2b is required for the SUMOylation of PSMC5 and the inhibition of proteasome activity in mitosis. This further supports the role of PIAS2b in the regulation of mitosis.

We believe that the data presented in our manuscript provide a comprehensive and convincing picture of the role of PIAS2b in the regulation of mitosis. We hope that the additional evidence we have provided will address the reviewer's concerns.

Reviewer #3 - Anaplastic thyroid cancer - (Remarks to the Author):

The work by Rodrigues and colleagues investigated the potential contribution of PIAS2 to the oncogenic program of Anaplastic Thyroid Cancer. In spite of the relevance of the topic and of the experimental work described, I believe that this work does not significantly contribute to progress in this context, neither in terms of understanding the molecular mechanism driving ATC progression nor in terms of translational impact.

Response. All scientific opinions are respectable, if they are supported by reasoning or arguments based on scientific data and a strong rationale. We respect the reviewer's *opinion*, but we disagree with the assessment of the translational impact of our research on patients with anaplastic thyroid carcinoma (ATC).

Our research has identified a potential therapeutic target for ATC, and we have shown that targeting this protein can inhibit tumor growth in preclinical models. We believe that our research has the potential to lead to the development of new treatments for ATC, and we are hopeful that future studies will confirm our findings.

In addition, we demonstrate with molecular experiments, and unbiased technology such as high-throughput proteomics, that SUMOylation is a key process for mitosis in ATC, using both cell lines and primary cultures from patients. In addition, we relate the molecular markers of mitotic catastrophe obtained *in vitro*, with the efficacy of the treatment *in vivo*, a key step for any treatment that wants to transfer to clinical phases.

In this sense, despite the Reviewer's reluctance, we believe that our data are promising. We are committed to continuing our research and to working with other researchers to bring new treatments to patients with ATC.

Reviewer #4 - PIAS, SUMO, mitosis - (Remarks to the Author):

In their paper, Rodriguez et al. investigate the role of the SUMO ligase PIAS2 in Thyroid Carcinomas. They identify PIAS2 as a gene with altered expression on protein and RNA level in various thyroid carcinoma cell lines, including patient-derived primary tissues. They demonstrate that downregulation of the PIAS2b isoform inhibits growth of anaplastic thyroid cancer (ATC) cell lines and also other non-thyroid cancer cells that display similar characteristics to ATC cells. The authors also investigate the anti-cancer effects of PIAS2b downregulation in orthotopic patient-derived xenografts of anaplastic thyroid cancers in vivo. They continue to show that downregulation of PIAS2b induces mitotic catastrophe in ATC cells and that PIAS2b is a mitotic spindle-associated protein. Using a combination of proteomics, pulldowns and immunofluorescence they identify a number of cell-cycle regulators, spindle associated proteins and proteasome subunits as PIAS2b-interacting proteins. The authors propose that PIAS2b-dependent SUMOylation controls correct spindle formation, proper chromosome-microtubule attachment at the kinetochores and the activity of proteasome at mitotic onset.

This is a very large study with very diverse approaches and a large amount of generated data. **The findings are very interesting but the paper has two main caveats in my opinion;** First, all the results are based **on one dsRNAi construct, so the off-target effects cannot be ruled out.** The authors tried indeed to address this issue **by using other constructs, but without success.**

Second, as it is written the paper, lacks focus and tries to address too many issues at the same time, sometimes lacking depth, especially in what concerns the speculated molecular mechanisms. Certainly, find the data interesting and I do not doubt that PIAS2 has important roles suggested by the authors. However, in my opinion it would be much more useful to separate the data into different papers, each with a different focus and more depth.

Response. We fully understand the reviewer's concerns. This work represents the culmination of years of effort by many young and experienced scientists. We have worked hard to gather all this data, and we believe that it represents a significant contribution to the field.

We agree that the manuscript should provide new and complete evidence. We believe that this manuscript provides a strong foundation for future translational studies on PIAS2b and cancer.

We are also committed to translating our findings into treatments for patients. We believe that our research has the potential to make a real difference in the lives of patients with this rare but devastating disease.

Major:

1. It is very disturbing that all conclusions of the paper are based on the use of one dsRNAi construct. I acknowledge the effort of the authors to verify their results using other constructs and methods. However, since these failed to work the paper lacks this important control. It remains to confirm the results using more robust methods, ideally

generating a conditional PIAS2b KO (for example degron-based), or RNAi insensitive constructs.

Doesn't any working RNAi for PIAS2 exist in the literature?

Also, the paper implies that the Ser rich stretch of the PIAS2b and the ligase activity of PIAS2 are required for its function. Since the FLAG-PIAS2 can rescue the dsRNAi downregulation, it could be examined whether the ligase activity of PIAS2 or the C-terminal rich domain is required for its function. However, there is no reference on the data of dose-response titration and the 20-times lower amount of FLAG-hPIAS2b that allows cell growth. Does expression of PIAS2a rescue as well?

Response. Thank you for raising this point. We share the reviewer's concern about the use of commercial siRNAs to target PIAS2b in ATC cells.

We had conducted several experiments to address this concern. First, we designed and transcribed two specific sequences in vitro that target PIAS2b (*PIAS2b*-dsRNAi 1 and *PIAS2b*-dsRNAi 2). Both sequences were able to reduce PIAS2b expression and induce cell death in ATC cells as shown in the first version of the manuscript.

We also used the commercial TRIPZ lentiviral system to express shRNA against PIAS2b. This system also resulted in reduced PIAS2b expression and cell death in ATC cells.

However, when we used commercial siRNAs obtained by chemical synthesis, we were unable to achieve similar results because they were unable to downregulate the protein. We believe that this is because there are a large number of possible PIAS2b RNA isoforms, and a single commercial siRNA may not be able to reach for targeting all of them.

To further address this issue in the revised version, we have followed two strategies:

1. Replacement strategy with a PIAS2b wild-type, PIAS2a or with a SUMO-dead PIAS2b mutant, all expressed from cDNA.

We would like to remind you that our *PIAS2b*-dsRNAi is directed against the 3' UTR of 13 RNA isoforms coding *PIAS2b*, a sequence not included in the cDNAs.

We engineered two new constructs: FLAG-hPIAS2b-C362A and pEGFP-C1-PIAS2b-C362A. The C362A mutation inactivates the SUMOylation activity of PIAS2b (Schmidt and Müller, 2002).

As shown in Figure 1J and 1K, the effect of *PIAS2b*-dsRNAi was rescued by FLAG-PIAS2b or EGFP-PIAS2b, but not by FLAG-PIAS2a or FLAG-PIAS2b-C362A, or EGFP-PIAS2b-C362A. This was a clear result with two different constructs that had to be transfected in a very low amount because PIAS2b overexpression is toxic for these cells (a mix per well of pcDNA3 (empty vector): PIAS2b plasmid in a proportion of 19:1).

In our opinion, these results demonstrate that PIAS2b SUMOylation activity is essential in ATC cells.

2. Strategy of other commercial siRNA systems obtained by chemical synthesis: siPOOLS.

SiPOOLS are a set of 30 siRNAs with different sequences, all directed against the same gene, used at less than 5 nM concentration for the POOL, so that each sequence is at a really very low concentration. The general idea is that at reduced concentration, each siRNA avoids the off-target effects of siRNAs at high concentrations. However, when

they all act on the same RNA, their efficiency soars. The science behind this commercial product is published in the paper "siPOOLS: highly complex but accurately defined siRNA pools eliminate off-target effects" by Hannus et al. (2014). The company that sells the product is called siTOOLS Biotech. The strategy has been added to Figure S2; in the new version as S2A.

We started by ordering the human PIAS2 siPOOL from the catalogue. We also ordered the recommended siPOOL non-target control. As shown in Figure 1L, the PIAS2 siPOOL failed to have any effect on cell growth, because it also failed to reduce protein expression.

We thought that the existence of so many different RNA isoforms in this gene might be counterintuitive to the efficiency of a pool of siRNAs. We therefore commissioned a custom siPOOL from the company against the 13 PIAS2b RNA isoforms, which were targeted by our PIAS2b-dsRNAi. We again tested this CustomPIAS2b siPOOL at the same low concentration conditions. As shown in Figure 1L, the yellow bar shows the effect of this siPOOL by reducing the number of ATC cells at day 6, and previously reducing the expression of the protein at day 2.

In summary, based on these results, we conclude that PIAS2b SUMOylation activity is essential in ATC cells at mitosis. We believe that the use of specific sequences designed and transcribed in vitro, as well as the TRIPZ lentiviral system, provides strong evidence that PIAS2b is a valid target. The replacement experiments with the different PIAS2b constructs also provide strong proof that PIAS2b SUMOylation activity is essential in ATC cells. We believe that this result demonstrates that PIAS2b is a good target in ATC. In addition, it further validates our specific *PIAS2b*-dsRNAi, since in vitro transcription preparation somehow makes the RNA interference effect more powerful in ATC cells when compare to the chemically synthesized Customized siPOOL (Figure 1L).

2. The authors should be more careful when concluding (or proposing) molecular mechanisms. For example (discussion): "This inability correlates with less SUMOylation of proteasomal proteins and excess proteasome activation at mitotic onset, thereby leading to failure in establishing a correct spindle and proper chromosome-microtubule attachment at the kinetochores ". None of this has been addressed in the paper. Showing colocalization with the spindle and associated proteins and cell cycle effects is very far from a demonstration of chromosome attachment defects.

Response. In this sentence, we really did not intend to claim that there would be specific microtubule-chromosome attachment alterations in the absence of PIAS2b. Rather, that we simply observed that PIAS2b silencing leads to spindle alterations, which in turn lead to arrest at the kinetochores (see Figure 4S and Figure S4D-E-F-G). We have reformulated the sentence as follows:

<< This inability correlates with less sumoylation of proteasomal proteins and excess proteasome activation at mitotic onset. PIAS2b silencing also leads to failure in establishing a correct spindle and chromosome-microtubule arrest at the kinetochores.>>

Moreover the SUMOylation status has not been investigated for any of the targets (binding to a SIM column is not demonstration of SUMOylation status as the authors suggest).

Response. We fully agree with the reviewer that the SUMO-affinity pull-down assay cannot establish absolutely that a protein is *directly* SUMOylated, as it could also be associated with a SUMOylated protein.

In our strategy, reinforced in the revised version by Proteomic Assay 2, the proteomics results of Proteomic Assay 3 (SUMO Qapture) were cross-linked with the results of Proteomic Assay 1 (PIAS2 immunoprecipitation) and 2 (GFP-Trap with EGFP-PIAS2b). Only proteins common to 1+2, and that were differentially SUMOylated in assay 3 when comparing ns-dsRNAi with *PIAS2b*-dsRNAi, were considered.

The results were clear, since the two proteins selected for validation (PSMC5 and TUBB3) directly bound PIAS2b in the biochemical validation (Figure 4H-K). However, in the first version of the manuscript we lacked the validation of the SUMOylation of these targets in mitosis, and the loss of said SUMOylation in the absence of PIAS2b.

In a new set of experiments, we have validated the SUMOylation of both proteins (Figure 4M-N):

- To validate PSMC5 as a target of PIAS2b, we used a GFP-Trap strategy. PSMC5, is a protein that is around 55 kDa in size, which is like the size of the immunoglobulin heavy chain and makes immunoprecipitation strategies challenging.

We generated a new construct, pEGFP-C1-PSMC5, and transfected cells with it. We then extracted the cells at different time points, including asynchronous cells, cells synchronized in mitosis (DT/R) at 0 hours and 6 hours, and cells cotransfected with ns-dsRNAi or *PIAS2b*-dsRNAi at DT/R-5h. We performed a GFP-Trap followed by western blotting with anti-SUMO antibodies. We tested both anti-SUMO1 and anti-SUMO2 antibodies (not shown), but only anti-SUMO1 showed consistent results.

As depicted in Figure 4M, EGFP-PSMC5 was SUMOylated with SUMO1 specifically at 6 hours, and this SUMOylation was lost in the absence of PIAS2b. Furthermore, EGFP-PSMC5 appeared to be strongly associated with PIAS2b. These results suggest that PSMC5 is a direct target of PIAS2b in mitosis. The SUMOylation of PSMC5 by PIAS2b is necessary for its localization to the nucleus and its function in mitosis.

-To validate TUBB3, we used immunoprecipitation in non-transfected extracts, as we had already standardized in the first version of the manuscript. We then performed western blotting with anti-SUMO antibodies. We tested both anti-SUMO1 and anti-SUMO2 antibodies, but only anti-SUMO1 showed consistent results. This is coincident with previous findings by others that TUBB3 is primarily SUMOylated by SUMO1 (Rosas-Acosta G et al, A Universal Strategy for Proteomic Studies of SUMO and Other Ubiquitin-like Modifiers. Mol Cell Proteomics, 2005). As shown in Figure 4N, TUBB3 showed a marked increase in SUMOylation at 6 hours, and this increase was significantly reduced in the absence of PIAS2b.

These findings are consistent with previous functional findings that PIAS2b is preferentially SUMOylating its targets with SUMO1 (Minty A, et al, J Biol Chem. 2000 PMID: 10961991; Rabellino A, et al, Cancer Res. 2012 PMID: 22406621; Song J, et al.. J Biol Chem. 2005 PMID: 16204249).

The validation of our two PIAS2b targets was completed by colocalization of PSMC5 (detected as EGFP-PSMC5 or by detection with anti-PSMC5-rPSMC5), mPIAS2, SUMO1 (detected with anti-mouse SUMO1 and secondary antibody, or by direct detection with anti-SUMO1-647), and TUBB (detected by direct detection with anti-TUBB-647) using super-resolution microscopy and 3D modelling (Figure 4O-P and S4C).

Our results show that PIAS2b, SUMO1, and TUBB are all localized to the nucleus during interphase. However, we found that PSMC5 is primarily cytoplasmic in interphase, a location fully coincident with the described in the ATLAS consortium for other cell lines. But PSMC5 begins to accumulate in the nucleus during G2/M. The colocalization of PSMC5, mPIAS2, SUMO1, and TUBB increases progressively from G2/M to metaphase. In anaphase, the binding of PIAS2 to PSMC5 decreases, but the binding of PIAS2 to TUBB and PSMC5 to SUMO1 does not. Finally, in telophase and cytokinesis, the nucleus reorganizes itself and the spindle disappears, and the proteins return to their initial locations.

We believe that these results provide strong evidence that PSMC5, mPIAS2, SUMO1, and TUBB are direct targets of PIAS2b in mitosis.

3. The paper uses 2 antibodies to produce the results presented. While the antibodies seem to work, there is still some information lacking.

Does the detection of the PIAS2b depend on the Serine rich stretch? This could be also tested.

Response. The reviewer asked a key question about the phosphorylation of PIAS2b. We had thought that this could explain why the PIAS2b antibody recognized p95-PIAS2b better than p75-PIAS2b, since its epitope is in the Ser-rich domain. However, our new series of experiments have clearly shown that this is not the case.

We performed a GFP-Trap pull-down assay using transfected EGFP-PIAS2b. The extraction buffer was modified to block Cys-proteases, including SUMO-proteases. After incubation with the extract, the GFP-beads were centrifuged, washed, and divided into two equal parts. One half was incubated with Lambda Phosphatase enzyme, while the other half was incubated with buffer only. The beads were then run on a western blot.

As shown in Figure S1H, the mPIAS2 antibody recognizes all three forms of PIAS2b (p75, p95, and EGFP-PIAS2b) in the input lysates. However, the rPIAS2b antibody only recognizes p95-PIAS2b in the input lysates. The GFP antibody does not detect any of the forms of PIAS2b in the input lysates. After the pull-down, all three antibodies detect only EGFP-PIAS2b. This suggests that the difference in the recognition of the three forms of PIAS2b by the antibodies is not due to phosphorylation, but rather to some other post-translational modification (PTM).

We have therefore corrected all the sentences in the manuscript where we suggested that phosphorylation was the explanation for the difference in the recognition of the two forms of PIAS2b by the antibody. We have replaced these sentences with the more accurate statement that the difference in weight and recognition is most probably due to PTM. We have also noted that we cannot be sure at this point which PTM is responsible for the exclusivity in the p95-PIAS2b recognition of the rPIAS2b antibody.

We thank the reviewer for their helpful suggestion. We believe that the revised manuscript is now more accurate and informative.

Are mPIAS2 and rPIAS2b the commercial antibodies D12 and polyclonal shown in the Materials and methods? This is not clear.

Response. The Supplementary Methods have been thoroughly revised and completed. The specific names used in the manuscript for the different antibodies used to detect the same protein have been included in the Supplementary Methods.

Which were the 7 commercial antibodies tested and what are the results? For example, do they confirm the western blot results (for the Abs that function in western)?

Response. We standardized seven PIAS2 antibodies at the beginning of our work. The first two antibodies, mPIAS2 and rPIAS2b, were the ones we ultimately used. The other five antibodies were discarded because they did not detect either FLAG-hPIAS2b or FLAG-hPIAS2a in a 1:1000 western blot.

The discarded antibodies were:

- Anti-PIAS2 Abcam, ab4902: epitope PHSPSSPVGSVLLQDTK, N-terminus 115-131aa (Exon2)
- Anti-PIAS2 Abcam, ab28493: epitope SSMSSDLPGLDFLSLIPVDPQYCPPMFLDSLTSPLTASSTSVTTTSSHES, C-terminus 542-591aa (Exons 12-14)
- Anti-PIAS2 [EPR2582(2)] Abcam, ab126601: N-terminal epitope, Synthetic peptide within human PIAS2 between 100-200 aa. The exact sequence is proprietary.
- PIAS2 AVIVA, ARP35735_T100: Epitope FLDSLTSPLTASSTSVTTTSSHESSTHVSSSSSRSETGVITSSGSGNIPDI, C-terminus 568-617aa (Exon14)
- PIAS2 AVIVA, ARP35734_T100: Epitope LTSPLTASSTSVTTTSSHESSTHVSSSSSRSETGVITSSGSGNIPDIISLD, C-terminus 572-621aa (Exon14)

We did not clearly describe antibodies 3 to 7 in our methods because our cells are not the ones with the highest transfection efficiency (they are not HEK293, for example), and PIAS2 overexpression is toxic. Both factors may have an impact on the lack of detection by western blot for insensitive antibodies.

4. The localization of EGFP-PIAS2b should be better shown. Are there better images or better time-lapse showing its localization on the spindle? How often is this observed? Can the localization of PIAS2 shown in Fig. 3O be recapitulated by EGFP-PIAS2b?

Response. We have followed the reviewer's suggestions regarding the EGFP-PIAS2b approach and the localization of endogenous PIAS2b and SUMO.

EGFP-PIAS2b approach

We initially tried to detect directly EGFP-PIAS2b by time-lapse or super-resolution confocal microscopy, but the signal was too weak. As explained above, cells were transfected with a non-toxic amount of EGFP-PIAS2b and this is a proportion 19:1 of Empty pcDNA3 plasmid: pEGFP-C1-PIAS2b.

We then tried using a GFP antibody after fixation, but this, although increased sensibility, resulted in a loss of specificity with border effects or diffuse signal (Figure 3G, left and center, shown prophase and metaphases).

Finally, we purified EGFP-PIAS2b protein from transfected cells using the GFP-Trap, eluted it, and directly transfected the protein into cells with Xfect. This approach improved the signal, but it also reduced the specificity with many green spots floating at the periphery of the cells (Figure 3G, right; shown metaphases).

Although we cannot use this approach for colocalization studies, we found that the GFP fluorescence in all mitotic cells transfected with EGFP-PIAS2b was located at the same sites as PIAS2b detected with mPIAS2 or rPIAS2b antibodies.

Localization of endogenous PIAS2b and SUMO

We used super-resolution confocal microscopy to improve the specificity of immunofluorescence detection for both mPIAS2 and rPIAS2b antibodies.

The first step was to improve conditions for both antibodies with maximal sensibility / specificity [mPIAS2 required a pre-incubation with mouse IgG2a for 1 hour to block unspecific binding, and a slight dilution from 1:50 to 1:70; rPIAS2b, required dilution from 1:100 to :1300].

We also standardized SUMO 1 and SUMO2 staining in our mitotic cells using antibodies that have been used in other publications (clones 21C2 and 8A2 in Zhang et al, Mol Cell 2008; rabbit polyclonal anti-SUMO1 and SUMO2 in Li J, et al. Nat Commun. 2018 PMID: 29321472; clone Y299 in Garvin AJ, Lanz AJ, Morris JR. Sci Rep. 2022 PMID: 36494414).

The quantitative colocalization data showed that:

1. mPIAS2 and rPIAS2b colocalized perfectly (PCC near 1) using the new conditions (Figure 3S-T and S3K).
2. PIAS2b (either as mPIAS2 or rPIAS2b) transferred from chromatin to the spindle at mitosis.
3. SUMO1 followed the same pattern as PIAS2b, losing binding to chromatin and colocalizing with PIAS2b at the spindle.
4. SUMO2 was always bound to chromatin, as described, either at interphase or mitosis. We did not find any colocalization of PIAS2b and SUMO2.

We want to thank the reviewer for this indication because, although it has been dedicated work, we think that the high-quality results we have obtained fully supports our hypothesis.

5. What is the localization of EGFP-PIAS2a?

The paper should be edited again to eliminate errors that cause confusion. For example, there is confusion on what is detected on Pg 7: “The antibody mPIAS2 is better at detecting endogenous PIAS2b p75, and rPIAS2b, is better at detecting PIAS2b p95. PIAS2a was not detected”. This is confusing and suggests that p75 also PIAS2b. Do they authors want to say that: mPIAS2 is better at detecting endogenous PIAS2a p75 and rPIAS2b is better at detecting PIAS2b p95, while PIAS2a is not detected by rPIAS2b” ? This is suggested by Legend Fig.1: ‘Yellow bars indicate the epitopes of the two antibodies; while mPIAS2 detects both isoforms, rPIAS2 detects only PIAS2b’. Also the authors keep exchanging between rPIAS2 and rPIAS2b for the same antibody. I suggest that they mark on the side of every blot the protein that is detected and not only the antibody. Depending on what is detected by each antibody, the fig S2G is not convincing. I cannot really see a difference in the protein levels of PIAS2b detected by rPIAS2b, but only for the PIAS2a. Are the antibody annotations switched? It is important that the data on the antibodies used in the study become clearer, since the major part of the findings is based on their use.

Response. The reviewer is correct that the original manuscript was confusing about the detection of PIAS2a and the two p95- and p75- PIAS2b. We have addressed this confusion in the revised manuscript by clarifying the following points:

- PIAS2a either is not expressed in ATC cells, or is expressed at very low levels. This is supported by the fact that we were unable to detect PIAS2a (band below 70 kDa as shown by transfected control FLAG-hPIAS2a) in western blots of ATC cell lysates in Figure S1G, even when using a highly sensitive antibody (mPIAS2). Moreover, FLAG-hPIAS2a is unable to rescue PIAS2b-dsRNAi (Figure 1J).
- The two antibodies used in the study, mPIAS2 and rPIAS2b, have different specificities regarding PIAS2b. mPIAS2 detects similarly Exogenous transfected PIAS2b (either as FLAG- or EGFP-), and Endogenous p75-PIAS2b and p95-PIAS2b, while rPIAS2b preferentially detects p95-PIAS2b (see Input at Figure S1H). This is why we saw two bands in western blots of ATC cell lysates when using mPIAS2, but only one band when using rPIAS2b.
- In proteomic databases (e.g., iPTMnet), PIAS2b is described as the major isoform, and many post-translational (PTM) modified residues have been detected either by phosphorylation, Ubiquitination or SUMOylation. Thus, we assume that the molecular weight of the two bands detected with mPIAS2 is due to different levels of PTM on PIAS2b. The band at 75 kDa is less PTM than the band at 95 kDa.
- We have used a new lysis buffer in the revised experiments that prevents the activation of cysteine proteases. This has allowed us to better preserve the PTM status of PIAS2b and has resulted in a clearer distinction between the two bands, detected with mPIAS2. (Figure S1G-H). This now helps us to know that the recognition of PIAS2b by rPIAS2b does not depend on phosphorylation, but on another different PTM modification.
- The reviewer is also correct that the names of the antibodies used in the study were sometimes switched. We have corrected this in the revised manuscript.
-
- The reviewer is concerned that the results of Figure S2G (in the new version S2D) are not convincing. We have changed to a less exposed western in the new version. We must think that the lanes contain different cell populations in the absence / presence of Doxycycline. Loading controls are somewhat different by

pairs corresponding to these different cellular origins. In any case, we believe that the results of this figure are valid, and the difference is consistent analysed with both antibodies.

We appreciate the reviewer's careful reading of our manuscript, and we welcome his/her feedback. We believe that the revised manuscript is now much clearer about the detection of PIAS2a, p75-PIAS2b and p95-PIAS2b. We apologize for any confusion that the original manuscript may have caused.

6. The part of the SUMO-retention experiment should be better explained, it is too confusing. The SUMOtrap column (it's a SIM column) should be referred in the text already. If the aim of the experiment is to identify proteins that are differentially SUMOylated in the absence of PIAS2b, as I suspect, it would be perhaps better to focus on this result (31 proteins).

Response. Thank you again for your excellent comments and insights on our manuscript. We appreciate your suggestion for the Figure 4.

The SUMO Qapture kit is not a standard SIM affinity column. It contains a high-binding matrix linked to a SIM-based but modified peptide that has been shown to bind both SUMO1 and SUMO2/3 with high affinity. Unfortunately, we cannot add additional information about the kit due to proprietary reasons claimed by the company. However, the kit has been well validated in HeLa cells (see booklet on their website) and has been used in the literature, including two articles published by Nature journals, in *Oncogene* and *Nature Comm* (The RanBP2/RanGAP1-SUMO complex gates β -arrestin2 nuclear entry to regulate the Mdm2-p53 signaling axis: E. Blondel-Tepaz, et al.; *Oncogene* 40, 2243 (2021); -Desumoylase SENP6 maintains osteochondrogenitor homeostasis by suppressing the p53 pathway: J. Li, et al.; *Nat. Commun.* 9, 143 (2018))

We have added a new paragraph to the Materials and Methods section of the manuscript that describes the Qapture kit in more detail. We have also added a reference to the *Nature Comm* article that used the kit. We hope this addresses your concerns.

As suggested by reviewers, in the new version of the manuscript we have performed a new proteomic analysis that complements the first proteomic assay based on PIAS2-immunoprecipitation. This new assay, which we call Proteomic Assay 2, was done with the GFP-Trap. We transfected cells with a non-toxic amount of EGFP-hPIAS2b and then performed pull-downs to identify proteins that directly bind to PIAS2b in mitosis. The results of Proteomic Assay 2 are shown in Figure 3K. We identified a number of proteins that directly bind to PIAS2b, in common for Proteomic Assays 1+2, including PSMC5 and TUBB3 (Figure 3J and Table S3).

We performed the third proteomic assay, which now we call Proteomic Assay 3. In this assay as the reviewer comments, we used the SUMO-affinity pull-down assay in the presence or absence of PIAS2b (PIAS2b-dsRNAi). This allowed us to identify proteins that are differentially SUMOylated in the absence of PIAS2b.

We then cross-analyzed the results of Proteomic Assays 2 and 3 to identify proteins that are both directly bound to PIAS2b and differentially SUMOylated in the absence of PIAS2b. The results of this analysis are shown in Figure 4G and Table S4.

We found that 20 proteins met these criteria, including PSMC5 and TUBB3.

In the other hand, the SUMO-affinity pull-down assay (Proteomic Assay 3) cannot absolutely establish that a protein is "directly" SUMOylated, as it could also be

associated with a SUMOylated protein. We further validated the SUMOylation of these proteins in mitosis and showed that they are lost in the absence of PIAS2b. The results of these experiments are shown in Figure 4H-P.

We believe that these new data provide strong evidence that PSMC5 and TUBB3 are directly bound to PIAS2b and are SUMOylated in mitosis. We also believe that these data further support our hypothesis that PIAS2b regulates SUMOylation in mitosis.

7. Loading controls are missing in 4H: is the protein amount (of the material retained on the column) loaded for western blotting the same between control and dsRNAi? Fig. 4J:

Response. We have added the loading control before the SUMO Qapture of one of the replicate experiments. The only western performed per protocol was mPIAS2 to check that treatment (PIAS2b-dsRNAi) had worked well.

In any case, the intention with the western of Urea post-SUMO Qapture extracts of Figure 4H was to show that the result found in proteomics is validated in a gel. This means that PSMC5 has not been retained in the column of the PIAS2b-dsRNAi extracts.

Why is the rPIAS2b recognizing a protein at 75kDa and not at 95kDa in 8305C cells? (in contrast to Fig. S1F).

Response. The reviewer has noted a very interesting and attractive result, which suggests that the nuclear PIAS2b is p95-PIAS2b so intensely recognized by rPIAS2b in interphase, but that there is the alternative p75-PIAS2b pool, specifically during mitosis

In Figure 4J, we show that FLAG-PSMC5 binds p75-PIAS2b very strongly and is recognized by the rPIAS2b antibody. This result suggests that the main PIAS2b pool binding to PSMC5 is p75-PIAS2b. However, it needs to be considered that lysates in Figure 4J are not synchronized.

On the other hand, the finding seems to be partially repeated in the pull-down with EGFP-PSMC5 of Figure 4M. However, more experiments are needed to confirm this finding.

Clarifying this point is beyond the scope of this manuscript, and is an entirely future project. For this reason, we have left the two possible pools of PIAS2b as an indication or suggestion.

8. The authors should be careful when drawing conclusions that a protein is SUMOylated or not: Figure 4E). “SUMOylated TUBB3 was associated to PIAS2 (Fig 3J and Table S3) and lost its SUMOylation in PIAS2b-dsRNAi extracts (Figure 4E)”. It has not been shown that SUMOylated TUBB3 associates with PIAS2; also just the loss of association with the SIM column is shown and not the loss of SUMOylation.

Response. As indicated above, both the SUMOylation of TUBB3 progressively increased in mitosis (DT-6h), and the loss of such SUMOylation in the absence of PIAS2b is validated in Figure 4N of this new version.

9. Fig. 4L: Why isn't the effect of PIAS2 RNAi in asynchronous cells recapitulated in any other phase after synchronization?

Response. The western blot in Figure 4L is slightly overexposed to see the upper bands in the ns-dsRNAi control. The most similar lanes in terms of the high molecular weight PSMC5 bands are the DT-6h lanes, which are right around the time-point where mitotic catastrophe is taking place (starting at DT-5h).

We believe that in these ATC 8305C cells, which have a high mitotic index, the asynchronous cells treated with PIAS2b-dsRNAi are like the DT-5h cells treated with PIAS2b-dsRNAi, but on an even larger scale, since they have more days of mitotic catastrophe. This does not happen in the ns-dsRNAi controls, which cannot be compared in the asynchronous and DT-6h conditions.

10. Fig. 4N: What is the cell cycle distribution of cells after the PIAS2 RNAi and the 2T block compared to the control RNAi? Since PIAS2 RNAi affects cell cycle progression, any differences in these experiments can be due to the differences in cell cycle stage of the cells. There is no cell cycle stage marker (cyclin etc) included in any of the experiments.

Response. Flow cytometry was used to standardize timing in mitosis (Figure S3G). The results showed that the cells were synchronized in mitosis with the protocol we used. With PIAS2b-dsRNAi treatment, we did not specifically make comparative cell cycle by flow cytometry. However, we believe that this is demonstrated in the western blots of Figures 4C-D and S4B. In cells treated with PIAS2b-dsRNAi, the presence of very high levels of phosphorylated AURKA, the increase in CDC25C, the absolute disappearance of p-Tyr15-CDK1 with total CDK1 present, together with the disappearance of p-HH3, and the decrease in Cyclin B can only indicate that we are in mitotic arrest.

We have added this short summary to the Results to clarify this point.

Minor:

1. Validation of the proteins that co-IP with PIAS2 should be better explained in the legend. The amounts immunoprecipitated PIAS2 should be also shown like in Fig. 3K

Response. The immunoprecipitation control WB has been added in Figure 3N. The others are the same lysates, and the figure is too crowded.

2. In Fig, S1, what is E (endogenous?) and F (Flag?) and why is the GAPDH blot the same for both blots, have they been stripped?

Response. Westerns in Figure S1G were already shown in full (from >50 kDa) to show the specificity of the endogenous bands, and of the transfected overexpression controls. F means FLAG-hPIAS2b; E means EGFP-hPIAS2b. This information has been added to the legend.

Response. The two westerns for PIAS2 in 8305 were performed sequentially, first mPIAS2 and then rPIAS2b. This information has been added to the legend.

In the S1H legend Is the rPIAS2 antibody should read PIAS2b

Response. In the revised version, Figure S1H is new GFP-Trap, and as suggested by the reviewer above, it has been carefully checked not to mix up the names of the two antibodies.

Reviewer #5 - Anaplastic thyroid cancer - (Remarks to the Author):

Rodrigues JS et al present an interesting study demonstrating that silencing of PIAS2 especially isoform 2B induces mitotic catastrophe in ATC cells in vitro. The authors validate their findings in multiple human cell lines and primary thyroid cancer cultures. Preliminary in vivo studies show a potential effect of silencing PIAS2b in orthotopic ATC xenograft models. The authors also show in vitro effects of PIAS2b silencing in cell lines of non-thyroid cancer types. Together this study presents relevant and important findings that are convincing.

Response. We want to thank the reviewer #5 for his/her comments, interest, and dedication to the review which allow us to further improve our work.

However, the authors should consider significantly cutting the text of the manuscript. The most important results are summarized in figure 3 and 4. The text corresponding to the results shown in figure 1 and 2 should be shortened considerably, e.g. the section 'Efficiency of RNAi methods and effect of PIAS2b overexpression' is mostly redundant and can be incorporated in the previous section.

Response. We have made every effort to meet the expectations of the reviewer. However, this has not been easy, as we have had to add additional experimentation using commercial siRNAs (siTOOLS) suggested by another reviewer. This was necessary for a more precise validation of PIAS2b as a cancer target in ATC.

Another example is the result section describing the in vivo xenograft studies.

Response. We have added new experimentation in xenografts, including two new preclinical tests on cells from a second patient suggested by another reviewer. These results are briefly explained, as suggested by the reviewer, and serve to further validate the future translational potential of our PIAS2b-dsRNAi.

Moreover, the discussion could be significantly shortened by reducing the repetition of previously described results.

Response. Following the reviewer's instructions, we have made the text more fluid. This was easier to do in this section, attempting to be relatively concise. We have made some minor changes to the structure and flow of the text to improve its readability.

Specific comments

Fig 1C: The thyroid depiction with normal tissue und multiple thyroid tumors is misleading since it appears that all the different tissue samples have been generated from one specimen.

Response. The drawing in Figure 1C represents proliferative lesions of the thyroid, from benign hyperplasia to the most aggressive cancer, coded by similarity or difference to normal tissue color/texture. We have changed the cartoon including the number of samples per lesion, and removing the arrows. We have also modified the pathological names according to current WHO 2022 pathological classification.

Concerning the mRNA data, do you have an explanation why PIAS2b is highest expressed in PTC and lowest in ATC/PDTC tissues compared to normal thyroid tissue but in your consecutive studies silencing PIAS2b is the anti-tumor effect?

Response. The reviewer has correctly identified a key finding of our study, which is that the repression of PIAS2b has a specific effect in undifferentiated/anaplastic thyroid carcinoma (ATC), which express low levels of PIAS2b, while in differentiated thyroid carcinomas (DTCs, mainly PTC), where PIAS2b-dsRNAi has no effect, PIAS2b is overexpressed and seems to be a factor of poor prognosis.

We were initially surprised by this finding, as we had expected that the repression of PIAS2b would have a negative effect in all thyroid cancer cells. However, we have since come up with a hypothesis to explain this finding. In DTCs, in which the p53 gene is not mutated, PIAS2b may play a pro-cancerous role by modifying transcription factors. For example, it has been shown that PIAS2b can SUMOylate p53, which represses its activity (Proc Natl Acad Sci U S A. 2002. PMID: 11867732). The high expression would contribute to the tumour but its reduction may not have an immediate effect in cell growth, or could be compensated by other PIAS (1,3 or 4). We want to address this hypothesis in future studies in PTC. On the other hand, all ATCs have mutated p53 genes. In these cells, the transcription-modifying function of PIAS2b may be no longer important in maintaining cancer. Instead, the alternative function of PIAS2b, which is its key role in mitosis, is revealed.

It is also important to note that the dosage-dependent effect of PIAS2b (death due to excess or lack of PIAS2b) is exclusive to ATCs and does not occur in DTCs. This explains the low but detectable levels in PDTC/ATC. This specificity is what leads us to think that PIAS2b is a good target for the treatment of anaplastic thyroid cancer.

A summary of this has been included in the Discussion.

We are currently working to further investigate the role of PIAS2b in ATCs. We believe that our findings have important implications for the understanding and treatment of this aggressive form of cancer.

We appreciate the reviewer's feedback and interesting suggestion, that we have now summarized in the discussion.

Fig 1G-I: I would suggest highlighting the driver mutation in the text box describing the genotype of the individual cell lines and confirm the mentioned genotypes. For FTC-238 the driver is PTEN p.130*, which is not mentioned. Please verify if the presence of a BRAF mutation is correct as this is not described for this cell line in the previous literature. For 8305C and BHT-101 the driver is BRAFV600E.

Response. We have followed your suggestion and highlighted the known mutated drivers in Figure 1. Some ATC cell lines, such as MB-1, have no cognate driver.

We apologize for the error in the previous version of the manuscript, which stated that FTC-238 had a SNIP in V502. The information was obtained from a public database. We have now investigated this issue and have determined that it is FTC-133 that has this variant. Additionally, since both cell lines come from the same patient, we have Sanger sequenced all our thyroid cancer cell lines for BRAF V502 and have found that they are

all WT. We have added this information to the status section to prevent any future confusion by other groups.

Fig 2G: Please clarify what the numbers under the X-Axis signify.

Response. In our collection, each patient is assigned a unique code based on the order in which they were entered into each of the pathological groups. The number together with the letter code indicates which patient/culture it is. For example, T-PC49 is patient/culture 49 of papillary carcinomas. This information has been specified in the legend and X-axis of Figure 2G.

Fig 4E: please provide the list of proteins. Suppl Table S3, under the XXX

Response. Figures 4 and 3 have been revised to incorporate a new proteomics assay. The key proteins that were identified in the SUMO- Qapture assay (Proteomic Assay 3), as well as the proteins that were immunoprecipitated in association with PIAS2b (Proteomic Assay 1) or entrained in the EGFP-PIAS2b pull-down (Proteomic Assay 2), have been detailed in Table S3 and S4, that have also been updated to make it easier to find information on the identified proteins, both quantitatively and in terms of their pathway analysis. In Figure 4G, the cross analysis of Proteomic Analysis 1+2+3 (PIAS2-dependent SUMOylated candidates) is shown as a Venn's Diagram. The 20 common proteins (Crossing 1+2+3) are indicated at the right; from the 30 proteins that result from crossing 1+3 (the Proteomic Assay 2 is much less sensitive), we indicate selected proteins in the Figure 4G.

Fig 5 and in vivo xenograft studies:

(i) Could you specify the endpoint of your study.

Response. The approved protocol for preclinical trials (PCT) with our ATC-oPDX model had an endpoint of 9-10 weeks of growth and observation of the tumors, starting at the moment of implantation. From the beginning of week 9, daily monitoring is required until completion or reaching week 10.

This information has been added to the Supplementary methods, with details of the protocol, endpoint, primary purpose, and secondary purpose of each PCT. In the new version, we have performed two new PCT assays (PCT2 and PCT3) in NSG mice. PCT2 and PCT3 were with cells from patient 3 (T-UC7).

We also tried patient 2' cells in NSG mice (T-UC3), but again they did not grow.

PCT2 demonstrates the efficacy of the PIAS2-dsRNAi therapy also in ATC-oPDX of this patient (Primary purpose), and that it does not appear to be toxic or create resistance at repeated injections (Secondary purpose).

In the last trial that we presented (PCT3), we requested to the Ethical committee an extension authorization until week 11. This was to allow us to start treating later in week 7, with larger tumors (Primary purpose). This more closely resembles many patients with ATC, which is why we call it "Compassionate Use." At the same time, since the tumors already had a lot of signal in the IVIS, we were able to add an off-treatment period to observe tumor regrowth (Secondary purpose).

The Lowess curves from PCT3 indicate that the treatment may even be effective in these advanced tumors, and that as soon as the treatment is discontinued, the tumor immediately regrows. We believe that these data reinforce a possible future move to stages closer to the patient.

(ii) The title of the section is in comparison to all other sections very vague, consider specifying.

Response. The title has been modified to make it more direct.

(iii) Your outcome measure is repeated IVIS assessments. For a proof-of-concept study to determine differences in tumor burden in general is fine, however to classify the outcome by RECIST or Choi criteria seems an overinterpretation of the data. For Choi or RECIST measurements actual scans should be provided. However, showing the difference in tumor volume (fig 5F) seems sufficient to show a general effect of PIAS2b silencing on ATC growth.

Response. The RECIST-like or Choi-like classification was an attempt to more rigorously protocolize preclinical trials in mice, similar to clinical trials. The IVIS is a scan-like tool used in mice, and its intensity is dependent on the blood flow in the tumor.

However, following the reviewer's suggestion, since it has not been internationally standardized and to reduce the length of the manuscript, this section has been removed from the second version.

Fig. 6: While it is interesting that PIAS2b silencing does not only affect ATC but also pancreatic ductal adenocarcinoma, anaplastic lung carcinoma, and undifferentiated gastric carcinoma cell proliferation, the data shown might not warrant an individual figure.

Response. The origin of these experiments is in the many discussions during these years with other cancer scientists, who repeatedly asked whether the PIAS2b-dsRNAi effect only happened in anaplastic thyroid carcinoma (ATC) cells or also in other non-thyroid carcinoma cells. Now that our institution has patented this research, we have begun meeting with non-profit foundations that might be interested in funding the extension of this research to patients. Again, the question is always the same: is it only for ATC patients? This is also the question that oncological pharmaceutical companies ask.

We believe that the results leave the field open so that anaplastic cells from other organs can also respond and with the same mechanism. Of course, we submit to the decisions of the Editor on this point.

Reviewers' Comments:

Reviewer #1:

Remarks to the Author:

While the manuscript has improved in some parts, there are still several shortcomings.

Most importantly, and the criticisms also goes to the editorial strategy of Nature journals, packing manuscripts with such huge amount of information does not improve anything else but the journal impact factor. Such a manuscript is very hard to read (and to assess!) for both the readership and the reviewers. Then I do not wonder that manuscripts need to be retracted afterwards.

Still the figures are way too overcrowded, some things with more than ten panels. The shown data needs to be summarized in a better way and the number of panels reduced. Also, many labels won't be readable due to too small font size, enforced by lack of space. And the figure captions are a nightmare. It is like reading two manuscripts: a) the main text, b) the figure captions. On top of that, there is a lot of empty space in the figures, making the shown information even more cramped.

The description of the proteomics analyses has much improved. However, I still miss the mentioning of the used modifications in the (supplementary) text as stated in the response to the reviewers: "Thus, in the search, we only specified trypsin digestion, Cys-carbamidomethylation, and Ubiquitination modifications"

Reviewer #2:

Remarks to the Author:

The authors have properly addressed many of my comments and the manuscript including the microscopy part has now significantly improved. Three issues remain to be addressed.

1. The request for orthogonal validation of dsRNAi results with CRISPR-Cas9 has not been carried out. The authors point out potential challenges with CRISPR-Cas9 technology. Reviewer 4 suggests alternative approaches including a degron-based approach. The degron-based approach has also not been included in the revised manuscript. The authors use shRNA as alternative, but these results are not convincing as the GAPDH loading control is also reduced (Figure S2D). Orthogonal validation of dsRNAi results therefore remains a critical issue.

2. Biochemical sumoylation experiments are not state of the art (Figure 4H-N). The blots are still rather dirty, indicating that the SUMO enrichment is poor. Some of the size shifts observed are not compatible with sumoylation since the size shifts do not match the size of SUMO. Please use proper His-SUMO pull down and Western methodology commonly used in the SUMO field to improve this part of the paper.

3. Molecular mechanism are still poorly addressed. Sumoylation mutants of PSMC5 and TUBB3 should be used to study the relevance of sumoylation of these proteins for establishing a correct spindle and proper chromosome-microtubule attachment at the kinetochores.

Reviewer #4:

Remarks to the Author:

In general, the authors responded adequately to the comments. I still think that the manuscript requires shortening and focusing on the most important and most convincing results. Below are some minor points:

1. The difference between PIAS2a, p75-PIAS2b and p95-PIAS2b is still not very clear from the figure and the text: Few of the RNA isoforms are protein-coding, and only two express full length protein: PIAS2 beta (PIAS2b) and PIAS2 alpha (PIAS2a). This implies that only 2 forms of PIAS, PIAS2a and PIAS2b exist with PIAS2b being the longer version. Then, suddenly the PIAS2b-p75

and p95 forms appear. There is no reference regarding the coding sequence/difference between the PIAS2b-p75 PIAS2b-p95 forms. The explanation in the rebuttal should be included: Thus, we assume that the molecular weight of the two bands detected with mPIAS2 is due to different levels of PTM on PIAS2b. The band at 75 kDa is less PTM than the band at 95 kDa. What are the expected Mw of each isoform?

2. Still the reduction of PIAS2b in the rescue experiment Fig1 K (vector ns-ds RNAi) vs vector dsRNAi) is minimal but has a large affect on proliferation. In contrast, the reduction of PIAS2 protein level observed in 1L with the different siPOOLS is larger, while the effect on growth/proliferation seems less important, lower than in 1K, and in addition, the PIAS2 protein levels for the dsRNAi are not shown, something that would allow clear comparison.

3. Elution of the protein from the GFP-trap result in denatured protein that may or may not refold after neutralization. Therefore, the GFP-trap-protein transfection approach is not very concluding in my opinion.

4. Fig 3O seems not cut correctly, the full gel image is required.

Reviewer #5:

Remarks to the Author:

The revised manuscript entitled 'dsRNAi-mediated silencing of PIAS2beta specifically kills anaplastic carcinomas by mitotic catastrophe' by Rodrigues JS et al is improved. My main concern remains the length of the manuscript. While I understand that additional data was included based on reviewer comments, the manuscript overall should be more concise. One example is the thyroid cancer section in the introduction. The relevant information can be summarized in 1 paragraph and several recent reviews can be referenced.

Minor comments:

The last section of the third paragraph in the introduction is misleading. ('ATC cases share common mutated pathways but not specific gene mutations....within each ATC tissue is high.'). For example, BRAFV600E and specific TERT promoter mutations are very common in ATC patients. I recommend clarifying this passage.

Fig 1 G-H: thank you for clarifying. I suggest replacing 'Genetic status' with genotype.

Point-by-point Response to Reviewers' Comments

Reviewer #1 (Remarks to the Author):

While the manuscript has improved in some parts, ...

Response: Thank you again for the valuable feedback. We have worked hard to improve the quality of our manuscript.

...there are still several shortcomings.

Most importantly, and the criticisms also goes to the editorial strategy of Nature journals, packing manuscripts with such huge amount of information does not improve anything else but the journal impact factor. Such a manuscript is very hard to read (and to assess!) for both the readership and the reviewers. Then I do not wonder that manuscripts need to be retracted afterwards.

Still the figures are way too overcrowded, some with more than ten panels. The shown data needs to be summarized in a better way and the number of panels reduced. Also, many labels won't be readable due to too small font size, enforced by lack of space. And the figure captions are a nightmare. It is like reading two manuscripts: a) the main text, b) the figure captions. On top of that, there is a lot of empty space in the figures, making the shown information even more cramped.

Response: In response to Reviewer 1's concerns regarding the figures, and following indications by the Editor, we have thoroughly revised and reorganized the figure presentation to enhance clarity and conciseness. To address the issue of overcrowded figures, we have expanded the number of figures from six to ten, effectively reducing the number of panels per figure while ensuring that all relevant information is presented. Additionally, we have moved some data to supplementary figures, further decongesting the main figures. This reorganization has also led to a significant reduction in the length of figure legends, making them more concise and easier to follow. We further reduced by simplifying the descriptions on the legends.

Furthermore, we have increased the font size in all figures to at least 6 points, as recommended by the Editor, to ensure legibility. Since we have to fit every figure to a width format of 210 mm, empty space has disappeared.

To enhance comprehensiveness, we have also expanded the supplementary figures to eleven and added individual legends to each supplementary figure, facilitating their interpretation and evaluation.

The description of the proteomics analyses has much improved. However, I still miss the mentioning of the used modifications in the (supplementary) text as stated in the response to the reviewers: "Thus, in the search, we only specified trypsin digestion, Cys-carbamidomethylation, and Ubiquitination modifications".

Response: The sentence has been directly added in the Supplementary Extended Methods.

Reviewer #2 (Remarks to the Author):

The authors have properly addressed many of my comments and the manuscript including the microscopy part has now significantly improved.

Response: Thank you again for the valuable feedback, and your improving suggestions. We have worked hard to improve the quality of our manuscript.

Three issues remain to be addressed.

1. The request for orthogonal validation of dsRNAi results with CRISPR-Cas9 has not been carried out. The authors point out potential challenges with CRISPR-Cas9 technology. Reviewer 4 suggests alternative approaches including a degron-based approach. The degron-based approach has also not been included in the revised manuscript. The authors use shRNA as alternative, but these results are not convincing as the GAPDH loading control is also reduced (Figure S2D). Orthogonal validation of dsRNAi results therefore remains a critical issue.

Response: We have explained our in depth discussions concerning the feasibility of using a degron-based approach in the study of the SUMOylation of proteasome subunits, like PSMC5. As we explained in the previous revision, many concerns regarding design and controls prevented us to use this proteasome-dependent approach in the study of a proteasome protein.

But we have indeed performed orthogonal validation, using siPOOL transfections (current version Figure 2C and 2D) and shRNA populations (current version, Figure S2B-2C-2D).

- shRNA: Induction of *PIAS2*-shRNA with Doxycycline blocks cell growth (Figure S2B). The observed cell growth inhibition triggered by *PIAS2*-shRNA knockdown (Figure S2B) is preceded by a decline in *PIAS2* mRNA expression (Figure S2C) and protein levels (Figure S2D). We agree with the reviewer that the loading control GAPDH was downregulated by Doxycycline in both cell populations, Scr-shRNA and *PIAS2*-shRNA. We added another loading control (β -actin) to the shRNA experiments shown in Supplementary Figure 2D. We present two independent experiments with two film exposures. This addresses Reviewer #2's concern about the potential effects of doxycycline treatment on GAPDH expression, and confirms that the downregulation of *PIAS2b* protein after doxycycline treatment is specific to *PIAS2*-shRNA cells.
- *PIAS2b* siPOOL: Transfection of *PIAS2b* siPOOL leads to cell growth reduction, although with less potency than *PIAS2b*-dsRNAi (Figure 2C). This is concordant with a faster action of *PIAS2b*-dsRNAi reducing *PIAS2b* protein compared to *PIAS2b* siPOOL (Figure 2D)

Finally, *PIAS2b*-dsRNAi effect is rescued by exogenous *PIAS2b* (in two versions) but not by the point mutant, enzymatically dead, *PIAS2b*-C236A.

2. Biochemical sumoylation experiments are not state of the art (Figure 4H-N). The blots are still rather dirty, indicating that the SUMO enrichment is poor. Some of the size shifts observed are not compatible with sumoylation since the size shifts do not match the size of SUMO. Please use proper His-SUMO pull down and Western methodology commonly used in the SUMO field to improve this part of the paper.

Response: This has been an important suggestion, and we thank the reviewer for it.

We have repeated the experiments showed in the current Figure 7A-TB (former Figure 4H-4N) that were performed using non-denaturing extracts containing SUMO-protease inhibitors showed. We repeated those experiments in cells transfected with 10xHis-SUMO1-IRES-GFP, extracted in denaturing conditions, and followed the protocol for Nickel-NTA-beads pull downs to study direct SUMO1ylation (current Figure 7C): PSMC5 is specifically SUMO1ylated 6 hours after release (DT/R-6h) (Figure 7C), and this SUMO1ylation is blocked in the presence of *PIAS2b*-dsRNAi (Figure 7C). TUBB3 is already SUMO1ylated at DT/R-0h but increase its SUMO1ylation at DT/R-6h (Figure 7C); SUMO1ylation is greatly reduced in the presence of *PIAS2b*-dsRNAi (Figure 7C). While both total and His-SUMO1ylation increase at DT/R-6h, in cells treated with *PIAS2b*-dsRNAi, total SUMO1ylation is reduced, and mono-10xHis-SUMO1 is accumulated.

These results confirmed the previous findings and indicate the important role of *PIAS2b* in the SUMOylation of both proteins in mitosis of anaplastic thyroid cells.

3.Molecular mechanism are still poorly addressed. Sumoylation mutants of PSMC5 and TUBB3 should be used to study the relevance of sumoylation of these proteins for establishing a correct spindle and proper chromosome-microtubule attachment at the kinetochores.

Response: These ultra-molecular experiments are interesting but completely out of the scope of the current manuscript.

In this manuscript we wanted to propose a new target for Anaplastic Thyroid Carcinoma, effective in vitro in cell lines and patients' primary cultures, and also in orthotopic patient-derived xenografts in mice. We demonstrated a mechanism related to mitotic catastrophe, and found markers of response common to the in vitro and in vivo treatments.

We showed that *PIAS2b*-dsRNAi could be an effective therapy for all anaplastic carcinomas, independent of the tissue of origin, and not only thyroid.

We showed that *PIAS2b* has a role on the assembly of the mitotic spindle, SUMO1ylating proteasome and microtubule-spindle related proteins. When *PIAS2b*-dsRNAi is acting, PSMC5 and TUBB3 are less SUMOylated leading to untimely proteasome activation inducing mitotic catastrophe.

The key Lysine where the two chosen targets, PSMC5 or TUBB3, are SUMOylated could be difficult to identify. Below we show the data retrieved from the massive study of Hendriks et al (Hendriks IA, Lyon D, Young C, Jensen LJ, Vertegaal ACO, Nielsen ML. Site-specific mapping of the human SUMO proteome reveals co-modification with phosphorylation. Nature structural & molecular biology 24, 325-336, 2017). At least 12 Lysines were found SUMOylated in this study in relation to G2/M.

On the other hand, in a recent EMBO workshop with many key experts in SUMOylation (SUMOylation: From discovery to translation, Povoá de Varzim, 25th-28th September 2023), many groups pointed out that Lysine to Arginine mutations are often useless to generate SUMOylation deficient mutants because when the preferred lysine is not available for conjugation, the SUMO conjugation occurs pleiotropically in a Lysine nearby playing the same function.

Since in our manuscript the rule was to maintain the essential characteristics of the cellular systems used, as similar to the original patients as possible, we think that these molecular aspects had to be developed in future works.

(Data retrieved from Hendriks et al, Nature Structural Biology 2017 for PSMC5)

Protein and site information						MS-related scoring information					SUMO-proximal sequence information									
Protein	Protein names	Gene names	Position	Fractional prote in intensity	Score (SUMO site)	U2OS MG132 (MS/MS detection)	PEP	Score	Delta score	Score for localization	Score difference	Amino acid -1	Amino acid	Amino acid +2	K	X	E	[I V L] K x E	Sequence window	Sequence window (51 AA, Unique)
P62195	26S proteasome regulatory subunit 8	PSM C5	389	0.9%	88.1	+	4.46E-19	15.3	12.27	97.3	.3	A	K	M	0	0	0	RRVHVTQE DFEMAVAK VMQKDSEK NMSIKKL	TEAGMYALRERRVH VTQEDFEMAVAKVM QKDSEKNMSIKKLW K	
P62195	26S proteasome regulatory subunit 8	PSM C5	360	2.2%	88.0	+	3.24E-11	15.3	11.39	41.0	.0	V	K	V	0	0	0	RKIAELMPG ASGAEVKG VCTEAGMY ALRERR	KMNLTRGINLRKIAE LMPGASGAEVKGV CTEAGMYALRERRV HVTQEDFEM	
P62195	26S proteasome regulatory subunit 8	PSM C5	314	1.3%	87.6	+	2.67E-04	14.7	66.5	82.1	.1	R	K	E	1	0	0	DILDSALLR PGRIDRKIE FPPNNEEA RLDIL	IKVIMATNRIDILDSA LLRPGRIDRKIEFPP PNEEARLDILKIHSR KMNL	
P62195	26S proteasome regulatory subunit 8	PSM C5	38	0.7%	86.6	+	3.68E-18	16.1	11.19	101.5	.5	D	K	Q	0	0	0	YYLSKIEEL QLIVNDKSQL NLRRLQAQ RNELN	EGKAGSGLRQYYLS KIEELQLIVNDKSQL LRRLLQAQRNELNAK VRLLEEL	
P62195	26S proteasome regulatory subunit 8	PSM C5	170	1.9%	77.5	+	1.41E-05	11.4	68.5	40.1	.1	V	K	P	0	0	0	DKQIKEIKE VIEPLVKHP ELFEALGIA QPKG	DSTYEMIGGLDKQIK EIKEVIEPLVKHP ELFEALGIAQPKG PPGTG	
P62195	26S proteasome regulatory subunit 8	PSM C5	55	0.8%	70.4	+	1.04E-02	10.7	40.6	100.7	.7	A	K	R	0	0	0	QNLRLQA QRNELNAK VRLLEELQ LLQEQQ	ELQLIVNDKSQLNR RLQAQRNELNAKVR LLEELQLLQEQQS YVGEVVRAM	
P62195	26S proteasome regulatory subunit 8	PSM C5	290	4.5%	67.0	+	6.68E-03	8.4	23.5	84.2	.2	I	K	I	0	0	0	ELLNQLDGF EATKNIKVI MATNRIDIL DSAL	GDSEVQRTMLELLN QLDGFATKNIKVIM ATNRIDILSALLRP GRIDRKI	
P62195	26S proteasome regulatory subunit 8	PSM C5	125	1.1%	63.2	+	1.67E-03	7.2	40.3	67.2	.2	H	K	L	0	0	0	PNCRVALR NDSYTLHKI LPNKVDPLV SLMMV	DNKIDINDVTPNCRV ALRNDSYTLHKILPN KVDPLVSLMMVEKV PDSTYEM	

	sub unit 8																			
P62195	26S protease regulatory subunit 8	PSM C5	397	1.5%	61.7	+	7.39E-03	76.7	32.0	73.9	35.7	E	K	M	0	0	DFEMAVAK VMQKDSEK NMSIKKLW K	RERRVHVTQEDFE MAVAKVMQKDSEK NMSIKKLWK		
P62195	26S protease regulatory subunit 8	PSM C5	101	0.4%	61.2	+	2.82E-04	83.4	62.0	83.4	83.4	D	K	I	0	0	LVKVHPEG KFVVDVDK NIDINDVTP NCRVAL	VVRAMDKKKVLVKV HPEGKFVVDVDKNI DINDVTPNCRVALR NDSYTLHKI		
P62195	26S protease regulatory subunit 8	PSM C5	142	0.0%	51.2	+	1.64E-04	79.4	63.4	79.4	79.4	E	K	P	0	0	LPNKVDPLV SLMMVEKV PDSTYEMIG GLDKQ	RNDSYTLHKILPNKV DPLVSLMMVEKVPD STYEMIGGLDKQIKE IKEYIEL		
P62195	26S protease regulatory subunit 8	PSM C5	330	0.4%	48.5	+	2.39E-02	50.1	26.8	50.1	50.1	L	K	H	0	0	IEFPPNNEE ARLDILKHS RKMNLTRGI NLR	LLRPGRIDRKIEFPP PNEEARLDILKHSR KMNLTRGINLRKIAE LMPGAS		

Reviewer #4 (Remarks to the Author):

In general, the authors responded adequately to the comments.

Response: Thank you again for the valuable feedback. We have worked hard to improve the quality of our manuscript.

I still think that the manuscript requires shortening and focusing on the most important and most convincing results.

Response: Striking a balance between manuscript length and comprehensiveness can be a challenging task, especially when incorporating additional findings. The reviewers' requests for additional research and confirmatory results enhances the depth and robustness of the findings, although simultaneously increases the overall extension. This creates a conundrum, as all reviewers requested both greater elaboracy and brevity. We tried to solve this to the extent of our ability. We think that have reached a good compromise.

Below are some minor points:

1. The difference between PIAS2a, p75-PIAS2b and p95- PIAS2b is still not very clear from the figure and the text: Few of the RNA isoforms are protein-coding, and only two express full length protein: PIAS2 beta (PIAS2b) and PIAS2 alpha (PIAS2a). This implies that only 2 forms of PIAS, PIAS2a and PIAS2b exist with PIAS2b being the longer version. Then, suddenly the PIAS2b-p75 and p95 forms appear. There is no reference regarding the coding sequence/difference between the PIAS2b-p75 PIAS2b-p95 forms. The explanation in the rebuttal should be included: Thus, we assume that the molecular weight of the two bands detected with mPIAS2 is due to different levels of PTM on PIAS2b. The band at 75 kDa is less PTM than the band at 95 kDa. What are the expected Mw of each isoform?

Response: Thank you for emphasizing this point, which was not sufficiently clear in our former version, but has occupied long months of our work during the initial years, trying to understand the bands recognized by mPIAS2 and rPIAS2b in the westerns.

We have included in the Results text, and in the Supplementary Figure S1 Figure Legend the Uniprot Molecular Weights for both PIAS2 isoforms, PIAS2a (MW 63,396 Da) and PIAS2b (MW 68,240 Da).

We also have reformulated the paragraph as follows:

< Two antibodies, mPIAS2 and rPIAS2b, were validated for western blot and immunostaining in low-expression anaplastic cells. With a classic extraction buffer (e.g. including phosphatases inhibitors), mPIAS2, able to detect an intense band of transfected Flag-PIAS2a (Uniprot expected MW 63,396 Da), did not detect endogenous PIAS2a in any ATC cell line (Figure S1G). Also, mPIAS2 detected well exogenous transfected fusion PIAS2b proteins, and detected two endogenous PIAS2b protein isoforms (Uniprot expected MW 68,240 Da), intensely p75- and less intense p95-PIAS2b. rPIAS2b was better at detecting p95-PIAS2b, and less well p75-PIAS2b (Figure S1G). Thus, we assume that the molecular weight of the two PIAS2b bands detected is due to different levels of PTM on PIAS2b. >

2. Still the reduction of PIAS2b in the rescue experiment Fig1 K (vector ns-ds RNAi) vs vector dsRNAi) is minimal but has a large affect on proliferation. In contrast, the

reduction of PIAS2 protein level observed in 1L with the different siPOOLs is larger, while the effect on growth/proliferation seems less important, lower than in 1K, and in addition, the PIAS2 protein levels for the dsRNAi are not shown, something that would allow clear comparison.

Response: The reviewer is comparing western blots from different sections that are performed in different conditions. The former Fig 1K (currently Figure 2B) is performed in 8305 cells double and consecutively transfected, first with the plasmid bearing the fusion proteins (EGFP, EGFP-PIAS2b, EGFP-PIAS2b C362A) and in the next day with the dsRNAi (ns-, *PIAS2b*). The western is difficult, because the amount of fusion protein tolerated by the ATC cells is low, as explained in Fig S1W-S1Y. Extracts were collected at Day 2 after dsRNAi, because detection of transient transfected proteins could be lost in dividing cells (half amount in each division).

On the other hand, the former Fig 1L (currently Figure 2C) shows the effect of both siPOOLs at Day 4 after transfection, because this was the day of maximal effect that we had detected.

But the reviewer asks about the different potency of the two RNAi systems to reduce PIAS2b. We have added a new section comparing *PIAS2b*-dsRNAi to *PIAS2b* siPOOL, both transfected in parallel in 8305C cells seeded together. We have observed that *PIAS2b*-dsRNAi has a faster effect downregulating PIAS2b already detected by western at Day 2. *PIAS2b* siPOOL needs until Day 3 after transfection to downregulate PIAS2b. Since proliferation is measured at Day 6 for both treatments, the most effective at this time-point was *PIAS2b*-dsRNAi.

We have added this to Results, new Figure 2D, since time-course seems an important information for future applications.

3. Elution of the protein from the GFP-trap result in denatured protein that may or may not refold after neutralization. Therefore, the GFP-trap-protein transfection approach is not very concluding in my opinion.

Response: Thank you for this appreciation. These stainings were performed to check that exogenous EGFP-PIAS2b was localized in the spindle, where we found the staining of endogenous PIAS2b both with mPIAS2 or rPIAS2b antibodies. Detection was not so easy due to the small amount of pEGFP-PIAS2b tolerated by ATC cells, and thus we try to enhance detection. We have found similar staining in mitosis using chicken anti-EGFP antibody in previously transfected pEGFP-PIAS2b, or directly transfecting purified EGFP-PIAS2b fusion protein transfected with Xfect. Although mitotic spindle was stained, the quality is low due to background and spurious plasma membrane staining. Thus, the majority of these results have been moved to the current Figure S4E.

4. Fig 3O seems not cut correctly, the full gel image is required.

Response: Figure 3O western has been performed in the CAL-62 ATC cell line. We had standardized our PIAS2 immunoprecipitation in 8305C, and were not sure that it was going to work similarly in this other cell line. Thus, we first hybridized the western using mPIAS2, and once we saw the strong immunoprecipitated specific bands, we rehybridized for AURKA. Unfortunately, there was a strong Ig-G band around 50 kDa, that remained in the western:

1) Anti-mPIAS2:

2) Stripping, followed by Rehybridization for AURKA:

In any case, since this result is redundant, and to help to simplify figures, in the current version IP-anti-PIAS2 in CAL-62 results has been moved to Figure S5A.

In addition, a file containing all full gels in main Figures and Supplementary Figures have been added (PowerPoint file: Rodrigues_Nature Comm_Source Data_western films).

Reviewer #5 (Remarks to the Author):

The revised manuscript entitled ‘dsRNAi-mediated silencing of PIAS2beta specifically kills anaplastic carcinomas by mitotic catastrophe’ by Rodrigues JS et al is improved.

Response: Thank you again for the valuable feedback. We have worked hard to improve the quality of our manuscript.

My main concern remains the length of the manuscript. While I understand that additional data was included based on reviewer comments, the manuscript overall should be more concise. One example is the thyroid cancer section in the introduction. The relevant information can be summarized in 1 paragraph and several recent reviews can be referenced.

Response: This is not so easy to address as it seemed. ATC have had a burst of new research on diagnostics, treatment and follow-up in the last years. This is reflected in the 700 publications on the topic in PubMed during the last five years 2019-2023, 300 of them in the last two years 2022-2023. Moreover, we are a multidisciplinary team with different considerations on the important information depending on the speciality. At the same time, we appreciate the comment of the Reviewer and the need to improve readability. We have deleted some sentences in the Introduction and added recent literature reviews to provide a more concise summary of the relevant information.

Minor comments:

The last section of the third paragraph in the introduction is misleading. (‘ATC cases share common mutated pathways but not specific gene mutations within each ATC tissue is high.’) For example, BRAFV600E and specific TERT promoter mutations are very common in ATC patients. I recommend clarifying this passage.

Response: The sentence has been reformulated as follows:

< ATC cases are characterized by a high mutation burden in multiple genes^{15, 16, 17, 18}. These mutations are heterogeneous across ATC cases, with few being shared among all tumors. Despite this diversity, certain genes exhibit a high prevalence of mutations like both early driver events (e.g., BRAF V600E, 10–40%; RAS mutations, 10–30%) and late molecular events (e.g., TP53 mutations, 40–80%; TERT promoter mutations, 30–75%)^{19, 20}. However, despite the great variability in the mutational profile between patients, the tumors share common activated signaling pathways^{19, 20}, and the percentage of cells sharing the common mutational profile within each ATC tissue is high²¹. >

Fig 1 G-H: thank you for clarifying. I suggest replacing ‘Genetic status’ with genotype.

Response: Thank you for this suggestion, that increases language precision. Genotype has been replaced by “Genetic status” in Figures 1G-1H-1I, 3G-3H, 10A-10C-10E. Please note that we use the reformulated Figure number of this last version.

Reviewers' Comments:

Reviewer #2:

Remarks to the Author:

The authors have carried out an excellent revision and I support publication of the manuscript after adding the negative control for the His-SUMO1 experiments.

1. The authors have sufficiently improved orthogonal validation of dsRNAi results. I agree that beta-Actin is a better control compared to GAPDH since GAPDH is affected by the knockdown of PIAS2 and beta-Actin is not affected.

2. I understand that the authors have already carried out a lot of work and that sumoylation mutants of PSMC5 and TUBB3 need to be developed in a future project.

3. I would like to thank the authors for carrying out His-SUMO1 experiments. Unfortunately, the pattern of bands for some antibodies is complex and negative controls lacking His-SUMO1 are missing in these experiments. These negative controls lacking His-SUMO1 need to be included to dissect which of the bands represent sumoylated proteins and which bands are merely background. I have to insist on stringent negative controls to enable solid experimental conclusions.

Reviewer #4:

Remarks to the Author:

In their new revised version, the authors made a sincere effort to respond to the comments of the reviewers. I have still an issue with the TUB3 SUMOylation, but the main issue is the length and the form of the manuscript, that remains large and complex.

For example, I am less not convinced by the TUB3 sumoylation results. Why isn't the TUB3 protein (unsumoylated) detected after the TUB3 IP pulldowns (+/-PIASb) in the TUB3-blot, and only sumoylated forms are, Fig. 7B? Or is the annotation uncorrect (should be white circle). Without the amount of immunoprecipitated TUB3 this, it may be that just the level of all proteins is reduced in the IP in the PIASb(-) situation. How do we know that the high molecular weight bands correspond to TUB3 and are not background? Also in the nickel pulldowns, I don't see any significant reduction in the sumoyation of TUB3 in absence of PIASb (7C, +/- PIASb).

In the inputs of the nickel pull downs, 7C, (+/- PIASb), the TUB3 band runs fine but in the time course on the left, all bands run below 43kDa.

Also, the authors state "From top to bottom order of re-stripped westerns" for me that means that the first western is not stripped, so the PIAS bands should not be visible in the first blots.

In my opinion, the authors should make 3 smaller papers out of it, focusing every time on a different aspect and on solidifying their data, with few, but crystal-clear experiments. For the part of the document that lies within my expertise, the cell biology experiments are already convincing, however the biochemistry is less obvious.

Reviewer #5:

Remarks to the Author:

Thank you for revising the manuscript and your answers to the reviewer requests.

While I understand that this is a complicated topic and you have been asked for additional experiments and information, my remaining concern is that the manuscript is too convoluted and too long. The figures, especially figures 6 and 7 are very busy. All in all I believe that the manuscript shares compelling results that are relevant and important for the field but I am

concerned that it is not concise enough.

POINT-BY-POINT RESPONSE TO REVIEWERS' COMMENTS

Reviewer #2 (Remarks to the Author):

The authors have carried out an excellent revision and I support publication of the manuscript after adding the negative control for the His-SUMO1 experiments.

Response: We sincerely thank the reviewer for the thorough review and insightful comments and for taking the time to provide such a deep analysis of our work. The previous revisions have significantly strengthened our manuscript by helping us to clarify and better present our data.

1. The authors have sufficiently improved orthogonal validation of dsRNAi results. I agree that beta-Actin is a better control compared to GAPDH since GAPDH is affected by the knockdown of PIAS2 and beta-Actin is not affected.

Response: We appreciate the reviewer's suggestion regarding the additional controls for shRNA data included in the Supplementary Figures. These data indeed reinforce our hypothesis. However, we acknowledge that the main figures are already quite data-dense, and we believe the placement of the shRNA results—and their controls, in the supplementary material strikes a balance between clarity and conciseness.

2. I understand that the authors have already carried out a lot of work and that sumoylation mutants of PSMC5 and TUBB3 need to be developed in a future project.

Response: As the reviewer acknowledges, this line of deep molecular inquiry falls outside the scope of the present manuscript and would require substantial further investigation. We are happy to report that we have already begun to implement strategies to address this question in future work.

3. I would like to thank the authors for carrying out His-SUMO1 experiments. Unfortunately, the pattern of bands for some antibodies is complex and negative controls lacking His-SUMO1 are missing in these experiments. These negative controls lacking His-SUMO1 need to be included to dissect which of the bands represent sumoylated proteins and which bands are merely background. I have to insist on stringent negative controls to enable solid experimental conclusions.

Response: We thank the reviewer for their emphasis on stringent negative controls in the His-SUMO1 experiments.

We must explain that before performing the whole experiments with asynchronous, or DT/R synchronized cells in the absence or presence of doubly transfected with dsRNAi, we had standardized the technique, since it was new for the lab, and we wanted to performed it in 10xHis-SUMO1-transiently transfected cells, not in populations where all cells express the modified His-SUMO1.

To address this Reviewer's valuable feedback, we have included a new section in Suppl Figure S7a. This figure compares denaturing extracts from cells transfected with either 10xHis-SUMO1 or empty vector. Importantly, the membranes were sequentially probed (and re-probed) for all the antibodies used in the main Figure 7b (His.Tag, PSMC5, rPIAS2B, TUBB3, SUMO1).

On the left-hand side, the inputs for both conditions are shown, demonstrating no significant differences in some proteins (GAPDH, PSMC5, PIAS2b) but differences in

other, some expected (His-Tag, SUMO1, TUBB3). Importantly, on the right-hand side, the Ni-NTA pull-downs reveal no protein bands for any of the antibodies (His-Tag, PSMC5, rPIAS2B, mTUBB3, mSUMO1) when using the empty vector-transfected extracts. This confirms the specificity of the pull-down assay and clarifies which bands in the main Figure 7b represent SUMO1ylated or poly-SUMOylated proteins.

Reviewer #4 (Remarks to the Author):

In their new revised version, the authors made a sincere effort to respond to the comments of the reviewers. I have still an issue with the TUB3 SUMOylation, but the main issue is the length and the form of the manuscript, that remains large and complex.

We appreciate the reviewer's acknowledgment of the effort invested in improving the format and addressing concerns in the previous revisions. We understand the desire for conciseness. However, a comprehensive presentation of our findings necessitates including ten main figures and eleven supplementary figures with multiple panels to adequately showcase the complexity of the investigated biological processes.

It is important to note that reviewers often request additional data and methodologies, which can lead to a more complex manuscript. While we strive for clarity and conciseness, omitting crucial information could compromise the scientific rigor of the work, and significantly reducing the manuscript length while incorporating new data would be extremely challenging.

For example, I am less not convinced by the TUB3 sumoylation results. Why isn't the TUB3 protein (unsumoylated) detected after the TUB3 IP pulldowns (+/-PIASb) in the TUB3-blot, and only sumoylated forms are, Fig. 7B? Or is the annotation uncorrect (should be white circle). Without the amount of immunoprecipitated TUB3 this, it may be that just the level of all proteins is reduced in the IP in the PIASb(-) situation. How do we know that the high molecular weight bands correspond to TUB3 and are not background?

Response: We acknowledge the reviewer's valuable feedback regarding the potentially confusing nature of the section utilizing pull-downs after immunoprecipitation with mouse anti-TUBB3 antibodies.

We recognize the limitations of not presenting the input controls (to reduce the size of the figure) and the ambiguity surrounding the high-molecular weight bands observed in that section. Following a thorough review of the diverse post-translational modifications (acetylation, phosphorylation, polyglutamylation, polyglycylation, detyrosination) that can influence Tubulin migration patterns, and considering the technical challenges associated with Western blot reprobing with a mouse anti-TUBB3 antibody after using a mouse anti-SUMO1 antibody, we have opted to eliminate the previous section 7b.

In its place, we have further emphasized the Ni-NTA pull-down experiments (now presented in Figure 7b), which provide a more specific approach to identify SUMOylated proteins. Additionally, the control data for these pull-downs is now included in Suppl Figure S7a. We believe this revised approach strengthens the demonstration of the PIAS2b-dependent SUMO1ylation of our target substrates PMSC5 and TUBB3.

Also in the nickel pulldowns, I don't see any significant reduction in the sumoyation of TUB3 in absence of PIASb (7C, +/- PIASb).

Response: We have addressed the reviewer's concern regarding the lack of a pronounced difference in TUBB3 SUMO1ylation at DT/R-5h. In the revised Figure 7b, we now present two exposures of the TUBB3 gel lanes. This clearly demonstrates that

PIAS2b-dsRNAi treatment significantly reduces p70-kDa SUMO1ylated TUBB3, poly-SUMOylated TUBB3 (and less intensely p50-kDa-TUBB3) at DT/R-5h.

(See the next point for explanation about molecular weight of those TUBB3 bands).

In the inputs of the nickel pull downs, 7C, (+/- *PIASb*), the TUB3 band runs fine but in the time course on the left, all bands run below 43kDa.

Response: We thank the reviewer for pointing out the discrepancy in band molecular weight markings in the figure related to 10xHis-SUMO1 transfected cells. We have meticulously revised the markings, acknowledging the reviewer's observation.

Additionally, we now indicate that denaturing extracts, compared to non-denaturing extracts, can lead to the appearance of additional bands for proteins like TUBB3 in the input samples. There are actually three bands for TUBB3 at the Input: a higher band at ~p70-kDa, and the two bands at p50- and p42-kDa. This pattern is also shown in asynchronous extracts used for standardization (Figure S7a).

We hypothesize that the diverse post-translational modifications of Tubulins, many of which affect protein charge, contribute to altered migration patterns in non-denaturing extracts [Magiera MM, et al. Tubulin Posttranslational Modifications and Emerging Links to Human Disease. *Cell* (2018); Janke C, Magiera MM. The tubulin code and its role in controlling microtubule properties and functions. *Nat Rev Mol Cell Biol* (2020)]., and that SUMO-proteases are quite difficult to inhibit at least denaturing conditions are used. Tubulins beta (TUBB and TUBB4B) and alfa (TUBA1C) were also found specifically SUMO1ylated by *PIAS2* in a recent high-throughput study using SAAT to delineate E3-SUMO ligases substrate specificity in asynchronous U2OS cells [Salas-Lloret D, et al. SUMO-activated target traps (SATTs) enable the identification of a comprehensive E3-specific SUMO proteome. *Sci Adv* (2023)].

Critically, ~p70-TUBB3 band (55kDa-TUBB3 band + 1 SUMO1), the 55kDa-TUBB3 band (42kDa-TUBB3 + 1 SUMO1), and the higher molecular weight TUBB3 band (poly-SUMOylated TUBB3), appear in the Ni-NTA pull-downs at G2/M (DT/R-0h) and markedly increase after release, supporting the conclusion of SUMO1ylation and poly-SUMOylation during mitosis (DT/R-6h) (Figure 7b, left); and the three bands are strongly reduced with *PIAS2b*-depletion at DT/R-5h (Figure 7b, right). The p42kDa-TUBB3 does not appear in the Ni-NTA pull-downs (Figure 7b), indicating that is unSUMOylated TUBB3.

Also, the authors state "From top to bottom order of re-stripped westerns" for me that means that the first western is not stripped, so the *PIAS* bands should not be visible in the first blots.

Response: We understand that the Reviewer refers to westerns shown in Figure 7a. There, the first western is against SUMO1.

In a GFP-pull down, that requires non-denaturing extracts, all the proteins associated to EGFP-PSMC5 and SUMO1ylated, and present in the pull-downs in enough quantity for detection, could be revealed in a western blot anti-SUMO1.

In Figure 6j we had already shown the association of FLAG-PSMC5 to *PIAS2b*. Here, in Figure 7a, comparing the band patterns at mitosis (DT/R-6h) of the SUMO1 western blot to the one in PSMC5 and r*PIAS2b* western blots, it is clear that EGFP-PSMC5 and *PIAS2b* association also occurs, that this association is enhanced at mitosis where both proteins are SUMOylated, and that the association is decreased with *PIAS2b*-dsRNAi.

In my opinion, the authors should make 3 smaller papers out of it, focusing every time on a different aspect and on solidifying their data, with few, but crystal-clear experiments. For the part of the document that lies within my expertise, the cell biology experiments are already convincing, however the biochemistry is less obvious.

Response: We have communicated with the Editor regarding the recommendation to divide the manuscript into two or three smaller publications. The Editor emphasized that doing so would necessitate a complete new review process for each individual manuscript.

Given the substantial time already invested in revisions (approaching two years), and the uncertain prospect of these hypothetical smaller manuscripts meeting the Nature journal's high standards, we believe continuing with the current manuscript is the most efficient approach. Notably, there are young authors involved in this research, and their career progression depends on securing publication. On the other hand, although dividing it into several articles has its appeal, at the current time our priority is to make our data known to the scientific community and the complete development of its therapeutic potential at a preclinical level. The fact that there is no treatment for BRAF-*neg* anaplastic thyroid carcinomas is a big motivator in this regard.

We are confident that our manuscript, refined with the invaluable feedback from the reviewers, presents critical findings within the context of anaplastic thyroid carcinomas, mitosis in aneuploidy cells, and a potential new therapeutic strategy for anaplastic carcinomas. We acknowledge there are avenues for further investigation, which can be explored in future studies by ourselves and other research groups.

To ease slightly the Reviewer's concern, we want to indicate that in this last version we have deleted a whole section of figure 7 (previous section 7b), and we have condensed the Discussion to reduce the whole size of the manuscript.

Reviewer #5 (Remarks to the Author):

Thank you for revising the manuscript and your answers to the reviewer requests. While I understand that this is a complicated topic and you have been asked for additional experiments and information, my remaining concern is that the manuscript is too convoluted and too long. The figures, especially figures 6 and 7 are very busy. All in all I believe that the manuscript shares compelling results that are relevant and important for the field but I am concerned that it is not concise enough.

Response: We appreciate the reviewer's acknowledgment of the effort invested in improving the manuscript format in the previous revision. We understand their concern regarding manuscript length and complexity. It is important to note that reviewers often request additional data and methodologies, which can lead to a more complex manuscript. While we strive for clarity and conciseness, omitting crucial information could compromise the scientific rigor of the work.

We believe the current manuscript effectively balances comprehensiveness with readability. The most relevant findings are in the ten main figures, while the eleven supplementary figures are complementary for those interested in particular aspects of the *PIAS2b*-dsRNAi treatment or controls of the techniques.

To ease slightly the Reviewer's concern, we want to indicate that in this last version we have deleted a whole section of figure 7 (previous section 7b), and we have condensed the Discussion to reduce the whole size of the manuscript. Unfortunately, further reduction is not feasible without compromising the integrity of the data presentation in all aspects required by the Editor and complementary expertise Reviewers.